# Nitrogen availability regulates topsoil carbon dynamics after permafrost thaw by altering microbial metabolic efficiency

Leiyi Chen[1], Li Liu[1,2], Chao Mao[1,2], Shuqi Qin[1,2], Jun Wang[1,2], Futing Liu[1,2], Sergey Blagodatsky [3,4], Guibiao Yang[1,2], Qiwen Zhang[1,2], Dianye Zhang[1,2], Jianchun Yu[1,2] & Yuanhe Yang[1,2]

Input of labile carbon may accelerate the decomposition of existing soil organic matter (priming effect), with the priming intensity depending on changes in soil nitrogen availability after permafrost thaw. However, experimental evidence for the linkage between the priming effect and post-thaw nitrogen availability is unavailable. Here we test the hypothesis that elevated nitrogen availability after permafrost collapse inhibits the priming effect by increasing microbial metabolic efficiency based on a combination of thermokarst-induced natural nitrogen gradient and nitrogen addition experiment. We find a negative correlation between the priming intensity and soil total dissolved nitrogen concentration along the thaw sequence. The negative effect is confirmed by the reduced priming effect after nitrogen addition. In contrast to the prevailing view, this nitrogen-regulated priming intensity is independent of extracellular enzyme activities but associated with microbial metabolic efficiency. These findings demonstrate that post-thaw nitrogen availability regulates topsoil carbon dynamics through its modification of microbial metabolic efficiency.

[1] State Key Laboratory of Vegetation and Environmental Change, Institute of Botany, Chinese Academy of Sciences, 100093 Beijing, China. [2] University of Chinese Academy of Sciences, 100049 Beijing, China. [3] Institute of Agricultural Sciences in the Tropics (Hans-Ruthenberg-Institute), University of Hohenheim, 70599 Stuttgart, Germany. [4] Institute of Physicochemical and Biological Problems in Soil Science, Russian Academy of Sciences, Pushchino, Russia 142290. Correspondence and requests for materials should be addressed to Y.Y. (email: yhyang@ibcas.ac.cn)

A warming climate is predicted to induce massive carbon (C) release from thawing permafrost, triggering a positive C-climate feedback[1,2]. However, the magnitude of this feedback remains highly uncertain, with potential C release from the permafrost zone ranging from 37 to 174 Pg by 2100 [2]. The uncertainty in permafrost C feedback is partly attributed to the fact that ecosystem C balance is regulated by intricate plant-microbial-soil interactions after permafrost thaw[3–5]. In these interactions, enhanced plant C fixation may mitigate part of the soil C loss[1], but it may also induce soil C loss by accelerating the turnover of native soil organic matter (SOM), the so-called priming effect[6–9]. Particularly, accompanying permafrost thaw and enhanced plant productivity under climate warming, more roots can grow down into the previously frozen soil horizon[10] where a large amount of C is stored[11]. This change in plant root distribution may subsequently trigger the priming effect in deep soils and make the prediction of post-thaw ecosystem C dynamics more complex. Therefore, our knowledge of the magnitude and mechanism of the post-thaw priming effect is a prerequisite for accurate prediction of permafrost C dynamics under climate warming.

Thermokarst is an abrupt permafrost thaw process (ground surface collapse caused by thawing of ice-rich permafrost) that can dramatically impact soil and hydrologic properties[12–14]. Post-thaw changes in soil nutrients, moisture, texture and pH[15,16] may influence SOM turnover by altering the priming intensity[8,17,18]. Among these changes, the widespread increase in nitrogen (N) availability after permafrost collapse[16,19], driven by enhanced N mineralization[16] and the additional N released from thawing permafrost[20], may play an important role in regulating the priming intensity, since both vegetation growth and topsoil microbial activity in arctic[21,22] and alpine ecosystems[23–25] are N limited. Despite this recognition, the link between N availability and the priming effect in permafrost ecosystems remains ambiguous because previous studies have revealed a positive[26], a negative[27,28] and no effect[26,28] of N addition on priming intensity. The diverging views probably reflect the ecosystem differences in microbial C and N demands[29], which are closely related to the microbial physiological responses. Thus, to better understand this discrepancy, it is urgently needed to reveal the underlying mechanisms governing this C–N interaction, especially the role of microorganisms.

Microbial metabolic efficiency, the partitioning of C substrate between microbial biomass and carbon dioxide ($CO_2$) production, is a key microbial physiological property that determines the fate of soil organic C (SOC)[30,31]. By definition, microorganisms with higher microbial metabolic efficiency allocate more C to microbial growth than to respiration[32,33], resulting in a lower soil $CO_2$ flux[30,34,35]. Consequently, higher microbial metabolic efficiency promotes C stabilization in soils, while lower microbial metabolic efficiency favours C loss via microbial respiration[30]. Besides affecting C cycling process, microbial metabolic efficiency may also play an important role in governing C–N interaction. Adjustment in microbial metabolic efficiency is a common microbial response to natural or experimental variations in soil N availability[35]. Microbial metabolic efficiency generally increases with N availability[30,32]. This physiological adjustment can facilitate the microorganisms to cope with their N demand and maintain a balanced biomass C:N ratio[36]. Therefore, shift in microbial metabolic efficiency is one of the potential pathways that post-thaw N availability could regulate soil C release after thermokarst formation. However, to date, it remains obscure whether and how microbial metabolic efficiency regulates the priming effect under different N conditions.

Here, we hypothesize that elevated N availability after permafrost collapse inhibits the priming effect by increasing the microbial metabolic efficiency. To test this hypothesis, we used a thermokarst-induced natural N gradient on the Tibetan Plateau (Supplementary Fig. 1) and a subsequent N addition experiment to explore changes in the priming effect and microbial metabolic efficiency under different N conditions. While gradient studies are valuable for exploring variations in the priming effect along a natural N gradient, they cannot explicitly rule out other confounding factors that co-vary with N along the thaw sequence[13,14]. In contrast, although N manipulation experiments can explicitly reveal the N effect[26–28], the single high-N concentration involved in previous experiments[28,37] prevents us from understanding the trajectory of priming intensity along the natural N gradient. The combination of these two approaches could thus generate a comprehensive understanding of the role of microbial metabolic efficiency in C–N interactions. Our results demonstrate that soil N availability is negatively associated with the priming effect along the thaw sequence. The high priming intensity at late stage of collapse is reduced by subsequent N addition. This N-induced deceleration of soil C release is attributed to the increased microbial metabolic efficiency that occurs under high-N conditions.

## Results

**Changes in the priming effect along the thaw sequence.** The addition of $^{13}$C-labelled glucose significantly promoted the release of unlabelled $CO_2$ from soils at all stages of permafrost collapse ($P < 0.05$; Fig. 1a–d), resulting in positive priming along the thaw sequence. Given that no significant reduction in unlabelled C was found in microbial biomass after glucose addition (Supplementary Fig. 2), the increased unlabelled $CO_2$ release observed in our study could be mainly due to a decrease in the SOC pool, but not apparent priming via C substitution in microbial biomass[38,39]. Despite the overall positive response to glucose addition, there was a significant interaction between glucose and thaw time on SOM-derived C release ($P < 0.05$; Supplementary Fig. 3), indicating that the magnitude of the priming effect varied with thaw time. Specifically, the primed $CO_2$–C release decreased in early permafrost collapse but subsequently increased with collapse time and approached its maximum (1.79 mg $CO_2$–C $g^{-1}$C) at the late-stage site (16 years after collapse) (Fig. 1e). The relative priming effect exhibited a similar pattern along the thaw sequence (Fig. 1f).

**Linking abiotic and biotic factors to the priming effect.** Permafrost collapse had significant influences on abiotic and biotic parameters (Supplementary Table 1). The soil moisture significantly decreased in the vegetated patches after permafrost collapse ($P < 0.05$). Similarly, both SOC and soil total N (STN) content decreased at the late-stage site ($P < 0.05$; Supplementary Table 1). Despite the concentration of soil dissolved organic C (DOC) not exhibiting significant changes along the thaw sequence ($P = 0.06$), the concentration of soil total dissolved nitrogen (TDN) increased significantly during the early stage and subsequently dropped with collapse time, leading to a 20.5% reduction at the late-stage site ($P < 0.01$, Supplementary Table 1). In addition to soil properties, the microbial stoichiometry also varied along the thaw sequence, with a significantly lower microbial C:N ratio ($B_{C:N}$) at the late-stage site ($P < 0.05$). Correspondingly, microbial N limitation, indicated by the C:N imbalance between resources and microorganisms (both the total form ($R_{C:N}/B_{C:N}$) (Supplementary Fig. 4a) and the labile form ($R_{DOC:TDN}/B_{C:N}$) (Supplementary Table 1), increased with collapse time, resulting in maximal microbial N limitation at the late-stage site. Moreover, compared to other collapse stages, glucose addition induced a larger increase in C:N imbalance at

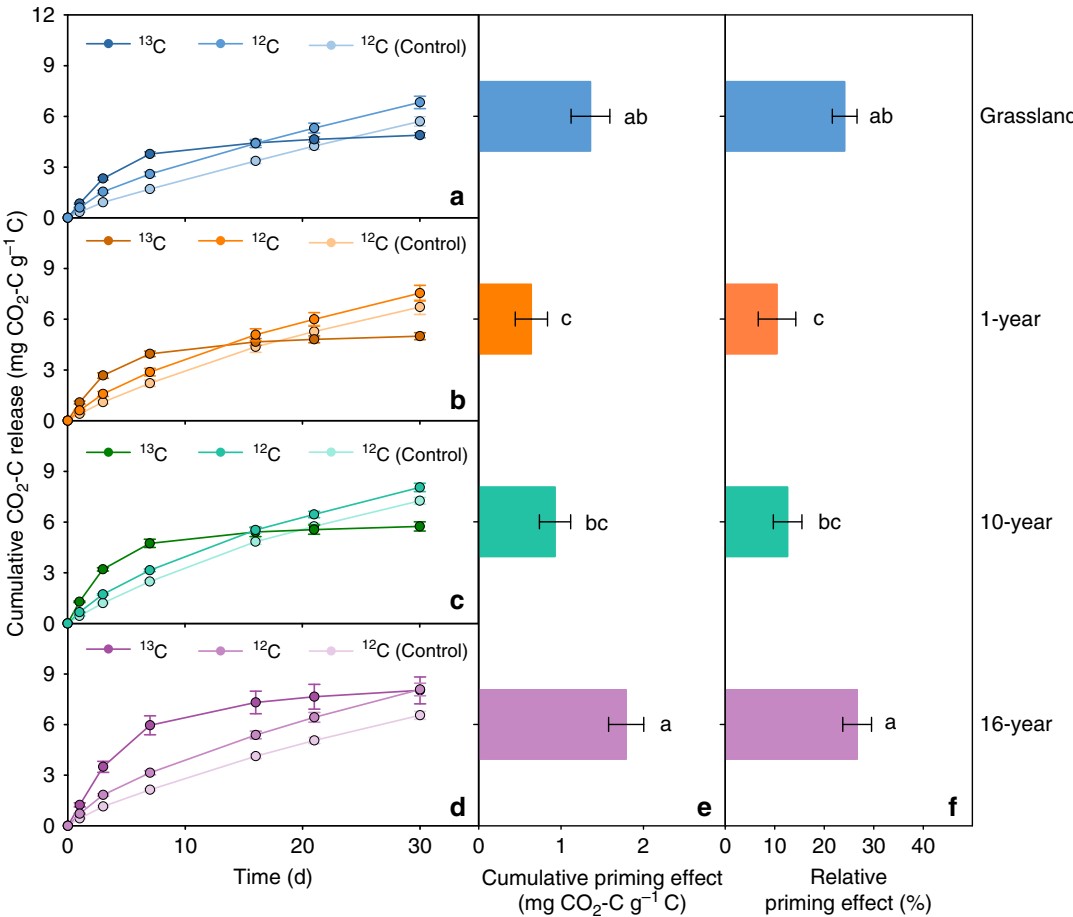

**Fig. 1** Cumulative $CO_2$ release and priming effect along the thaw sequence. Cumulative $CO_2$–C release from glucose ($^{13}C$) and existing SOM ($^{12}C$) at different thaw stages of permafrost collapse: **a** grassland, **b** 1-year since collapse, **c** 10-year since collapse and **d** 16-year since collapse. Comparison of the priming effect (**e**) and relative priming effect (**f**) among thaw stages. Error bars indicate standard error. Significant differences are denoted by different letters ($P < 0.05$)

the late-stage site ($P < 0.05$; Supplementary Fig. 4b), further aggravating microbial N limitation at this stage.

Although permafrost collapse altered many soil and microbial parameters, the priming effect along the thaw sequence was only negatively related to TDN concentration ($P < 0.01$; Fig. 2a) and was positively associated with C:N imbalance ($P < 0.01$; Fig. 2b). No significant correlation was observed between the priming effect and other factors, such as the soil DOC concentration, C:N ratio, $B_{C:N}$, pH, clay and silt content (Supplementary Fig. 5). These results indicated that soil N availability and C:N imbalance might induce the variations in primed $CO_2$–C release along the thaw sequence. The assumption was confirmed by the subsequent N addition experiment, in which significant interactions between glucose and N addition on both C:N imbalance ($P < 0.01$; Supplementary Fig. 6b) and SOM-derived C release ($P < 0.05$; Supplementary Fig. 6c) were found. These interactions indicated that the magnitude of the glucose effect on C:N imbalance and SOM-derived C release decreased after N addition. Specifically, N addition led to a significant increase in TDN concentration ($P < 0.01$; Fig. 3a, Supplementary Fig. 6a), and a consequent decline in C:N imbalance ($P = 0.02$; Fig. 3b), and priming effect when glucose was simultaneously added ($P < 0.01$; Fig. 3c). Taken together, these results confirmed the negative correlation between priming and post-thaw N availability observed along the thaw sequence.

**Roles of enzyme activity and microbial metabolic efficiency**. In the gradient experiment, glucose (G) × thaw sequence (time) interactions had no significant effects on enzyme activities and two parameters of microbial metabolic efficiency. Glucose addition did not affect the activities of β-1,4-glucosidase (BG) ($P = 0.95$), phenol oxidase (POX) ($P = 0.41$) or β-1,4-N-acetylglucosaminidase (NAG) ($P = 0.26$; Supplementary Fig. 7 a–c) but increased the activity of leucine aminopeptidase (LAP) ($P < 0.05$; Fig. 4a). Labile C supply also significantly decreased microbial C use efficiency (CUE) ($P < 0.01$; Fig. 4b) and enhanced the metabolic quotient ($qCO_2$) ($P < 0.01$; Fig. 4c). Meanwhile, microbial metabolic efficiency varied significantly among the thaw stages ($P < 0.05$; Fig. 4b–c). Specifically, CUE decreased while $qCO_2$ increased with thaw time, with the lowest microbial metabolic efficiency occurring at the late-stage site. Further regression analyses showed that despite the LAP activity being closely correlated with SOM-derived $CO_2$ release ($r^2 = 0.34$, $P < 0.01$; Supplementary Fig. 7d), no significant association was observed between the priming effect and LAP activity along the thaw sequence ($P = 0.61$; Fig. 4d). In contrast, the priming effect was significantly correlated with both CUE ($r^2 = 0.31$, $P < 0.05$; Fig. 4e) and $qCO_2$ ($r^2 = 0.43$, $P < 0.01$; Fig. 4f). These results demonstrated that lower microbial metabolic efficiency rather than higher enzyme activity was responsible for the greater priming effect under low-N conditions (i.e., the late stage of collapse).

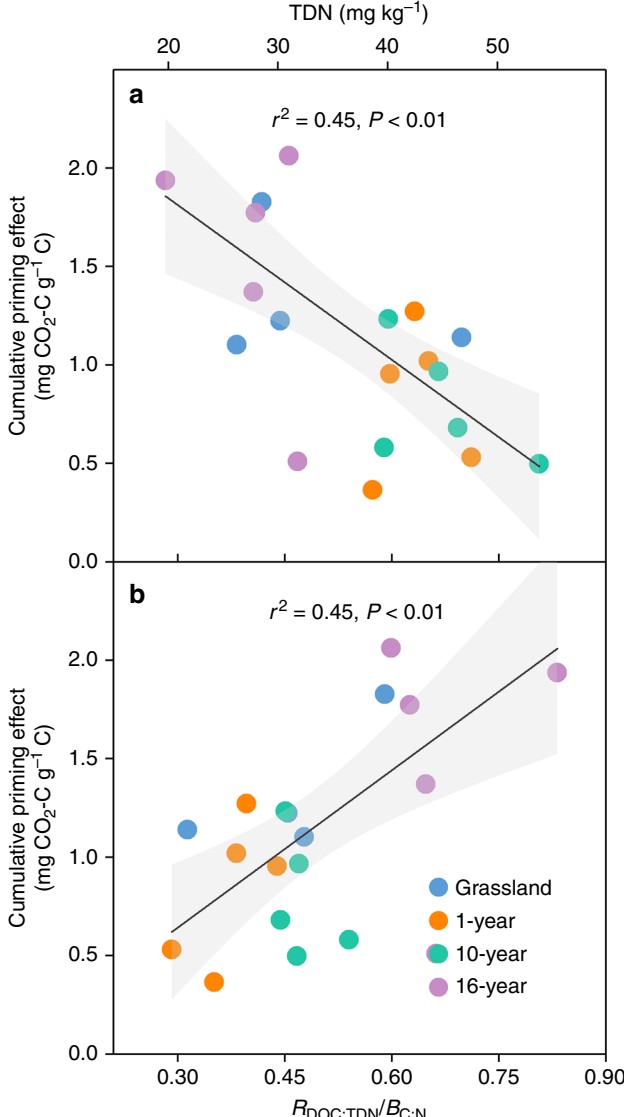

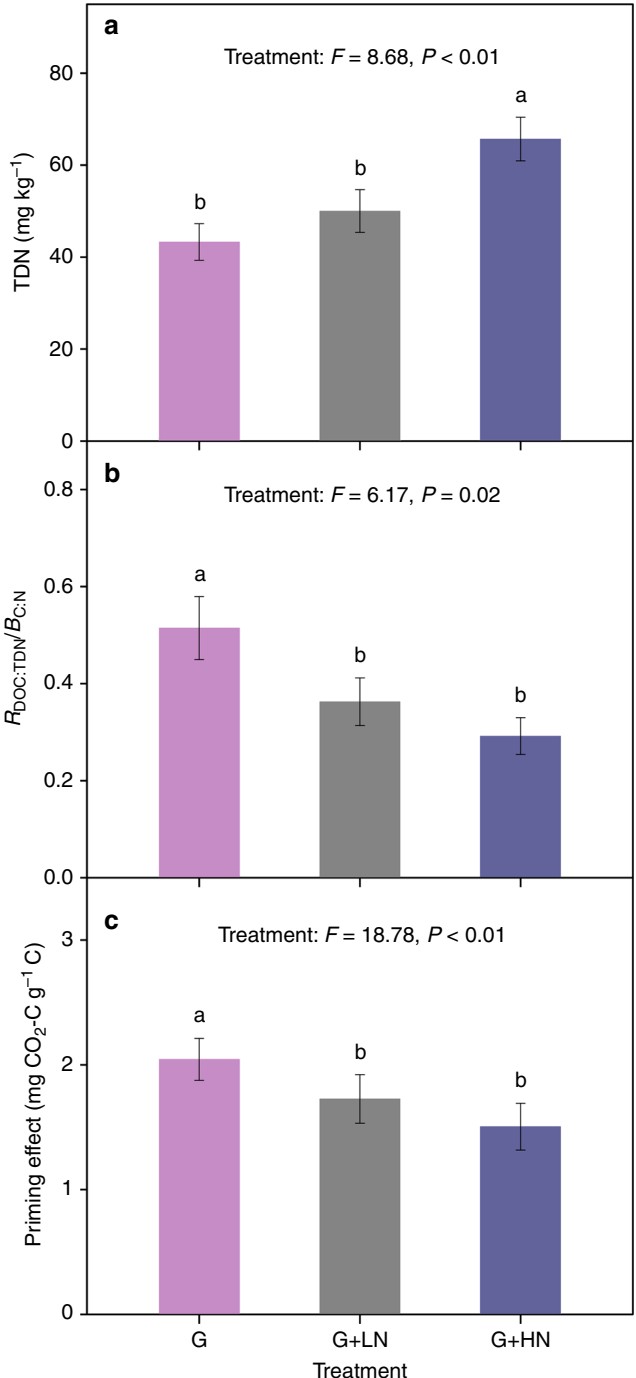

**Fig. 2** Relationships between cumulative priming effect and key soil N parameters along the thaw sequence. **a** Total dissolved nitrogen (TDN) concentration and **b** C:N imbalance ($R_{DOC:TDN}/B_{C:N}$) under the glucose treatment

**Fig. 3** Effects of N addition on key soil N parameters and priming effect. **a** Soil total dissolved N (TDN) concentration, **b** C:N imbalance ($R_{DOC:TDN}/B_{C:N}$) and **c** the priming effect from late-stage soils amended with different levels of N. Data are represented as the means ± SE (standard error). G: glucose addition, G + LN: glucose and low N addition, G + HN: glucose and high N addition. Significant differences are denoted by different letters ($P <$ 0.05)

Consistent with the gradient experiment, there was no significant glucose (G) × N addition interaction on CUE in the subsequent N addition experiment (Supplementary Fig. 8a). The supply of mineral N significantly increased CUE (Fig. 5a). A significant interaction between glucose and N addition affected $qCO_2$ ($P = 0.04$, Supplementary Fig. 8b, for a detailed discussion on the interaction effect, see Supplementary Note 1); specifically, N addition only decreased $qCO_2$ when glucose was simultaneously added ($P = 0.04$, Fig. 5b). Interestingly, the priming effect increased significantly with declining CUE ($r^2 = 0.55$, $P < 0.01$; Fig. 5c) and increasing $qCO_2$ ($r^2 = 0.66$, $P < 0.01$; Fig. 5d), confirming the observed association between the priming effect and microbial metabolic efficiency along the thaw sequence.

**Discussion**

Our study illustrated a relatively high priming susceptibility of organic soil in upland thermokarst regions on the Tibetan Plateau. In our laboratory incubation, the supply of glucose induced

positive priming effect on all collapse stages along the thermo-erosion gully, with the maximum relative priming intensity of 26.6% from the late-stage topsoil. The response in our organic soil was comparable to that of mineral soils (14–31%)[26,29], but much larger than that of organic soils in arctic ecosystems (−3 to 7%)[27–29]. Such a difference could be due to variations in microbial C

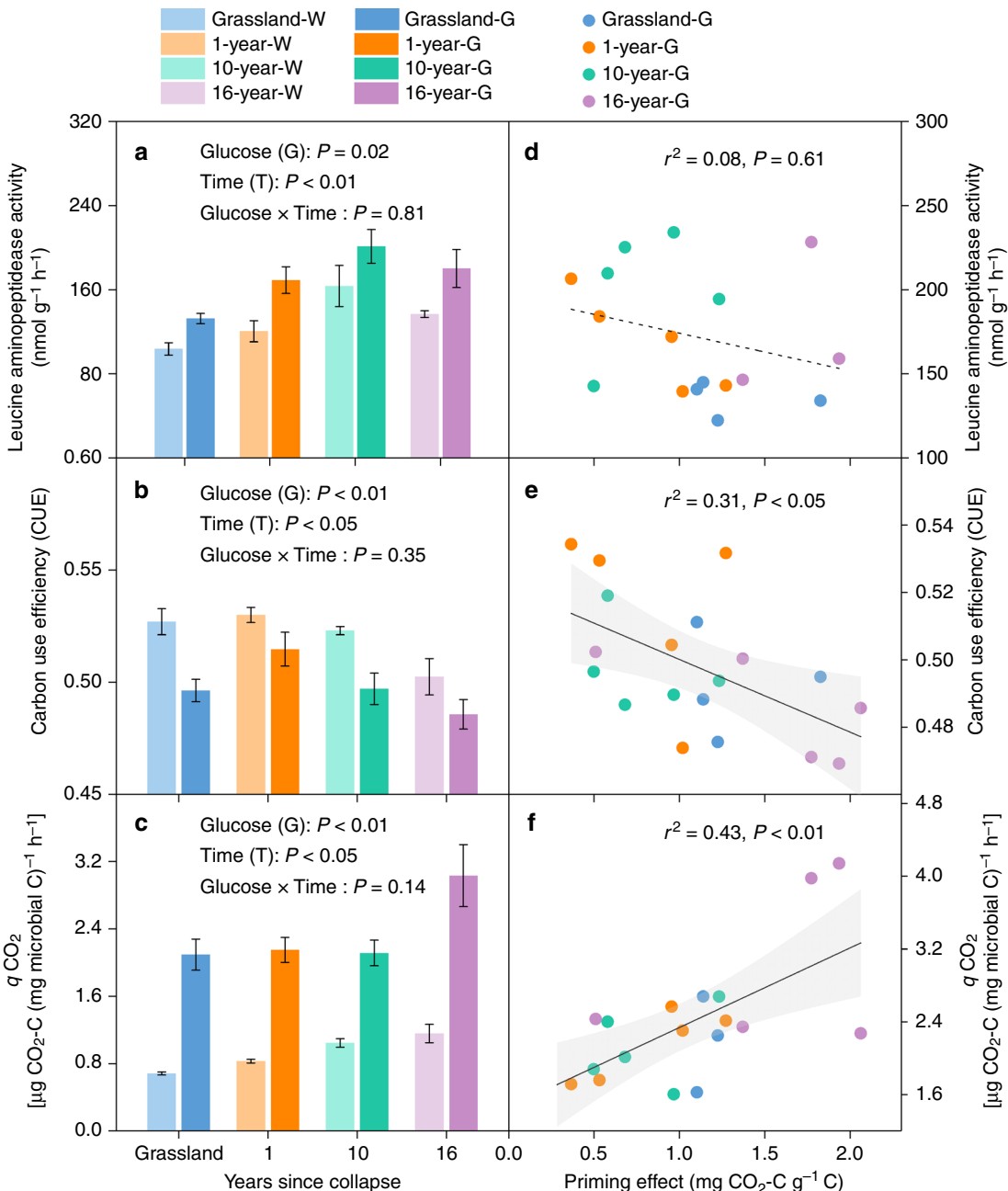

**Fig. 4** Extracellular enzyme activities and microbial metabolic efficiency along the thaw sequence. Changes in **a** leucine aminopeptidase (LAP) activity, **b** microbial C use efficiency (CUE) and **c** microbial metabolic quotients ($qCO_2$) after glucose addition along the thaw sequence. Relationship between the priming effect and **d** LAP, **e** CUE and **f** $qCO_2$. Data are represented as the means ± SE (standard error). Significant differences are denoted by different letters ($P < 0.05$)

and N limitation across soil horizons and ecosystems[26,29]. In arctic tundra ecosystem, the decrease in soil C concentration (21.4–48.6% vs. 3.0–20.1%) and C:N ratio (25.6–41.3 vs. 11.9–27.0) from organic soil to mineral soil[27–29] illustrated a shift from predominant microbial N limitation to C limitation with increasing soil depth[21,26,29]. In contrast, the soil C concentration and C:N ratio of organic soils in our study site were only 18.7% and 12.3, respectively (Supplementary Table 1). The lower C availability compared to the arctic tundra soils indicated that the microbial activity in Tibetan organic soils is still limited by C[40], thereby contributing to its high susceptibility to labile C input. Notably, the higher priming susceptibility in Tibetan organic soil

than in arctic organic soil highlights the variation in priming intensity among permafrost ecosystems. When this regional difference is not considered, the role of priming effect in regulating soil C dynamics at the global scale may be underestimated.

Our study also presented a negative linkage between N availability and priming effect. Such a pattern is consistent with the microbial N mining hypothesis, which assumes that microorganisms use labile C to decompose recalcitrant SOM and thus acquire N[41]. This hypothesis suggests that higher demands of microorganisms for N result in their greater response to the addition of labile C. However, the synthesis of extracellular enzymes in the N mining process (i.e., SOM decomposition) has a

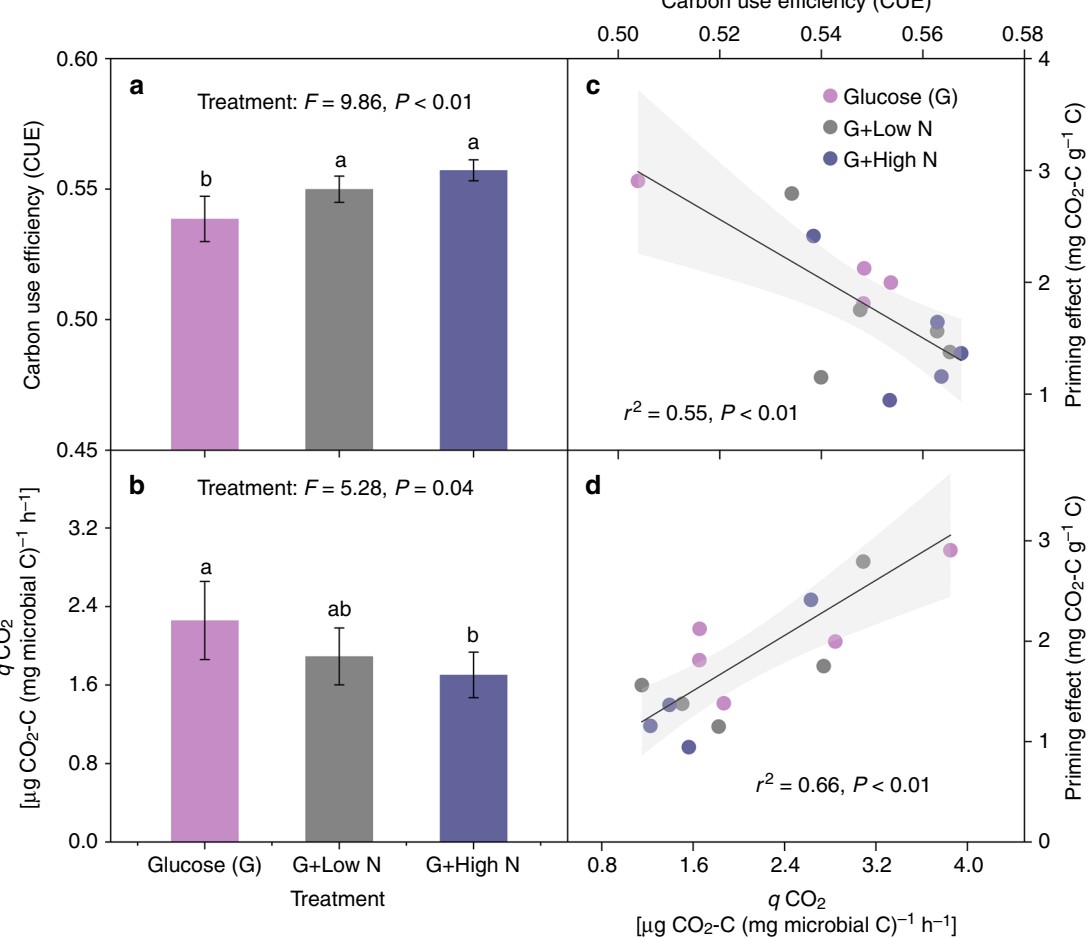

**Fig. 5** Microbial metabolic efficiency and its association with the priming effect in the N addition experiment. The effects of N addition on **a** microbial C use efficiency (CUE) and **b** metabolic quotients ($q$CO$_2$). Relationship between the priming effect and **c** CUE and **d** $q$CO$_2$. Data are represented as the means ± SE (standard error). Significant differences are denoted by different letters ($P < 0.05$)

high energy cost[42]. Increased N availability induced by N supply might thus switch the microbial preferred substrate from SOM to added labile N[18,43], thereby constraining C release from SOM. The negative association between the priming effect and N availability observed in this study differed from previous findings that N addition enhanced SOM decomposition in some arctic tundra[21,44]. These contrasting findings point to the context dependency of the N impact on the priming effect, which rely on variations in microbial N limitation with ecosystem, soil layer and season[26,45]. For those ecosystems where the predominant microbial N-limitation occurs in organic topsoils (characterized by a high soil C:N ratio, 45–64)[21,44], microbial N mining hypothesis does not hold true since nutrients could directly stimulate decomposer activity, thereby accelerating SOM decomposition. Nevertheless, even in these ecosystems, the extent of microbial N limitation exhibits large seasonal variations, resulting in the diverse responses of microbial activity to N addition[45].

Our results further demonstrated that lower microbial metabolic efficiency rather than higher enzyme activity accounted for the greater priming effect under low N availability (for a detailed discussion on enzyme activity, see Supplementary Note 2 and Supplementary Fig. 9). This finding supports our hypothesis that the increase in post-thaw N availability inhibits the priming effect by elevating the microbial metabolic efficiency. An interesting question arises: why does microbial metabolic efficiency increase

with elevated soil N availability? Such a pattern might be explained by the following two aspects. First, this change could occur because of the important mechanism by which microorganisms cope with the large variability in C:N imbalance[35,36]. Microorganisms, especially bacterial communities with higher N demands[46], can lower their CUE under high C:N imbalance conditions (i.e., N-deficient conditions) to maintain a balanced biomass C:N ratio[30,32,33]. Consistent with this deduction, microbial metabolic efficiency of the bacterial-dominated communities in our study site decreased with their C:N imbalance in both the gradient experiment (CUE: $r^2 = 0.55$, $P < 0.01$; $q$CO$_2$: $r^2 = 0.56$, $P < 0.01$, Supplementary Fig. 10 a–b) and the N addition experiment (CUE: $r^2 = 0.88$, $P < 0.01$; $q$CO$_2$: $r^2 = 0.80$, $P < 0.01$, Supplementary Fig. 10 c–d). In contrast, microbial CUE did not reveal any significant association with microbial enzyme stoichiometry (EEA$_{C:N}$, the ratio of enzyme activities directed towards acquiring C and N from the environment and is calculated as BG/(NAG + LAP)) (Supplementary Fig. 11). These results demonstrated that the large decline in microbial metabolic efficiency at the late stage of permafrost collapse could be attributed to the higher C:N imbalance and stronger microbial N limitation. Second, shifts in the microbial community structure to dominance by organisms that have lower efficiency may also result in concomitant changes in microbial metabolic efficiency[33,35]. Bacteria are commonly assumed to have lower

CUEs than fungi due to their lower biomass C:N ratios[35,46]. Thus, a greater fungal/bacterial (F/B) ratio can be expected to result in higher microbial metabolic efficiency. Consistent with this assumption, the increment in the F/B ratio was largest at the early stage of permafrost collapse but subsequently decreased with the collapse time (Supplementary Fig. 12a). More directly, the change in F/B ratio was positively correlated with CUE along the thaw sequence ($r^2 = 0.36$, $P < 0.01$; Supplementary Fig. 12b), demonstrating that the shift in microbial community structure was partly responsible for the variation in microbial metabolic efficiency along the thaw sequence.

While our study provides the experimental evidence for the role of N availability in regulating the priming effect after permafrost thaw, some uncertainties still exist. First, given that the priming effect is determined through laboratory incubation without plants, biotic attributes regulating the rhizosphere priming in situ have not been considered. For example, the quantity[8,17], quality[26,47] and timing of the C inputs[48] could affect the priming intensity, but it is challenging to simulate these biotic impacts with substrate addition under laboratory conditions. In situ rhizosphere priming experiments are thus needed to better elucidate this important plant–soil interaction in permafrost ecosystems. Second, this study concentrates on soil N availability and microbial metabolic efficiency, but other edaphic properties, such as water availability[49], SOM quality[47] and soil aggregation[17], may also contribute to the varied priming intensity. Extending studies to incorporate these alternative factors should further improve our understanding of potential determinants of the priming effect. Third, this study focuses on one typical upland thermokarst site. Although the observed decrease in soil C content is similar to that found in another thermokarst site on the Tibetan Plateau[50] and the observed magnitude of the priming effect (5.8–31.8%) is comparable to those obtained across the whole Tibetan Plateau (3.8–45.1%) (Chen et al., in preparation), site-level observations may induce uncertainties in predicting the response of soil $CO_2$ release to increased N availability after permafrost thaw. Future studies with a greater number of thermo-erosion gullies and grassland types are necessary to better explore the C–N interactions after permafrost thaw.

In summary, our study used a thermokarst-induced natural N gradient combined with a subsequent N addition experiment to demonstrate that post-thaw N availability drives the priming effect through adjusting the microbial metabolic efficiency. At the early stage of permafrost collapse, high soil N availability would relieve the microorganisms from N limitation and thus induce a smaller C:N imbalance after the supply of high-energy substrate. Meanwhile, the supply of labile C also resulted in a larger increase in the F/B ratio at this collapse stage. The smaller C:N imbalance together with changes in microbial community structure jointly led to higher microbial metabolic efficiency and the consequently smaller C loss after glucose supply (Fig. 6). However, at the late stage of permafrost collapse, soil N availability decreased due to its lateral transfer via water flow, further resulting in higher C:N imbalance after glucose supply. This higher microbial N limitation together with the smaller increase in the F/B ratio induced by labile C input at this stage contributed to lower microbial metabolic efficiency and the consequent higher SOM-derived $CO_2$ release (Fig. 6). These findings have two important implications for understanding permafrost C dynamics across Tibetan upland thermokarst regions. First, the negative association between post-thaw N availability and the priming effect suggests that the release of N from Tibetan upland thermokarst regions may decelerate soil C loss by reducing the SOM-derived $CO_2$ release. Combined with its positive effect on vegetation C sequestration[3], increased soil N availability after permafrost thaw is thus expected to regulate ecosystem C balance across Tibetan upland thermokarst

regions. Second, the significant linkage between the priming effect and microbial metabolic efficiency indicates that the shift in microbial metabolic efficiency is an essential pathway that regulates the priming effect under different N conditions. This finding highlights the fundamental role of microbial metabolic efficiency in disentangling the complex C–N interactions after permafrost thaw. More effort is thus needed to characterize the broad patterns of microbial metabolic efficiency and uncover its responses to permafrost thaw for better understanding the post-thaw C dynamics under global warming.

## Methods

**Site description.** The Tibetan Plateau is the largest alpine permafrost region in the world[51], storing approximately 15.3 Pg C in its top 3-m soil[52]. During the past several decades, climate warming induced widespread thermokarst across swamp meadows on the plateau[50,53]. In contrast to the ice-poor permafrost deposits in two other grassland types (alpine steppe and meadow)[52], the permafrost under swamp meadow is characterized as ice rich due to its poor drainage[53,54]. Moreover, soils in swamp meadow exhibit the highest C content among the soils of various grassland types on the plateau[52], with a SOC density comparable to that of the high-latitude permafrost regions (65.0 vs. 58.2 kg C m$^{-2}$)[52]. However, unlike arctic and subarctic permafrost regions, where the microbial activity in organic soil is solely N limited[21], soil microbial activity in swamp meadow is co-limited by C and N availability, as indicated by the low soil C concentration and C:N ratio[40,55] (C limitation) combined with negative net N mineralization rate[23,24] and the high ratio of N immobilization to total gross N mineralization[25] (N limitation). More importantly, in contrast with the shallow root distribution in arctic tundra ecosystems[10,29,56], approximately 81% of plant roots are distributed in the top 30 cm of the soil in the swamp meadow (Supplementary Fig. 13a), less than in arctic tundra (96%). The relatively deep root distribution may facilitate the potential regulation of the priming effect on deep soil C dynamics across Tibetan upland thermokarst regions.

The study site (~3848 m altitude, 37°28′ N, 100°17′ E) is located in the eastern tributary of the Altun-Qilian mountains ranges, which is one of the five typical permafrost zones on the Tibetan Plateau[57]. The mean annual temperature of the study site is −3.3 °C and the mean precipitation is 460 mm. The grassland type of the study site is swamp meadow, with dominant species of *Kobresia tibetica*, *K. royleana* and *Carex atrofusca*. The maximal rooting depth in this region is approximately 70 cm (Supplementary Fig. 13). The soil is a Gelisol and is periodically subjected to water logging and characterized by high SOC density and water content (Supplementary Table 1). The average thickness of the organic horizon (SOM content >20%) is approximately 70 cm, and under that lies the mineral soil including the fluvial of glacial till and fluviological deposits (Supplementary Fig. 14). The soil pH in the top 15 cm is between 6.0 and 6.8. In addition, permafrost occurs within 1 m from the soil surface (i.e., the average active layer depth is 0.86 m), and thermokarst landscapes develop due to the high ground ice content in the permafrost layer[53]. Notably, the C density and C:N ratio of the study site is higher than the average of the whole Tibetan Plateau (C density: 54.1 vs. 10.9 kg C m$^{-3}$; C:N ratio: 12.3 vs. 8.9)[40,52] but comparable to the average of swamp meadow (54.1 vs. 46.1 kg C m$^{-3}$; 12.3 vs. 11.9)[40,52]. Thus, our study site is well representative of swamp meadow, a typical grassland type where thermokarst mainly develops across the Tibetan permafrost region.

**Field sampling design.** In this study, we explored the variation of the priming effect along a thermo-erosion gully located within the study site described above. The thermo-erosion gully first developed in the 1990s and is approximately two decades old. The ages of the different sites along the gully were estimated as the distance between the site and the gully headwall divided by the estimated retreat rate of gully, which was determined by combining Google Earth aerial photographs (2007–2013) and repeated field measurements (2014–2016)[53].

Based on their ages and stages of permafrost collapse, four sites were established in 2014: undisturbed grassland site, early-stage site, middle-stage site and late-stage site. Of them, the undisturbed grassland site is the control site, without permafrost collapse. The early-stage site (1 year since collapse) is the most recent collapse site, with the deepest trench to the undisturbed soil surface (~2 m). Permafrost collapse tears apart the grassland into numerous vegetated rafts, resulting in substantial soil exposure. Substantial water drainage occurs for the vegetated patches at this stage, possibly altering the microbial activities and thereby inducing changes in soil C dynamics[14]. The middle-stage site was set at a location that had experienced 10 years of permafrost collapse, where the vertical distance between the original soil surface and collapsed soil was approximately 80 cm. Both above- and belowground plant biomass increased at this stage (Supplementary Fig. 15) and the cover of *K. tibetica* increased at the expense of that of *C. atrofusca*. These changes in plant production and community composition could induce shifts in the priming effect[8]. The late-stage site was located at a downstream position of the gully with 16 years of permafrost collapse. The surface inside the gully was almost level with undisturbed area. Plant recolonizations had occurred in the original exposed area at this stage, indicating the collapse has entered the oldest stage of recovery. Despite

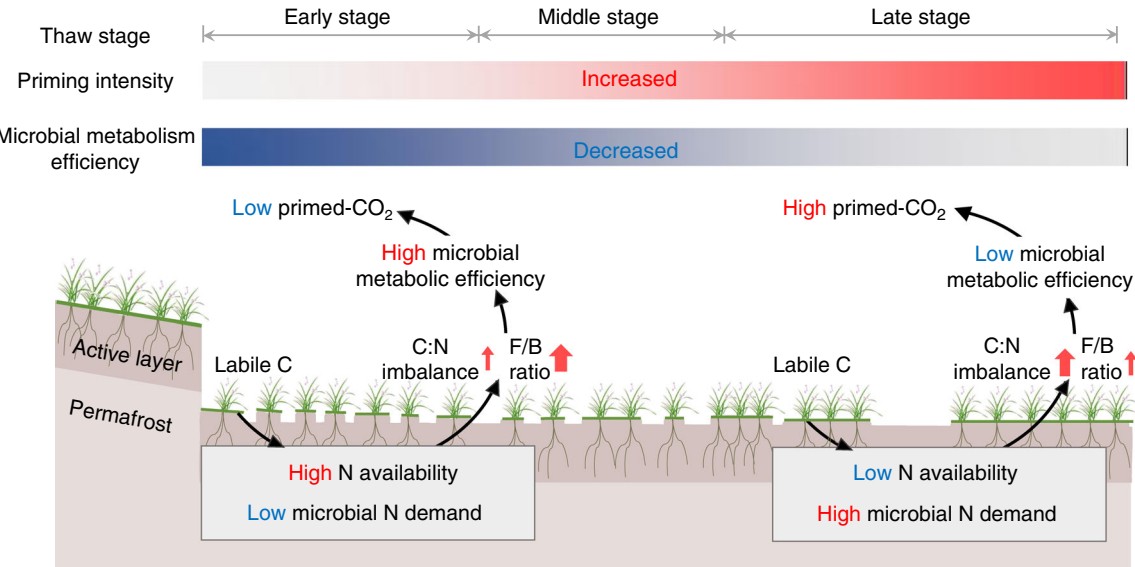

**Fig. 6** Changes in priming effects after permafrost collapse and their links to N availability and microbial metabolic efficiency. At the early stage of permafrost collapse, microbial N limitation is relieved by high N availability as a consequence of enhanced microbial N mineralization, further resulting in a lower C:N imbalance. The lower N limitation at this stage, together with the smaller increase in the C:N imbalance (lower N demand) and the larger increase in the fungal/bacterial (F/B) ratio induced by labile C input, jointly contribute to the higher microbial metabolic efficiency. This higher metabolic efficiency would therefore result in less primed microbial respiration at the early stage of collapse. By contrast, at the late stage of permafrost collapse, N availability decreases significantly owing to its lateral transfer with the water flow, resulting in the higher C:N imbalance. This higher N limitation at this stage, together with the larger increase in C:N imbalance and the smaller increase in the F/B ratio induced by labile C input, results in the lower microbial metabolic efficiency. This lower microbial metabolism ultimately leads to higher priming effect at the late stage of permafrost collapse

a shift in plant community composition with thaw time, root distributions did not reveal significant changes along the thaw sequence (Supplementary Fig. 13 b–e).

In August 2015, five independent plots (~3 m × 5 m) were set up within a 10 m × 20 m area at each collapse stage as replicates for soil sampling (Supplementary Fig. 1). Within each plot, the top 15 cm of soil was collected from all the vegetated patches using a 3-cm diameter hand corer to reduce spatial heterogeneity. On average, 10–12 soil cores were collected inside each plot and then mixed in the field as one replicate for subsequent analyses. Soil samples were only collected at the vegetated patch but not the exposed area because the exposed soil was a mixture of surface and deep soil resulting from the redistribution of surface materials inside the gully. Correspondingly, soil samples from five independent plots (~3 m × 5 m) in the undisturbed grassland (within a 10 m × 20 m area) were collected as controls. Samples were immediately transported to the laboratory (State Key Laboratory of Vegetation and Environmental Change, Institute of Botany, Chinese Academy of Sciences), where they were sieved to 2 mm and then stored at −20 °C until the $^{13}$C-labelled glucose and N addition experiment.

**Isotope-labelled glucose and N addition experiment**. We designed two experiments to explore the role of post-thaw N availability in regulating the priming effect. The first experiment (termed gradient experiment) was conducted along the thaw sequence and included two types of treatments: collapse times (grassland without collapse or with 1, 10 and 16 years of collapse) and substrate addition (deionized water and $^{13}$C-labelled glucose solution (3 at% $^{13}$C)). After detecting the correlation between the priming effect and post-thaw N availability in the first experiment, we performed the second experiment to further rule out the potential impact of other concomitant factors besides N. The second experiment (termed N addition experiment) was conducted with soils only from the late-stage site and included a $^{13}$C-labelled glucose treatment with two levels of N addition in the form of NH$_4$NO$_3$. This approach resulted in the following treatments: water (control), glucose (G), low N, G + low N, high N and G + high N. The N addition treatments were designed to simulate the high N availability at the early stage of collapse. Given that the natural difference in TDN concentration between early- and late-stage sites ranged from 20.1 to 50.8 mg N kg$^{-1}$ soil (Supplementary Data 1), the rate of NH$_4$NO$_3$ addition was set to 20 and 50 mg N kg$^{-1}$ soil in our low- and high-N treatment, respectively. In both experiments, the $^{13}$C-glucose solution (3 at% $^{13}$C) was obtained by mixing $^{13}$C-labelled glucose (Sigma-Aldrich, uniformly labelled, 99 at% $^{13}$C) with unlabelled glucose before application. The $^{13}$C-glucose solution was evenly added at a rate of 2.2 mg $^{13}$C g$^{-1}$ dry soil, which approximately equalled 1% of SOC, similar to previous priming studies (range: 0.5–2%)[26,27,47].

Before the experimental treatment, fresh sieved soil (approximately 20 g dried basis) was homogeneously mixed with the abovementioned substrates in 400-ml amber bottles with airtight lids. The incubation temperature was set to 15 °C,

representing the upper end of the soil temperature during the growing season at the study site (0.1–18 °C). Soil moisture was adjusted to 70% of water holding capacity (WHC) and maintained by deionized water addition. Duplicate bottles were established to enable destructive sampling at days 3 and 30 to determine microbial biomass C (MBC), microbial biomass N (MBN), DOC, TDN, microbial community composition and enzyme activity. Nine more empty bottles were incubated simultaneously as blanks. CO$_2$ sampling was conducted for all the bottles on six occasions (days 1, 3, 7, 14, 21 and 30).

**CO$_2$ flux analyses and priming effect calculations**. Before measurement, each bottle was flushed with CO$_2$-free air for 20 min to make identical initial gas conditions. Thereafter, all bottles were placed in an incubator for a short period (ranging from 1 h to 8 h for different sampling times), and 50 ml of headspace gas was then sampled. This gas sample was evenly divided into two halves to measure the CO$_2$ flux and $^{13}$C abundance. The CO$_2$ flux was measured with an infrared gas analyser (EGM-5; PP Systems, Haverhill, MA, USA), and the $^{13}$C abundance was determined with isotopic ratio mass spectrometry (IRMS 20-22, SerCon, Crewe, UK).

The fraction ($f_{SOM}$) and amount of CO$_2$–C derived from the SOM pool (SOM-C$_R$) (mg CO$_2$–C g$^{-1}$ C d$^{-1}$) were determined following Eqs. (1) and (2):

$$f_{SOM} = \left( \text{at\%}_{total} - \text{at\%}_{glucose} \right) / \left( \text{at\%}_{SOM} - \text{at\%}_{glucose} \right), \quad (1)$$

$$SOM - C_R = C_{total} \times f_{SOM}, \quad (2)$$

where at%$_{total}$, at%$_{glucose}$ and at%$_{SOM}$ are C isotope composition (in at% $^{13}$C) of total CO$_2$ respiration, added glucose and SOM, respectively. $C_{total}$ is the total CO$_2$–C release (mg CO$_2$–C g$^{-1}$ C d$^{-1}$).

The priming effect (primed SOM-derived CO$_2$, mg CO$_2$–C g$^{-1}$ C d$^{-1}$) was determined following Eq. 3:

$$Priming\ effect = SOM - C_R - C_{control}, \quad (3)$$

where SOM-C$_R$ is the amount of CO$_2$–C derived from the SOM pool (mg CO$_2$–C g$^{-1}$ C d$^{-1}$) and $C_{control}$ is the CO$_2$–C released from control soil (water addition) (mg CO$_2$–C g$^{-1}$ C d$^{-1}$). The cumulative priming effect and cumulative CO$_2$ efflux from controls for the 30-day incubation were estimated by integrating the absolute priming effect and CO$_2$–C release from control soil over time, respectively.

**Soil and microbial properties analysis**. The SOC and STN concentration were measured on soil samples after air drying. The SOC concentration was determined

by potassium dichromate oxidation method. STN concentration was measured using an elemental analyser (Vario EL III, Elementar, Germany). To determine the available microbial resources, we further measured the soil DOC and TDN. To be specific, 5 g of fresh soil was mixed with 20 ml of 0.5 M $K_2SO_4$ in a 100 ml bottle and shaken for 30 min[22]. The suspension was subsequently filtered and used for determination on an Elementar TOC analyser (Liqui TOCC II, Germany). Gravimetric soil moisture was determined by drying a 20-g subsample of fresh soil at 105 °C for 24 h. Soil samples collected with cutting rings were dried in the same manner to determine bulk density. WHC was measured gravimetrically using a sample that had been air-dried for 72 h and then rewetted until no more water could be absorbed[58]. Soil pH was measured in 1:2.5 soil:water suspension and analyzed using a pH electrode (PB-10, Sartorius, Germany).

MBC and MBN were analyzed using the chloroform fumigation method[59]. Using universal conversion factors of 0.45 for MBC[60] and 0.54 for MBN[61], we calculated the amounts of MBC and MBN at day 0, 3 and 30. The C:N imbalance between resources and microorganism (both the total form ($R_{C:N}/B_{C:N}$) and labile form ($R_{DOC:TDN}/B_{C:N}$)) were calculated as the ratios of the resource C:N ratios ($R_{C:N}$ and $R_{DOC:TDN}$) normalized to $B_{C:N}$[32]. Higher C:N imbalances correspond to lower N availability relative to C availability and could thereafter be used as a proxy of microbial N limitation[22,32]. The ratio of soil DOC to TDN was presumed a better representative of the available microbial resource stoichiometry than the bulk soil C:N ratio because the dissolved C and N are more easily available for microbial communities[22].

To explore the role of microorganisms in regulating the priming effect, we further analyzed three types of microbial properties, including the enzyme activities, microbial metabolic efficiency (both CUE and $q$CO$_2$)[33,34,62] and microbial community composition. Enzyme activities were assayed in soils along the thaw sequence immediately after a 3-day incubation. The enzymes activities determined in this study included those of the C-acquiring enzymes BG (EC 3.2.1.21), which is involved in labile C decomposition, and POX (EC 1.10.3.2), which is involved in the degradation of recalcitrant C, and those of the N-acquiring enzymes NAG (EC 3.1.6.1), involved in the degradation of chitin, and LAP (EC 3.4.11.1), involved in the hydrolysis of organic N compounds and proteins. The activities of BG, NAG and LAP were assayed following the method described by German et al.[63] using fluorometric techniques, which included the construction of calibration curves for each sample[64]. The standards (4-methylumbelliferone (MUB) for BG and NAG and 7-amino-4-methylcoumarin (AMC) for LAP) at concentrations of 0, 2.5, 5, 10, 25, 50 and 100 μM were added to soil slurries to account for quenching. Assay wells in a black microplate included 200 μl of soil slurry and 50 μl of fluorometric substrate solution (200 μM, saturating concentration, 4-MUB-β-D-glucoside for BG, 4-MUB-N-acetyl-β-D-glucosaminide for NAG and L-leucine-7-amido-4-methylcoumarin for LAP) and were incubated for 6 h at 25 °C. Sixteen replicates for each soil sample were set up in each plate. The amount of fluorescence was determined using a fluorometer (Beckman Coulter DTX 880, Indianapolis, IN, USA) with 365-nm excitation and 450-nm emission.

The activity of POX was assayed using L-3,4-dihydroxy-phenylalanine (L-DOPA) as substrate[64]. Then, 50 μl of L-DOPA (25 mM) was combined in each sample well with 150 μl of soil slurry and 50 μl of EDTA. The blank wells received 100 μl of acetate buffer and 150 μl of soil slurry. The negative controls were created using 150 μl of buffer, 50 μl of EDTA and 50 μl of DOPA. EDTA was added to eliminate the potential effects of reducing metal ions (e.g., $Fe^{2+}$)[65]. Sixteen replicates were set up for each soil sample, blank and control. Activity was quantified using a multimode detector (Beckman Coulter DTX 880, Indianapolis, IN, USA) to measure the absorbance at 450 nm.

Based on the above measurements, we calculated two common parameters of microbial metabolic efficiency (i.e., CUE and $q$CO$_2$). The microbial CUE was calculated based on C:N stoichiometry[33] following Eqs. (4) and (5):

$$CUE_{C:N} = CUE_{max}[S_{C:N}/(S_{C:N} + K_N)], \qquad (4)$$

$$S_{C:N} = (1/EEA_{C:N})(B_{C:N}/L_{C:N}), \qquad (5)$$

where $S_{C:N}$ is a scalar that represents the extent to which the allocation of enzyme activities offsets the disparity between the DOC:TDN ratios of resources and microbial biomass[33]. The half-saturation constant $K_N$ was set to 0.5. $CUE_{max}$ is the upper limit for microbial growth efficiency and was set to 0.6 based on thermodynamic constraints[33]. $EEA_{C:N}$ is the ratio of enzyme activities directed towards acquiring C and N from the environment and is calculated as BG/(NAG + LAP). $L_{C:N}$ is the labile organic matter C:N ratio.

Additionally, we calculated the $q$CO$_2$ as the total CO$_2$–C release rate per unit of microbial C[66] following Eq. (6):

$$qCO_2 = R_{total}/C_{mic}, \qquad (6)$$

where $R_{total}$ and $C_{mic}$ are the total C release rate (mg CO$_2$–C g$^{-1}$ soil d$^{-1}$) and MBC (mg kg$^{-1}$) at day 3, respectively. As done in previous studies[62,67], $q$CO$_2$ was used as a proxy of CUE, although it does not equal CUE[66]. Higher $q$CO$_2$ values were assumed to correspond to lower C allocation to biomass but greater respiration losses, and thus an increase in $q$CO$_2$ indicates a decreased microbial metabolic efficiency[62,67].

Furthermore, we used phospholipid fatty acid (PLFA) analysis to examine the soil microbial community composition. PLFAs were extracted from the soils following a previously described method[68] and further identified by an Agilent 6890 gas chromatograph (Agilent Technologies, Palo Alto, CA, USA) and the MIDI Sherlock Microbial Identification System (MIDI Inc., Newark, DE, USA). FAME 19:0 was used as standard to quantify the fatty acids. The abundance of individual fatty acids was determined in units of nmol g$^{-1}$ of dry soil. PLFAs specific to bacteria (i14:0, i15:0, a15:0, i16:0, a17:0, i17:0, br-17:0, 16:1ω7c, cy17:0, 18:1ω7c, 18:1ω5c, cy18:0 and cy19:0) and fungi (18:2ω6c, 18:1ω9c) were quantified.

**Data analyses**. We used log transformation to normalize the data when necessary and then analyzed data by the following three steps. First, one-way ANOVAs with Least Significant Difference (LSD) multiple comparisons were used to explore the differences in soil and microbial parameters (soil moisture, SOC, TN, soil C:N ratio ($R_{C:N}$), DOC, TDN, clay and silt content, $B_{C:N}$, C:N imbalance ($R_{C:N}/B_{C:N}$ and $R_{DOC:TDN}/B_{C:N}$)) as well as the priming effect along the thaw sequence. During these analyses, thaw time was treated as between-subject effect. Linear regression analysis was then conducted to explore relationships between the priming effect and the abiotic and biotic variables along the thaw sequence.

Second, two-way mixed-effect models were used to assess whether and how SOM-derived CO$_2$–C release, enzyme activity, CUE, $q$CO$_2$ and microbial community composition were affected by glucose addition and thaw time in the gradient experiment, with treatment (glucose vs. control) and collapse time (grassland and 1, 10 and 16 years of collapse) as fixed effects and replicate as a random effect. In the N addition experiment, two-way mixed-effect models were used to assess the interaction between glucose and N addition. Moreover, one-way mixed-effect models were performed to directly compare the effect of N addition; in this case, N addition was set as a fixed effect and replicate was set as a random effect. LSD test was further used for multiple comparisons.

Finally, linear regression analyses were conducted to explore the relationships between enzyme activity, CUE, $q$CO$_2$ and the priming effect to reveal the underlying role of microorganisms in regulating the priming effect. All statistical analyses were performed using R statistical software v3.2.4 (R Development Core Team, 2016).

## Data availability

The data that support the findings of this study and those not presented within the article and its Supplementary Information file are available from Y.Y. upon reasonable request.

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

## Acknowledgements

We are grateful to Dr. Biao Zhu at Peking University and Prof. Xingliang Xu at the Institute of Geographic Sciences and Natural Resources Research, CAS, for providing helpful comments on an early version of this manuscript. We also thank Prof. Bernhard Schmid at the University of Zurich and Prof. Even Siemann at Rice University for their assistance in statistical analyses. This work was supported by the National Key R&D Program of China (2017YFC0503903), the National Natural Science Foundation of

China (31770557 and 31670482), the Key Research Program of Frontier Sciences, CAS (QYZDB-SSW-SMC049), the Youth Innovation Promotion Association of Chinese Academy of Science, and the Chinese Academy of Sciences-Peking University Pioneer Cooperation Team.

## Author contributions

Y.Y. and L.C. conceived the idea. L.C., Y.Y. and L.L. designed the research. L.L., L.C., C. M., S.Q., J.W., Q.Z., G.Y., D.Z. and J.Y. performed the experiments. F.L., L.C., C.M., L.L., J.W. and J.Y. performed the field sample collection. L.C. and L.L. analyzed the data. L.C., Y.Y., L.L. and S.B. wrote the manuscript.

## Additional information

**Competing interests:** The authors declare no competing interests.

