## [Peer Review File · Nature Communications]

Reviewers' comments:

Reviewer #1 (Remarks to the Author):

The manuscript 'Nitrogen availability regulates soil carbon dynamics by altering microbial carbon metabolism after permafrost thaw' reports the results on an incubation study using isotopically labeled glucose to measure the priming effect (an increase in soil organic matter respiration in response to the availability of labile carbon versus a control soil, which in the field can be induced by plant root exudates). The incubated soils are sampled from the organic part of the active layer at an alpine permafrost thaw gradient on the Tibetan Plateau. The authors claim that the permafrost thaw gradient is also a nitrogen availability gradient, and that differences observed in the priming effect in the incubated soils can be explained by nitrogen availability. The incubation study is not coupled to field measurements of carbon respiration and/or plant measurements such as changes in rooting depth or community composition.

Very few studies currently report on the occurrence of the priming effect in permafrost (-affected) soils. Indeed, as the authors point out correctly, this is surprising because permafrost soils store a large part of the terrestrial carbon, which when it thaws becomes physically available to both microbes and plants. At the moment, estimates of future permafrost carbon losses are largely based on incubation studies that account for temperature effects (Schuur 2015, Schaedel 2014), but do not take into account plant-soil interactions that could affect SOM respiration (the priming effect). Since the priming effect can increase basal respiration by up to 380 % (Huo, 2017), better knowledge about the factors influencing its occurrence in permafrost (-affected) soils might allow for more accurate predictions of future C-losses from permafrost soils. I am aware of only two studies addressing this issue thus far, in arctic permafrost soils (Wild 2014, 2016). Hence, the data presented in the manuscript provided by Cheng et al. could be a welcome addition.

However, I do not agree with the author's claim that results of this work should be included in Earth system models or that it will help us to better understand the permafrost carbon feedback, for two reasons:

- 1) The potential C-losses due to permafrost thawing of the Tibetan Plateau permafrost are relatively low (C-loss by 2100 under RCP8.5): with roughly 0.28-0.37 Pg C from Tibetan Plateau permafrost (Chen 2016), vs. 100 Pg C from northern circumpolar permafrost soils (Schuur 2015);
- 2) As the authors point out multiple times, the permafrost soils from the Tibetan Plateau differ in many ways from the northern circumpolar permafrost soils. Hence, mechanistic understanding gained from studies on the Tibetan Plateau permafrost is not directly applicable to other types of permafrost.

I find the manuscript currently not well-written. I urge the authors to revise the entire manuscript and pay specific attention to structure, sentence construction and the logic behind their reasoning. Further, please make sure all presented results are discussed in the text, also the interaction effects. Below some specific comments, but my comments do not address everything.

(Selected) comments by line:

-comments related to use of literature-

L. 69 Please provide a reference for 'N availability is a key regulator of the PE in permafrost ecosystems'

L.72 I am not sure why Joergensen 1996 is referred to here? (reference 12)

L. 75 'Due to this point, several N addition experiments have been conducted...' Please explain

how three studies performed in 2012, 2016 and 2016 (references 13, 14, 15) could have led to conducting experiments in 1996, 2004 and 2010 (reference 12, 4 and 9) and adapt your reasoning accordingly.

L. 75 Also, reconsider the validity of references 12, 4 and 9 for your argument.

L.120 'microbial activity in Tibetan alpine permafrost is more subjected to energy limitation owing to low soil organic C concentration' referring to Chen et al. (2016), Nature communications. This reference could potentially be very useful to the current paper since it is written by the same authors and deals somewhat with the same system. However, I did not read any reference to whether or not microbes are energy limited in Tibetan alpine permafrost soils in Chen et al. 2016?

Moreover, the 2016 article states high lability of the C in Tibetan alpine permafrost, and a large range of SOC concentrations between sites ranging from 2.2 – 79.9 g kg⁻¹ with alpine meadows not listed as having low SOC content. This seems to contradict the current claim of energy limitation?

Additionally, in Chen et al. 2016, soil cores were taken from larger depth (up to 3 m.) which most likely consisted mostly of mineral soil (see Fig. 6, this article). I am therefore not sure that this information is exactly useful for the current article, where only the upper organic layer was sampled (L.166-168)? Since the authors wrote the referred-to article, I would like to ask them to explain better what they mean exactly, in L. 120.

L. 386 Please specify on which part of the results presented in reference 40 you base this conclusion?

-other comments-

L. 30 'priming effect', here and in the rest of the text, change to: 'the priming effect'

L.47 'melting' should be 'thawing' (ice melts, soil and meat thaw);

L.47 depends (grammar); please check grammar throughout the manuscript

L.54 'the strongest' compared to what?

L. 58 'accompanied by', change to: 'due to'

L. 64-65 'the fact that most C is stored in soil' – this is not very specific, rephrase entire sentence (62-65) and include something about the frozen state of the soil and whether or not plants can access it.

L.69-70 Why would the general statement that 'C and N cycles are closely coupled in most ecosystems' mean that N-availability is a key regulator of the PE in permafrost ecosystems'?? Even if you add 'permafrost' to 'ecosystems' in line 70, this is still not a good argument. Please improve.

Additionally, this reasoning which is based on arctic and subarctic permafrost ecosystems, is used to build up to the hypothesis that N-availability will also regulate the PE in the Tibetan Plateau permafrost. However, further on in the manuscript it is mentioned that, whereas in arctic and subarctic ecosystems plant and microbial growth is N-limited, this is not the case in the Tibetan

Plateau permafrost, where apparently C-limitation is prevailing (L. 120). Hence, not only this sentence, but also the entire build up to the hypotheses is not solid. Please improve.

L. 75 'Due to this point, several N addition experiments have been conducted...' Please re-write this section, it is crucial to your hypotheses but currently full of nonsense, I am sorry to say.

L. 96 'ultimately resulting in reduced decomposition and negative priming effect' – this goes too fast for me, please elaborate

L. 98 'induce shift', change to: 'induce a shift', and please check your use of articles in the entire manuscript

L. 108 – 111 As I understand the priming effect (PE), it is an increase in SOM-derived CO₂ in response to a labile C input, without further specification about whether or not the microbial biomass carbon (C) increases. An increase in qCO₂ is an increase in SOM-derived CO₂ relative to microbial biomass C. Please explain in your manuscript, how exactly do you discern between the PE and an increase in qCO₂?

L.120 'microbial activity in Tibetan alpine permafrost is more subjected to energy limitation owing to low soil organic C concentration', seems to not be in line with referred to study by the same authors. Crucial to current rationale, please revise with scrutiny;

L.120 Additional to the above, I would like to ask the authors to provide more information on the soil quality. A table with soil parameters including at minimum soil C and N and DOC would be very interesting.

L. 133 (hypothesis i) magnitude? direction?

L.134 (hypothesis ii) The 'how' in this hypothesis suggest that you were already certain before performing your research, that N-availability regulates the priming effect in alpine permafrost ecosystems. If so, please provide a reference.

L.135. Please specify how you discern between an increase in SOM-respiration as a result of increased enzyme activity and an increase in SOM respiration per microbial biomass C. This is not clear to me.

L.145 'the soil is a silty loam with pH of' – please give uncertainty estimates and more information on soil parameters, this is crucial information for the reader to be able to understand your findings;

L.148-149 'within the past several decades' – it is UNCLEAR HOW THE AGE OF YOUR TRANSECT STAGES WAS DETERMINED, please elaborate or give an appropriate reference

L. 167 'exposed mineral soils were not included in the soil sampling' – seems to contradict with the 'silty loam' earlier?

L. 170 Did you perform bulk density measurements in the field in order to be able to express your results in meaningful units?

L. 178 "to further demonstrate that" see my comment about 'positivistic urge'

L. 184 What was the final C:N ratio of the other treatments?

L. 186 how much dry weight?

L. 188 How was water holding capacity determined?

L. 205 and further: when working with labeled substrates it is generally preferred to work with abundances (not $\delta^{13}\text{C}$), please adapt this.

L.232 and L.243-244 There is debate on whether this is indeed always a 'microbial available resource' (Mooshammer et al. 2014), please give a reference here, and why not also refer here to your nice supplementary figure where you express data in the more common way, C:N instead of DOC: TDN?

L 245-246 I am not sure what you mean here?

L.289 which protocol?

L 295 unfinished sentence

L. 299 Why not also DOC?

L. 300 Unclear how you 'measured' microbial N-limitation, did you determine a Threshold elemental ratio?

L. 315 Where your data normally distributed?

L. 350 Fig. 3c I interpret the not significant interaction between glucose and nitrogen as indicative of NO effect of nitrogen availability on the PE, please discuss this

L. 366 decrease relative to what?

L. 367 Fig. 5a, b Please discuss your interaction effects (Fig. 5) here and in the discussion.

L. 367 Fig 5a If your 'time' is the N-gradient, than what does it mean that there is no interaction effect in Fig. 5a (no effect of time/N-gradient on the increase in $q\text{CO}_2$ due to the PE?) whereas there IS an interaction effect in Fig. 5b? Please discuss this (apparent?) contradiction in the text as well.

L. 368 Fig. 5c: I see a saturating function, which linear increase are you referring to?

L. 368 Fig 5c: please explain the reader in your methods or results section which information this graph offers, since it is not an easy to interpret graph. If I understand it correctly, what you plot is in fact the increase in $q\text{CO}_2$ DUE TO the PE against the PE itself. Would this mean that we would expect a linear 1:1 relationship if all increase in $q\text{CO}_2$ was due to the PE, and all deviation would be changes in $q\text{CO}_2$ caused by other factors than the PE? I might be wrong, please explain and discuss it in the text

L. 379-380 Based on the findings presented in Chen et al. 2016, which show that the potential C-losses due to permafrost thawing of the Tibetan Plateau permafrost are relatively low (C-loss by 2100 under RCP8.5): with roughly 0.28-0.37 Pg C from Tibetan Plateau permafrost (Chen 2016), vs. 100 Pg C ($\pm 3\%$ s.e.) from northern circumpolar permafrost soils (Schuur 2015), I find this statement ('important implications ...feedback to climate warming') not applicable to this study.

L. 383-384 Please rephrase this sentence into a more modest statement, reflecting your actual results. The priming effect in this study was not actually measured in the field, but in an incubation experiment. You can not conclude from this that 'the supply of labile C induced a widespread positive priming across the thaw gradient'.

L. 386 Please compare your priming effect results with the referred studies 16 and 29, using actual numbers instead of using the phrasing 'barely affected'.

L. 391 Please express your results per volume, not per area, using bulk density data;

L.393 What do you mean with 'stronger response to organic C input'? Which experiment do you refer to?

L. 401 Fig. 1e does not back up this statement;

L. 403 Fig. 1f does not show this

L. 485 -486 'the efficiency of C incorporated into microbial biomass versus released as CO₂' this sounds as carbon use efficiency (CUE), you claim this is the 'reciprocal of qCO₂'. However, "while both the CUE and the qCO₂ are related to C use, they cannot directly be converted into each other" (Spohn 2014). Please take this difference between the CUE and the qCO₂ into account in your Discussion.

Fig. 1: The + G bars are in my opinion superfluous (we can understand this from adding the cumulative priming effect and the control CO₂ release). I am not sure which additional information the analysis of the total CO₂ release offers, if presented here at least it should be discussed in the text. Further, it would be interesting to know if there are interactive effects between time and G-treatment/PE.

Fig. 2: To better understand what the result presented here indicates, I would like to have to extra panels: Microbial C:N and DOC.

Could the imbalance also mean a surplus of available C?

Fig. 3 The colour scheme used here to indicate 'treatment' is the same as used in the previous graph to indicate 'years since collapse'. Please make this less confusing.

Fig. 5 I think your unit for the qCO₂ is not correct: why do you include soil here? Please explain in your methods the difference between your approach and, for example, Spohn 2014.

Fig. 6 I much appreciate this schematic representation, which I believe improves the understanding of the context of the results presented in this manuscript. However, I also have many questions and comments related to this Figure:

6a. Please mention in the legend what F/B (fungal bacteria ratio) stands for, it is mentioned in the main text but not in the legend;

6b. Why 'bacterial' biomass and not 'microbial' biomass, this is not discussed in the text?

6c. This Figure suggest that there is a rather thick organic layer in the grassland stage, which contradicts the reported very thin organic layer in Tibetan permafrost areas due to a dry climate (Gao et al., 1985; Wu & Zhang, 2010)? Please adapt or include explicitly in your Discussion what this means for the general relevance of your data on the priming effect in organic soil of Tibetan alpine permafrost;

6d. this Figure also suggest that the permafrost is contains ice-wedges in the intact 'grassland'-stage, the melting of which can indeed lead to severe karst. However, as I understood from Chen

et al. 2016 (Supplementary Table 2), the Tibetan plateau permafrost is particularly dry and does not have a high ice-content. Please address this in the text and/or make consistent with this Figure.

Also, to prevent confusion when studying this Figure 6, please make the site description more informative and consistent with previous studies on Tibetan Plateau permafrost (or indicate why and how this specific site is different from other sites on the Tibetan plateau and how this might affect the interpretation of the results).

6e. THE PLANTS HERE ARE ROOT-LESS. For this particular article that is inappropriate, since the priming effect is driven by labile C-inputs from plant-roots. It is thus relevant to see which layers of the soil these roots have access to, and whether this changes after permafrost thawing. Please adapt the figure accordingly.

6f. Do plants not grow on the mineral soil? If so, the priming effect would only occur in the thin organic layer, which is not frozen even in the grassland stage.

6g. The Figure is slightly misleading as it suggests that the results explain processes in the permafrost itself, or at least in the thawed mineral soil layer. In fact, only the organic soil was sampled. This Figure needs to be adapted to reflect that.

6h. Are there no carbon dynamics in the mineral soil? Only the organic layer was sampled, yet from this Figure it becomes clear that in the later stages the organic layer is not the main soil pool. How much of the total Tibetan Plateau permafrost soil is stored in these organic patches? Perhaps change the title to soil carbon dynamics in the organic layer, in order to not overstate findings?

One last general comment: Without stating here that all research should strictly follow the principles of the critical rationalism (I know this is unrealistic), I would like to ask the authors to acquaint themselves with the idea behind 'falsification' and the importance of formulating testable hypotheses. While the work presented here is truly interesting, the authors seem to want to 'proof that' nitrogen availability regulates soil carbon dynamics. In my view, wanting to 'proof' can reduce scientific open-mindedness (and the validity of drawn conclusions).

For example, in the literature, multiple mechanistic explanations have been suggested for the priming effect. These include, but are not limited to, 'microbial N mining'. Other explanations are 'energy limitation', 'co-metabolism' and 'microbial community shifts' (Kuzyakov 2000, Fontaine 2007, Bengtson 2000). In field-situations, the rhizosphere priming effect (induced by living plant roots, as opposed to the PE which is induced by a substrate added under laboratory conditions, Huo 2017) can be influenced by even more factors, such as water availability, pH values, redox potentials and soil aggregation (Bengtson 2000, Huo, 2017). None of these alternative underlying mechanisms are addressed by the authors. Neither is the difference between what can be measured in an incubation study versus what actually happens in the field (priming effect versus rhizosphere priming effect). Moreover, the way the data are currently presented does not allow the reader to judge whether another explanation than the one presented by the authors (N availability) could also be considered. I would like to see this changed, throughout the manuscript (Introduction, Results, Discussion).

Lastly, although I am aware that I did not review this version of the manuscript very favourably, I do wish to thank you for all the hard work that is obviously behind these data. I find the data conceptually very interesting, and I look forward to reading this manuscript in its revised form.

Reviewer #2 (Remarks to the Author):

Nature comm. Review

Here the authors present an interesting study linking carbon dynamics and N availability in permafrost, and explore how these interactions may change over time since thaw. Further, they address a very important microbial component, demonstrating that microbial metabolism can be important in predictive models. The study is relevant for a number of reasons including (1) the obvious effects of climate change on carbon feedback cycles, (2) the increased awareness of including microbes into global models, (3) significant remaining questions regarding N availability (especially N mining) with soil respiration and finally (4) acknowledging the large differences in permafrost systems (alpine vs arctic). However, while there are many compelling points within this study there are a number of significant points that need to be addressed.

1. The sampling design is problematic: (L158 Field design) It can be argued that this is pseudo replication within single replicate plots of each time stage, instead of 5 replicates at each stage. Ideally, each time stage should have 3-4 separate thermos-erosion gullies. There needs to a strong defense of this experimental design –it is a critical point – especially if the results are to be used for broad/global statements about carbon dynamics, microbial response and the use of enzyme analyses.

2. The relationship between priming and N availability (and therefore N mining) is interesting and could be made a larger part of the manuscript. As of how it is really only mentioned in the discussion, but instead could be used in the introduction to really set up the discrepancy studies have observed and be presented as a hypothesis. On that point, figure 6 is interesting and a case could be made to having it moved up in the text (even the introduction) as the hypothesis for how this system works.

3. Too much weight is being put on the enzyme results, because they are not in line with other results. This would be more convincing if the dataset was broader. Further, these data are correlative, and a better explanation/discussion needs to be made on why these results are different from all other results. It cannot merely be that this site functions different from all others, and if it does why?

Abstract:

L27: 'aggravate' is a weird word choice to describe soil C.

L33: as presented, this study does little to shed light on microbial mechanisms, rather creates more questions in the end. Perhaps highlight a more specific gap this study addresses.

Introduction: Much of this section could be rewritten to clarify better the challenges in understanding C-N dynamics and how a better understanding of microbial activity is the answer. Many studies have already examined this, what new research does this study specifically add?

Background references should show breadth of study, and how there is quite a bit of variation and difficulty in predicting permafrost/C response to warming (Potential carbon emissions dominated by carbon dioxide from thawed permafrost soils, Nature Climate Change 6, 950–953, (2016), doi:10.1038/nclimate3054. AND "Predicted responses of arctic and alpine ecosystems to altered seasonality under climate change."

L54: 'may' predicted?

L56: clarify direct and indirect

L74: relieve

L78-90: The point needs to be made stronger how this study will address the gap. Why can't N addition experiments or N gradients be informative on their own? Clarify the challenge/propose the

mechanism for why combining them in one study will be the solution?

L113: But does it hold a lot of the world's carbon? Why is it an important ecosystem

L114: clarify

Methods:

L140: typical for the area?

L147/149: remove 'typical'

L147-156: The site description needs to be clarified. From this description it seems it was not ideal for the study, but I don't think that was intended.

L161: how were these stages designated? Is 16 year the oldest possible? Is 1 year the youngest necessary to see effects? Explain more directly.

L165: how many cores per replicate were taken?

L251: add the reference

Results

L323: add 'stage'

Discussion:

L373-380: This does not clarify why THIS study shows a strong relationship between N and priming – clearly state what was observed and why this means there is a connection.

L382: For this section – need to elucidate why differences between the ecosystems are important and what this means for our understanding at a global scale.

Fig. 3 Colors should be different so it does not appear to be related to time stage

Reviewer #3 (Remarks to the Author):

General

The authors describe a study focussing on priming in Tibetan alpine permafrost, and the importance of N availability in this context. They combine two laboratory incubation experiments to reach this aim, (1) inducing priming by glucose addition along a natural, thermokarst-induced N availability gradient, and (2) specifically testing the interaction of glucose and N availability by adding both in a factorial design to one of the soils along the gradient. The data suggest that Tibetan permafrost soils are susceptible to priming, that priming is strongest where N availability is low, and that this effect is likely not driven by changes in enzyme activities as often assumed, but by changes in microbial C use efficiency. I find the approach appropriate and the data conclusive. However, I am wondering to what extent the observed patterns can be generalized. First, it is not clear whether the C input simulated in the experiments is realistic, both quantitatively (amount of C input) and qualitatively (chemical composition of C input, timing, etc.). The authors did not test the impact of actual plant C input, but simulate it in the laboratory, and the discussion needs to be more careful to account for this limitation (e.g., lines 391-398). Second, only one site is studied. There are of course many limitations when performing complex experiments at remote field sites, and this study is a valuable contribution. Still, it is impossible to tell if the same patterns would be observed at other, similar sites. Third, there are many studies on the connection between C and N cycling in the context of priming. These studies give very different results, as the authors also outline in the introduction (lines 75-78). These findings suggest that the effect of N on priming depends on parameters we currently do not understand. I therefore ask for caution when generalizing the findings of this study in the discussion, and in particular when suggesting to incorporate them into Earth System Models.

The authors further hint at systematic differences in C content, C vs. N limitation of microorganisms, and susceptibility to priming, between arctic and Tibetan alpine permafrost surface soils (e.g., lines 114-117). This is an interesting point, but no data or references are provided that support it. In fact, not even C contents from the soils used in this study are

presented. I suggest adding these data and discussing how representative they are for Tibetan permafrost soils. I particularly recommend to pay attention to differences between mineral and peat soils, and between soil horizons.

I would also like to read more discussion that links the findings of this laboratory study to natural ecosystems. In arctic permafrost soils, plant rooting and consequently plant C input are quite shallow (Iversen et al., 2015, *New Phytologist*), and the shallow soil shows low susceptibility to priming (Wild et al., 2016, *Scientific Reports*). Do the data presented here imply a different pattern in Tibetan alpine permafrost? The authors seem to hint at that. I would like to read some discussion on the matter that considers typical plant rooting depths and indicators of C versus N limitation in the shallow soil in Tibetan permafrost, as well as changes in vegetation and rooting with thermokarst formation. I would also like to know to what extent the increase in N availability with thermokarst formation could be linked to decreased plant N uptake due to physical disturbance. I suppose that there is not a lot of literature on these topics, but they should be at least (carefully) addressed.

Finally, although the language is understandable, there are many errors that should be fixed. I did not point them out in detail, but I recommend some serious language polishing.

Abstract

Line 46: Change to "thawing permafrost".

Introduction

Line 52-53: Both cited references are estimates of future losses. So "might induce" is more adequate. Also, change to "a strong C-climate feedback".

Line 55: Please elaborate in more detail by what mechanisms N availability could increase in permafrost systems with warming.

Line 57: "Increased plant-derived C input" is not correct here. I think it should be "increased plant C stocks" or "increase plant C fixation" or similar.

Line 68: I agree that N availability is likely to be important for priming effects in permafrost ecosystems, but I do not think that this statement can be supported by previous studies. So I would either remove "in permafrost ecosystems" or weaken the statement.

Lines 69-71: A range of studies has shown that plants can take up not only mineral N forms, but also some small organic N forms such as amino acids (e.g., Lærkedal Sorensen et al., 2008, *Arctic, Antarctic, and Alpine Research*; Nordin et al., 2004, *Ecology*; Schimel and Chapin, 1996, *Ecology*). Nitrogen mineralization is thus not a requirement for N availability. However, there is consensus on wide-spread N limitation of plants in permafrost systems. I suggest re-writing this argument. Also, ref. 12 seems wrong here.

Line 72: See above – I think somewhere in the introduction the mechanisms behind the expected increase in N availability should be explained. See also Koven et al., 2015, *PNAS*.

Line 76: Ref. 12 does not fit (I suppose some mix up when constructing the literature list; this extends through the whole manuscript). I also think the sentence would be clearer with some more text (e.g., "... with either positive or negative effects of increased N availability on the magnitude of priming.").

Line 80: Ref. 16 does not test the effect of N on priming.

Line 85: Ref. 11 does not fit here.

Line 87: Ref. 17 is not about priming, I do not think it is a very good fit here.

Line 116: Ref. 12 is wrong (see above). See also Sistla et al., 2012, *Soil Biology & Biochemistry*, but also Melle et al., 2015, *Soil Biology & Biochemistry*.

Lines 126-127: I suggest changing to "an index of microbial C use efficiency".

Material & Methods

Please provide some more information on the sampling site: Elevation, soil type, active layer depth, dominant vegetation.

I also suggest adding a (supplementary) table with basic parameters measured in the sampled soils, in particular also showing how they changed along the thaw gradient. This table should at least include organic C content and C/N ratios.

Line 143: "Perennially wet" as in water logged? Were soils anoxic? How did water content change along the thaw gradient? Please provide data.

Lines 149-151: Please elaborate on dominant vegetation and changes along the thaw gradient. The priming effect occurring under natural conditions might differ between vegetation types (e.g., linked to rooting patterns or mycorrhizal association), so details on the vegetation are valuable for setting the findings of this study in context.

Line 173: Where the soils homogenized after adding the label?

Line 184: What is the soil temperature in the field?

Line 210: I do not understand. According to line 172, glucose was enriched to 99 atom% ^{13}C – this would equal much more than 1752‰ (but should in fact not be expressed in delta values).

Line 236: Do you mean TOC (total organic carbon) or DOC (dissolved organic carbon)? To match with TDN, I suppose the latter. There are some discrepancies between text and figures throughout the manuscript, please fix that.

Line 243: Delete "thermokarst-induced".

Line 245: Change to "an index of microbial C use efficiency".

Lines 247-251: Please be more specific when describing the enzyme target compounds.

Lines 251: Please add a brief description of the measurement principle (addition of fluorescence-labelled substrates). What was used as standard, and were standards added to soil slurries to account for quenching? I am not sure what you mean with "the usual single-point correction"; the protocols I am familiar with include a multi-point calibration curve. Please also add substrate concentrations.

Line 260: Change to "L-dihydroxy-phenylalanine". What was the concentration of DOPA? Why was EDTA added?

Line 275: How did you measure the ^{13}C content of fumigated and unfumigated extracts?

Results

Lines 347-356: I find it difficult to distinguish between the two parts of the study (addition of glucose to thaw chronosequence and addition of glucose, N or both to the 16-year samples). It is additionally confusing that glucose had no significant effect on BG activity in the thaw chronosequence part, but in the 16-year part.

Discussion

Lines 377-379: See my general comments. Please present more information on the soils studied here to permit comparisons with organic or mineral horizons of arctic permafrost soils. With the data provided, I cannot tell if this comparison is justified.

Line 384: I suggest comparing C contents per dry soil, not square meter. Carbon content per square meter also depends on bulk density which is not relevant here.

Line 422-426: I think this discrepancy might be due to variability within arctic soils, e.g., changes in N limitation with soil depth or season (e.g., Melle et al., 2015, Soil Biology & Biochemistry).

Figures

Figure 4: There is a typo (nitorgen) in panel (d).

Response to Reviewer #1:

[Comment 1] The manuscript 'Nitrogen availability regulates soil carbon dynamics by altering microbial carbon metabolism after permafrost thaw' reports the results on an incubation study using isotopically labeled glucose to measure the priming effect (an increase in soil organic matter respiration in response to the availability of labile carbon versus a control soil, which in the field can be induced by plant root exudates). The incubated soils are sampled from the organic part of the active layer at a alpine permafrost thaw gradient on the Tibetan Plateau. The authors claim that the permafrost thaw gradient is also a nitrogen availability gradient, and that differences observed in the priming effect in the incubated soils can be explained by nitrogen availability. The incubation study is not coupled to field measurements of carbon respiration and/or plant measurements such as changes in rooting depth or community composition.

[Response] We are very grateful to the reviewer for the insightful comments on our manuscript! Following the reviewer's comments, we have added more descriptions about the changes in vegetation community composition and plant rooting depth along the thaw sequence in the *Methods* session as follows: "The grassland type of the study site is swamp meadow, with dominant species of *Kobresia tibetica*, *K. royleana* and *Carex atrofusca*. Both above- and belowground plant biomass increased at the mid-stage of permafrost collapse (Fig. S14) and the cover of *K. tibetica* increased at the expense of that of *C. atrofusca*. Despite a shift in plant community composition with thaw time, root distributions did not reveal significant changes along the thaw sequence. Approximately 78% of plant roots in the study site are distributed in the top 30 cm of soil (Fig. R1b), with a maximal rooting depth of approximately 70 cm" (Page 17-19, line 347-350; line 385-386; line 391-393). With regard to soil respiration, we cannot provide this kind of data at this moment since we haven't conducted field measurements yet. Thanks for your understanding!

Figure R1. Vertical distributions of roots in the whole swamp meadow across the Tibetan Plateau (a) and the specific swamp meadow along the thaw sequence involved in this study: undisturbed grassland site (b), early-stage (c), middle-stage (d), late-stage (e) of the permafrost collapse. The vertical distribution of roots was fitted by the function proposed by Gale and Grigal (1987). This was characterized as $Y = 1 - \beta^d$, where Y is cumulative percentage of root biomass from the soil surface to depth d (cm), and β is the fitted parameter. The data for root distributions in the whole swamp meadow across the plateau were obtained from the regional survey in 2005 (Yang *et al.*, 2009).

[Comment 2] Very few studies currently report on the occurrence of the priming effect in permafrost (-affected) soils. Indeed, as the authors point out correctly, this is surprising because permafrost soils store a large part of the terrestrial carbon, which when it thaws becomes physically available to both microbes and plants. At the moment, estimates of future permafrost carbon losses are largely based on incubation studies that account for temperature effects (Schuur 2015, Schaedel 2014), but do not take into account plant-soil interactions that could affect SOM respiration (the priming effect). Since the priming effect can increase basal respiration by up to 380 % (Huo, 2017), better knowledge about the factors influencing its occurrence in permafrost (-affected) soils in might allow for more accurate predictions of future C-losses from permafrost soils. I am aware of only two studies addressing this issue

thus far, in arctic permafrost soils (Wild 2014, 2016). Hence, the data presented in the manuscript provided by Chen et al. could be a welcome addition.

[Response] Thanks for the reviewer's positive comment.

[Comment 3] However, I do not agree with the author's claim that results of this work should be included in Earth system models or that it will help us to better understand the permafrost carbon feedback, for two reasons: 1) The potential C-losses due to permafrost thawing of the Tibetan Plateau permafrost are relatively low (C-loss by 2100 under RCP8.5): with roughly 0.28-0.37 Pg C from Tibetan Plateau permafrost (Chen 2016), vs. 100 Pg C from northern circumpolar permafrost soils (Schuur 2015); 2) As the authors point out multiple times, the permafrost soils from the Tibetan Plateau differ in many ways from the northern circumpolar permafrost soils. Hence, mechanistic understanding gained from studies on the Tibetan Plateau permafrost is not directly applicable to other types of permafrost.

[Response] We agree that the mechanistic understanding obtained from this study may not be directly applicable to other permafrost types. Thus, following the reviewer's comments, we have deleted the argument about the "results of this work should be included in Earth system models" and "These findings have important implications in understanding permafrost C dynamics and its feedback to climate warming". Moreover, we have reorganized the whole manuscript to focus on the priming effect in the context of Tibetan upland thermokarst.

[Comment 4] I find the manuscript currently not well-written. I urge the authors to revise the entire manuscript and pay specific attention to structure, sentence construction and the logic behind their reasoning. Further, please make sure all presented results are discussed in the text, also the interaction effects. Below some specific comments, but my comments do not address everything.

[Response] Following the reviewer's comments, we have nearly rewritten the whole manuscript. During this process, we have carefully reorganized the structure and checked the logical relationships between sentences, and also carefully checked the

references throughout the revised MS. We have also added more descriptions about all the presented results, especially the interaction effects in both *Results* and *Discussions* sections of the revised MS (Page 6, line 127-130; Page 8, line 163-167, 175-177; Page 9, line 193-198).

[Comment 5] L. 69 Please provide a reference for 'N availability is a key regulator of the PE in permafrost ecosystems'

[Response] Combining the suggestions from this reviewer and the other two reviewers, the paragraph has been reorganized and this sentence has been deleted in the revised MS (Page 3-4, line 69-83).

[Comment 6] L.72 I am not sure why Joergensen 1996 is referred to here? (reference 12)

[Response] We apologize for the mix up of references when constructing the literature list using the Endnote. In the revised MS, we have updated the reference as follows: “Among these changes, the widespread increase in nitrogen (N) availability after permafrost collapse (Harms *et al.*, 2013, Finger *et al.*, 2016), driven by enhanced N mineralization (Finger *et al.*, 2016) and the additional N released from thawing permafrost (Keuper *et al.*, 2012), may play an important role in regulating the priming intensity, since both vegetation growth and topsoil microbial activity in arctic (Sistla *et al.*, 2012, Wild *et al.*, 2015) and alpine ecosystems (Song *et al.*, 2007, Gao *et al.*, 2016, Kou *et al.*, 2017) are N limited”(Page 3-4, line 72-77).

[Comment 7] L. 75 'Due to this point, several N addition experiments have been conducted...' Please explain how three studies performed in 2012, 2016 and 2016 (references 13, 14, 15) could have led to conducting experiments in 1996, 2004 and 2010 (reference 12, 4 and 9) and adapt your reasoning accordingly.

[Response] Sorry for the poor logic among these sentences in the original MS. Combining the suggestions from this reviewer and the other two reviewers, we have rewritten the paragraph as follows: “Among these changes, the widespread increase in

nitrogen (N) availability after permafrost collapse (Harms *et al.*, 2013, Finger *et al.*, 2016), driven by enhanced N mineralization (Finger *et al.*, 2016) and the additional N released from thawing permafrost (Keuper *et al.*, 2012), may play an important role in regulating the priming intensity, since both vegetation growth and topsoil microbial activity in arctic (Sistla *et al.*, 2012, Wild *et al.*, 2015) and alpine ecosystems (Song *et al.*, 2007, Gao *et al.*, 2016, Kou *et al.*, 2017) are N limited. Despite this recognition, the link between N availability and the priming effect in permafrost ecosystems remains ambiguous because previous studies have revealed a positive (Wild *et al.*, 2014), a negative (Hartley *et al.*, 2010, De Baets *et al.*, 2016), and no effect (Wild *et al.*, 2014, De Baets *et al.*, 2016) of N addition on priming intensity” (Page 3-4, line 72-79).

[Comment 8] L. 75 Also, reconsider the validity of references 12, 4 and 9 for your argument.

[Response] We have reorganized the paragraph as follows: “Despite this recognition, the link between N availability and the priming effect in permafrost ecosystems remains ambiguous because previous studies have revealed a positive (Wild *et al.*, 2014), a negative (Hartley *et al.*, 2010, De Baets *et al.*, 2016), and no effect (Wild *et al.*, 2014, De Baets *et al.*, 2016) of N addition on priming intensity” (Page 4, line 77-79).

[Comment 9] L.120 'microbial activity in Tibetan alpine permafrost is more subjected to energy limitation owing to low soil organic C concentration' referring to Chen et al. (2016), Nature communications. This reference could potentially be very useful to the current paper since it is written by the same authors and deals somewhat with the same system. However, I did not read any reference to whether or not microbes are energy limited in Tibetan alpine permafrost soils in Chen et al. 2016? Moreover, the 2016 article states high lability of the C in Tibetan alpine permafrost, and a large range of SOC concentrations between sites ranging from 2.2 – 79.9 g kg⁻¹ with alpine meadows not listed as having low SOC content. This seems to contradict the current

claim of energy limitation? Additionally, in Chen *et al.* 2016, soil cores were taken from larger depth (up to 3 m.) which most likely consisted mostly of mineral soil (see Fig. 6, this article). I am therefore not sure that this information is exactly useful for the current article, where only the upper organic layer was sampled (L.166-168)? Since the authors wrote the referred-to article, I would like to ask them to explain better what they mean exactly, in l. 120.

[Response] Very good comments! Regarding the first part of the reviewer's comment, we agree that Chen *et al.* (2016a) *Nature Communications* is not suitable to be cited here given that microbial C limitation on the Tibetan Plateau was not directly mentioned in that paper. To avoid this confusion, we have deleted this reference in the revised MS, reorganized the paragraph, and added more references to support the argument that topsoil microbial activity in Tibetan alpine ecosystem is assumed to be co-limited by C (Chen *et al.*, 2016b, Tian *et al.*, 2017) and N availability (Song *et al.*, 2007, Gao *et al.*, 2016, Kou *et al.*, 2017).

Regarding the second part of the reviewer's comment, we would like to mention that the high C lability in Tibetan permafrost compared to the high-latitudes mentioned by Chen *et al.* (2016a) was only referred to the subsoils in active layer (20-30 cm) and permafrost layer. However, in this study, the comparison of the microbial C and nutrient limitation between Tibetan and arctic permafrost ecosystems focused on the organic topsoil. Moreover, this high C lability is mainly attributed to **the high proportion of labile SOM fraction rather than the high C availability (i.e., high amounts of labile SOM fraction)**. It should also be noted that compared to the pasture or grassland soils in non-permafrost region (Fontaine *et al.*, 2011, Zhu *et al.*, 2014), the microbial activity in Tibetan swamp meadow is less C limited due to the relatively higher C content. However, **compared to the arctic tundra organic soils, the microbial activity in Tibetan thermokarst region is still limited by C, indicated by relatively lower C content** (18.7% in our study site vs. 21.4~48.6 % in arctic tundra) **and soil C:N ratio** (12.3 in Tibetan swamp meadow vs. 25.6~41.3 in arctic tundra) (Hartley *et al.*, 2010, De Baets *et al.*, 2016, Wild *et al.*, 2016).

Similarly, the negative net N mineralization rate in alpine meadow and wetland (Song *et al.*, 2007, Gao *et al.*, 2016), higher proportion of microbial N immobilization to total gross N mineralization in Tibetan alpine grasslands (1.6 ± 0.2) (Kou *et al.*, 2017) and the increase in soil CO₂ flux after low N supply (Deng *et al.*, 2015b) jointly suggested that **N availability could also constrain the function of microbial communities in this region.** Hence, **microbial activity in Tibetan alpine permafrost is assumed to be co-limited by C and N availability.** Additionally, in all these references, soils were also collected from the top 10 or 20 cm, which is similar to the soil depth (0~15 cm) involved in this study. We have clearly mentioned these points in the revised MS (Page 16, line 334-338). Notably, to improve the readability of the *Introduction* section, we have moved the descriptions about Tibetan permafrost from the *Introduction* session to the *Methods* session in the revised MS.

[Comment 10] L. 386 Please specify on which part of the results presented in reference 40 you base this conclusion?

[Response] Sorry about the mix up of references in previous MS. We have deleted this reference in the revised MS.

[Comment 11] L. 30 'priming effect', here and in the rest of the text, change to: 'the priming effect'

[Response] Done as suggested.

[Comment 12] L.47 'melting' should be 'thawing' (ice melts, soil and meat thaw);

[Response] Done as suggested.

[Comment 13] L.47 depends (grammar); please check grammar throughout the manuscript

[Response] Following the reviewer's suggestion, we have asked Dr. Maggie C.Y. Lau from Princeton University and an English language editing service (*i.e.*, Springer

Nature Author Services) for language check. Please see the certification at the end of this response letter.

[Comment 14] L.54 'the strongest' compared to what?

[Response] This sentence has been modified as follows: “A warming climate is predicted to induce massive carbon (C) release from thawing permafrost, triggering a positive C-climate feedback” (Page 3, line 53-54).

[Comment 15] L. 58 'accompanied by', change to: 'due to'

[Response] This sentence has been modified as follows: “In these interactions, enhanced plant C fixation may mitigate part of soil C loss (Koven *et al.*, 2011), but it may also induce soil C loss by accelerating the turnover of native soil organic matter (SOM), the so-called ‘priming effect’ ” (Page 3, line 58-61).

[Comment 16] L. 64-65 'the fact that most C is stored in soil' – this is not very specific, rephrase entire sentence (62-65) and include something about the frozen state of the soil and whether or not plants can access it.

[Response] Following the reviewer’s suggestion, we have rephrased the sentence as follows: “Particularly, accompanying permafrost thaw and enhanced plant productivity under climate warming, more roots can grow down into the previously frozen soil horizon (Iversen *et al.*, 2015) where a large amount of C is stored (Hugelius *et al.*, 2014). This change in plant root distribution may subsequently trigger the priming effect in deep soils and make the prediction of post-thaw ecosystem C dynamics more complex” (Page 3, line 61-65).

[Comment 17] L.69-70 Why would the general statement that 'C and N cycles are closely coupled in most ecosystems' mean that N-availability is a key regulator of the PE in permafrost ecosystems'?? Even if you add 'permafrost' to 'ecosystems' in line 70, this is still not a good argument. Please improve.

[Response] Sorry about the poor logic. Following the reviewer’s comment, we have

rewritten the whole paragraph as follows: “Thermokarst is an abrupt permafrost thaw process (ground surface collapse caused by thawing of ice-rich permafrost) that can dramatically impact soil and hydrologic properties (Pizano *et al.*, 2014, Abbott & Jones, 2015, Olefeldt *et al.*, 2016). Post-thaw changes in soil nutrient, moisture, texture and pH (Abbott *et al.*, 2015, Finger *et al.*, 2016) may influence SOM turnover by altering the priming intensity (Cheng *et al.*, 2014, Nottingham *et al.*, 2015, Huo *et al.*, 2017). Among these changes, the widespread increase in nitrogen (N) availability after permafrost collapse (Harms *et al.*, 2013, Finger *et al.*, 2016), driven by enhanced N mineralization (Finger *et al.*, 2016) and the additional N released from thawing permafrost (Keuper *et al.*, 2012), may play an important role in regulating the priming intensity, since both vegetation growth and topsoil microbial activity in arctic (Sistla *et al.*, 2012, Wild *et al.*, 2015) and alpine ecosystems (Song *et al.*, 2007, Gao *et al.*, 2016, Kou *et al.*, 2017) are N limited” (Page 3-4, line 69-77).

[Comment 18] Additionally, this reasoning which is based on arctic and subarctic permafrost ecosystems, is used to build up to the hypothesis that N-availability will also regulate the PE in the Tibetan Plateau permafrost. However, further on in the manuscript it is mentioned that, whereas in arctic and subarctic ecosystems plant and microbial growth is N-limited, this is not the case in the Tibetan Plateau permafrost, where apparently C-limitation is prevailing (L. 120). Hence, not only this sentence, but also the entire build up to the hypotheses is not solid. Please improve.

[Response] Following the reviewer’s comments, we re-established our arguments by citing the related literatures on the Tibetan Plateau. By doing so, we notice that both vegetation productivity and microbial activity on the Tibetan Plateau are N limited. **On one hand, the widespread stimulation in vegetation production by N inputs** indicates that the vegetation production in Tibetan alpine grasslands is N limited (Fang *et al.*, 2012, Peng *et al.*, 2017). **On the other hand, the negative net N mineralization rate in alpine meadow and wetland** (Song *et al.*, 2007, Gao *et al.*, 2016), **high ratio of microbial N immobilization to total gross N mineralization in**

Tibetan alpine grasslands (1.6 ± 0.2) (Kou *et al.*, 2017) **and the increase in soil CO₂ flux after low N supply** (Deng *et al.*, 2015b) jointly suggested that **N availability could also constrain the function of microbial communities in this region**. Hence, microbial activity in Tibetan alpine permafrost is assumed to be co-limited by C and N availability. It is thus expected that once the energy limitation is alleviated by the supply of glucose, the regulation of N availability on the decomposition of soil organic matter will be more predominant. We have reorganized this paragraph and clearly mentioned these points in the revised MS (Page 4, line 75-77; Page 16, line 334-338). Notably, to improve the readability of the *Introduction* section, we have moved the descriptions about Tibetan permafrost from the *Introduction* session to the *Methods* session in the revised MS.

[Comment 19] L. 75 'Due to this point, several N addition experiments have been conducted...' Please re-write this section, it is crucial to your hypotheses but currently full of nonsense, I am sorry to say.

[Response] Sorry for the poor logic among these sentences in the original MS. Combining the suggestions from this reviewer and the other two reviewers, we have rewritten the whole paragraph as follows: “Among these changes, the widespread increase in nitrogen (N) availability after permafrost collapse (Harms *et al.*, 2013, Finger *et al.*, 2016), driven by enhanced N mineralization (Finger *et al.*, 2016) and the additional N released from thawing permafrost (Keuper *et al.*, 2012), may play an important role in regulating the priming intensity, since both vegetation growth and topsoil microbial activity in arctic (Sistla *et al.*, 2012, Wild *et al.*, 2015) and alpine ecosystems (Song *et al.*, 2007, Gao *et al.*, 2016, Kou *et al.*, 2017) are N limited. Despite this recognition, the link between N availability and the priming effect in permafrost ecosystems remains ambiguous because previous studies have revealed a positive (Wild *et al.*, 2014), a negative (Hartley *et al.*, 2010, De Baets *et al.*, 2016), and no effect (Wild *et al.*, 2014, De Baets *et al.*, 2016) of N addition on priming intensity. The diverging views probably reflect the ecosystem differences in microbial C and N demands (Wild *et al.*, 2016), which are closely related to the microbial

physiological responses. Thus, to better understand this discrepancy, it is urgently needed to reveal the underlying mechanisms governing this C-N interaction, especially the role of microorganisms” (Page 3-4, line 72-83). Thanks for your understanding!

[Comment 20] L. 96 'ultimately resulting in reduced decomposition and negative priming effect' – this goes too fast for me, please elaborate

[Response] Combining the suggestions from this reviewer and the other two reviewers, the paragraph has been reorganized and this sentence has been deleted in the revised MS (Page 4-5, line 85-100).

[Comment 21] L. 98 'induce shift', change to: 'induce a shift', and please check your use of articles in the entire manuscript

[Response] Combining the suggestions from this reviewer and the other two reviewers, the paragraph has been reorganized and this sentence has been deleted in the revised MS (Page 4-5, line 85-100). Nevertheless, we have carefully checked the grammars throughout the revised MS.

[Comment 22] L. 108 – 111 As I understand the priming effect (PE), it is an increase in SOM-derived CO₂ in response to a labile C input, without further specification about whether or not the microbial biomass carbon (C) increases. An increase in qCO₂ is an increase in SOM-derived CO₂ relative to microbial biomass C. Please explain in your manuscript, how exactly do you discern between the PE and an increase in qCO₂?

[Response] Thanks for your reminder! We agree that it is difficult to exactly discern the priming effect and changes in SOM-derived qCO₂ (Δ SOM-qCO₂), because of the potential non-independence between them. To resolve this issue, **we directly calculated the microbial CUE based on C:N stoichiometry** (Sinsabaugh *et al.*, 2013, Sinsabaugh *et al.*, 2016) following the Eq. (1) and Eq. (2):

$$CUE_{C:N} = CUE_{\max} [S_{C:N} / (S_{C:N} + K_N)] \quad (1)$$

$$S_{C:N} = (1/EEA_{C:N})(B_{C:N}/L_{C:N}) \quad (2)$$

where $S_{C:N}$ is a scalar that represents the extent to which the allocation of enzyme activities offsets the disparity between the C:N ratio of available resources and microbial biomass (Sinsabaugh *et al.*, 2016). The half-saturation constant K_N is set to 0.5. CUE_{\max} is the upper limit for microbial growth efficiency and is set to 0.6. $EEA_{C:N}$ is the ratio of enzyme activities directed toward acquiring C and N from the environment and is calculated as $BG / (NAG + LAP)$. $L_{C:N}$ is labile organic matter C:N ratio.

These additional analyses showed that **the CUE decreased after glucose addition (Fig. R2a) and increased after N supply (Fig. R2b). Moreover, the negative correlation between the priming effect and CUE was observed in both gradient experiment (Fig. R2c) and N addition experiment (Fig. R2d). These findings jointly support the hypothesis that decreases in SOM-derived C release (that is, the priming effect) could be due to increased microbial metabolism efficiency.**

Figure R2. Changes in C use efficiency (CUE) in thaw gradient experiment (a) and N addition experiment (b) and their associations with the priming effect (c-d). Data represent means + SE (standard error). Significant differences are denoted by different letters ($P < 0.05$).

Additionally, we would like to mention that the **microbial metabolic quotient could also be indexed as the CO₂ release rate relative to microbial biomass C without partitioning the SOM and glucose derivation**, as done by Spohn and Chodak (2015). By doing so, the CO₂ production rate used in the calculation is the total CO₂ release containing both SOM and glucose-derived CO₂ release. **This could reduce the potential data non-independence between the priming effect and $q\text{CO}_2$** . Based on this consideration, we re-analyzed the data and found that $q\text{CO}_2$ increased after glucose addition (Fig. R3a) but decreased after N supply (Fig. R3b). The positive correlation between the priming effect and $q\text{CO}_2$ was found in both gradient experiment (Fig. R3c) and N addition experiment (Fig. R3d), supporting the results of CUE. **Overall, based on these new analyses, we used the CUE data instead of SOM- $q\text{CO}_2$ in the revised MS, and also used the $q\text{CO}_2$ results as a complement.** Nevertheless, **we could delete the $q\text{CO}_2$ result if the reviewer still has this concern.**

Figure R3. Changes in metabolic quotients ($q\text{CO}_2$) in thaw gradient experiment (a) and N addition experiment (b) and their associations with the priming effect (c-d). Data represent means + SE (standard error). Significant differences are denoted by different letters ($P < 0.05$).

[Comment 23] L.120 'microbial activity in Tibetan alpine permafrost is more subjected to energy limitation owing to low soil organic C concentration', seems to not be in line with referred to study by the same authors. Crucial to current rationale, please revise with scrutiny;

[Response] As mentioned above, the higher C decomposability in Tibetan permafrost compared to the high-latitudes mentioned by Chen *et al.* (2016a) is only referred to the subsoils in active layer and permafrost layer, and this high C decomposability is mainly attributed to **the high proportion of labile SOM fraction rather than the high amount of labile SOM fraction.** However, in this study, the comparison of the microbial C or nutrient limitation between Tibetan and arctic permafrost ecosystems focused on the top organic soil. **Compared to the arctic tundra organic soils, the microbial activity in Tibetan thermokarst region is still limited by C,** indicated by

relatively lower C content (18.7% in our study site vs. 21.4~48.6% in arctic tundra) **and soil C:N ratio** (12.3 in Tibetan swamp meadow vs. 25.6~41.3 in arctic tundra) (Hartley *et al.*, 2010, De Baets *et al.*, 2016, Wild *et al.*, 2016).

Based on these new analyses, we deleted Chen *et al.* (2016a) *Nature Communications*, and added more new references to support the argument that topsoil microbial activity in Tibetan alpine ecosystem is assumed to be co-limited by C (Chen *et al.*, 2016b, Tian *et al.*, 2017) and N availability (Song *et al.*, 2007, Gao *et al.*, 2016, Kou *et al.*, 2017)” (Page 16, line 334-338).

[Comment 24] L.120 Additional to the above, I would like to ask the authors to provide more information on the soil quality. A table with soil parameters including at minimum soil C and N and DOC would be very interesting.

[Response] Following the reviewer’s comment, we have added a supplementary table (Table R1) including the key soil and microbial properties (soil moisture, bulk density, soil organic C content, organic C density, total N content, N density, soil C:N ratio, DOC concentration, TDN concentration, clay and silt content, microbial biomass C:N ratio, stoichiometric imbalance) in Supplementary Information of the revised MS.

Table R1. Soil properties in the top 15 cm along a typical permafrost thawing sequence on the Tibetan Plateau.

	Control	1 year	10 years	16 years
Moisture (wt %)	202.0±8.3 a	175.0±6.6 b	187.5±5.0 ab	154.8±9.5 b
Bulk density (g cm ⁻³)	0.29±0.01 b	0.31±0.02 ab	0.30±0.01 ab	0.32±0.01 a
SOC (g kg ⁻¹)	186.6±4.8 a	178.9±4.7 a	183.7±3.1 a	152.1±10.5 b
SOCD (kg C m ⁻³)	54.1± 1.4 a	55.5±1.5 a	55.1±0.9 a	48.7±3.3 a
STN (g kg ⁻¹)	15.2±0.4 a	14.8±0.3 ab	15.3±0.3 a	12.6±0.9 b
STND (kg N m ⁻³)	4.4±0.1 a	4.6±0.1 a	4.6±0.1 a	4.0±0.3 a
R _{C:N}	12.3±0.06 a	12.1±0.05 a	12.0±0.06 a	12.0±0.07 a
DOC (mg kg ⁻¹)	161.7±5.9 a	161.5±6.9 a	195.9±7.3 a	178.2±10.8 a
TDN (mg kg ⁻¹)	33.1±1.5 c	47.5±3.0 a	43.1±2.3 ab	37.8±1.8 bc
Clay (%)	2.1±0.04 a	2.1±0.05 a	2.1±0.03 a	2.0±0.06 a
Silt (%)	53.1±0.6 a	53.6±1.0 a	53.4±0.7 a	49.5±1.3 a

$B_{C:N}$	17.5 ± 0.4 a	17.1 ± 1.0 a	15.5 ± 0.6 ab	14.6 ± 0.7 b
$R_{TOC:TDN}/B_{C:N}$	0.28 ± 0.01 b	0.20 ± 0.01 c	0.29 ± 0.01 a	0.32 ± 0.02 a

Mean values \pm SE are displayed. SOC, soil organic carbon content; SOCD, soil organic carbon density; STN, soil total N content; STND, soil total N density; $R_{C:N}$, soil C:N ratio; DOC, dissolved organic carbon; TDN, total dissolved nitrogen; $B_{C:N}$, microbial biomass C:N ratio; $R_{TOC:TDN}/B_{C:N}$, stoichiometric imbalance calculated as the ratios of resource C:N ($R_{DOC:TDN}$) normalized to microbial biomass C:N ($B_{C:N}$).

[**Comment 25**] L. 133 (*hypothesis i*) magnitude? direction?

[**Response**] Based on the major comment from Reviewer 2#: “*The relationship between priming and N availability (and therefore N mining) is interesting and could be made a larger part of the manuscript. As of how it is really only mentioned in the discussion, but instead could be used in the introduction to really set up the discrepancy studies have observed and be presented as a hypothesis*”, we have reorganized the whole paragraph and this sentence has been deleted. Instead, **we proposed a hypothesis based on the earlier results from theoretical models and experiments** (Manzoni *et al.*, 2012, Mooshammer *et al.*, 2014b, Sinsabaugh *et al.*, 2016) as follows: “Microbial metabolic efficiency generally increases with N availability (Manzoni *et al.*, 2012, Mooshammer *et al.*, 2014a). This physiological adjustment can facilitate the microorganisms to cope with their N demand and maintain a balanced biomass C:N ratio (Mooshammer *et al.*, 2014b). Therefore, shifts in microbial metabolic efficiency may also contribute to the varied priming effects along the N gradient after thermokarst formation. However, to date, it remains obscure that whether and how microbial metabolic efficiency regulates the priming effect under different N conditions. Here, we hypothesize that elevated N availability after permafrost collapse inhibits the priming effect by increasing the microbial metabolic efficiency” (Page 5, line 94-103).

[**Comment 26**] L.134 (*hypothesis ii*) The 'how' in this hypothesis suggest that you were already certain before performing your research, that N-availability regulates the priming effect in alpine permafrost ecosystems. If so, please provide a reference.

[**Response**] As mentioned above, we have reorganized the whole paragraph based on

the comments from Reviewer 2# and this sentence has been deleted. We proposed a hypothesis based on the earlier results from theoretical models and some experiments (Manzoni *et al.*, 2012, Mooshammer *et al.*, 2014b, Sinsabaugh *et al.*, 2016) (Page 4-5, line 85-97). The related references have been provided to support the newly developed hypothesis.

[Comment 27] L.135. Please specify how you discern between an increase in SOM-respiration as a result of increased enzyme activity and an increase in SOM respiration per microbial biomass C. This is not clear to me.

[Response] Sorry for the confusion. In this study, we discerned the increase in SOM-respiration as a result of increased enzyme activity or an increase in SOM respiration per microbial biomass C based on the statistical relationships among them. Specifically, we conducted two linear regression analyses to examine the relationships between the priming effect and enzyme activity / $q\text{CO}_2$. The R^2 and P value from these relationships were then used to compare the impacts of these two variables on the priming effect. We observed the priming effect exhibited significant associations with microbial metabolic efficiency (*i.e.*, CUE and $q\text{CO}_2$) (see Figs. 4-5 in the revised MS), but did not find any significant relationship between the priming effect and enzyme activity (see Fig. 4 in the revised MS). Based on these results, we argued that lower microbial metabolic efficiency rather than higher enzyme activity accounted for the greater priming effect under low N availability. Nevertheless, it should be mentioned that we have reorganized the whole paragraph based on the comments from Reviewer 2# and this sentence has been deleted.

[Comment 28] L.145 'the soil is a silty loam with pH of' – please give uncertainty estimates and more information on soil parameters, this is crucial information for the reader to be able to understand your findings;

[Response] Following the reviewer's comment, we have added the uncertainty estimates of pH (Page 17, line 354-355) and also a table including the key soil parameters in the revised MS (Table R1).

[Comment 29] L.148-149 'within the past several decades' – it is UNCLEAR HOW THE AGE OF YOUR TRANSECT STAGES WAS DETERMINED, please elaborate or give an appropriate reference

[Response] The age of different collapse stages was estimated as follows: ① We first estimated the average rate of gully retreat ($\sim 8 \text{ m yr}^{-1}$) through the comparison to satellite images from 2007 to 2013 (Fig. R4) and repeated ground measurements between 2014 to 2016; ② We then determined the age for each collapse stage by dividing the distance between gully head and each site with the rate of gully retreat (Yang *et al.*, 2018). We have clearly mentioned this method in the revised MS (Page 18, line 369-373).

Figure R4. Two aerial photographs obtained from Google Earth depicting the development of thermo-erosion gully on the Tibetan Plateau from 2007 to 2013. The solid orange line describes the shape of the gully in 2007, and the dashed line represents the retreated outline of the gully six years later.

[Comment 30] L. 167 'exposed mineral soils were not included in the soil sampling' – seems to contradict with the 'silty loam' earlier?

[Response] Sorry for the inappropriate use of the word “mineral” in the original MS. Actually, the “exposed soil” in the original MS refers to the exposed subsoil C within the gully (as shown in the area within yellow line in Fig. R5), rather than the “mineral soil” defined by the C content ($<20\%$). To avoid the confusion, we have deleted word “mineral” in the revised MS. As shown in Table R1, the detailed texture in our study site is characterized by 2.1% of clay, 53.1% of silt and 44.8% of sand. Thus,

according to the USDA textural classification triangle, the soil in our study belongs to silty loam.

Figure R5. Picture showing vegetated patches and exposed patches within the thermo-erosion gully.

[Comment 31] L. 170 Did you perform bulk density measurements in the field in order to be able to express your results in meaningful units?

[Response] Yes, we did. Following the reviewer’s comment, we have added this data, soil organic C density and soil TN density in the supplementary Table S1 of the revised MS.

[Comment 32] L. 178 "to further demonstrate that" see my comment about 'positivistic urge'

[Response] Actually, the second experiment was done after observing the significant correlation between N availability and priming effect in the first experiment. To avoid the potential confusion, we have rephrased the sentences as follows: “After detecting the correlation between the priming effect and post-thaw N availability in the first experiment, we performed the second experiment to further rule out the potential impact of other concomitant factors besides N” (Page 20, line 414-417).

[Comment 33] L. 184 What was the final C:N ratio of the other treatments?

[Response] In the thaw gradient experiment, glucose addition treatment did not alter

the final soil C:N ratio, generating a final C:N ratio as 12:1 under all treatments. In the original N addition experiment, the C:N ratio under the water (W) and glucose addition (G) treatment was also 12:1, while the C:N ratio under nitrogen addition (N) and glucose + nitrogen treatment (GN) was 10:1.

It should be noted that, when replying the reviewer's comment [44], we noticed that the concentration of added N in the previous N addition experiment (3000 mg N kg⁻¹ soil) was far beyond the natural range of the difference in soil TDN concentration (20.1~50.8 mg N kg⁻¹ soil). **To resolve this issue, we took two months to conduct a new N addition experiment including ¹³C-labeled glucose treatment with two levels of N addition in the form of NH₄NO₃.** This approach resulted in the following treatments: water (control), glucose (G), low N, G + low N, high N and G + high N. The N addition treatments were designed to simulate the high N availability at the early stage of collapse. Given that the natural difference in TDN concentration between early- and late-stage sites ranged from 20.1 to 50.8 mg N kg⁻¹ soil (Yang *et al.*, unpublished data), the rate of NH₄NO₃ addition were set to 20 and 50 mg N kg⁻¹ soil in our low- and high-N treatment, respectively. Consequently, in our new N addition experiment, two levels of N addition did not alter the final soil C:N ratio, leading to a final C:N ratio as 12:1 under all treatments.

[Comment 34] L. 186 how much dry weight?

[Response] The dry weight is 20 g. We have clearly stated this point by revising this sentence as follows: “Before the experimental treatment, fresh sieved soil (approximately 20 g dried basis) was homogeneously mixed with the abovementioned substrates in 400-ml amber bottles with airtight lids” (Page 21, line 432-434).

[Comment 35] L. 188 How was water holding capacity determined?

[Response] Water holding capacity was measured gravimetrically using a sample that had been air-dried for 72 h and then rewetted until no more water could be absorbed

(Treat *et al.*, 2014). We have added these description in the *Methods* session of the revised MS (Page 23, line 479-480).

[Comment 36] L. 205 and further: when working with labeled substrates it is generally preferred to work with abundances (not delta13C), please adapt this.

[Response] Following the reviewer's suggestion, we have adapted this throughout the revised MS.

[Comment 37] L.232 and L.243-244 There is debate on whether this is indeed always a 'microbial available resource' (Mooshammer *et al.* 2014), please give a reference here, and why not also refer here to your nice supplementary figure where you express data in the more common way, C:N instead of DOC: TDN?

[Response] Following the reviewer's suggestion, we have added a reference (Wild *et al.*, 2015) to support the argument "The ratio of soil DOC to TDN was presumed a better representative of the available microbial resource stoichiometry than the bulk soil C:N ratio because the dissolved C and N are more easily available for microbial communities (Wild *et al.*, 2015)". We also referred to both $R_{C:N}/B_{C:N}$ and $R_{DOC:TDN}/B_{C:N}$ as follows: "The stoichiometric imbalance between resources and microorganism (both the total form ($R_{C:N}/B_{C:N}$) and labile form ($R_{DOC:TDN}/B_{C:N}$)) were further calculated as the ratios of the resource C:N ratios ($R_{C:N}$ and $R_{DOC:TDN}$) normalized to $B_{C:N}$ (Mooshammer *et al.*, 2014b)" (Page 23-24, line 486-493).

[Comment 38] L 245-246 I am not sure what you mean here?

[Response] To avoid the confusion, we have rephrased the sentences as follows: "Higher C:N imbalances correspond to lower N availability relative to C availability and could thereafter be used as a proxy of microbial N limitation (Manzoni *et al.*, 2012, Mooshammer *et al.*, 2014b, Wild *et al.*, 2015)" (Page 23, line 489-491).

[Comment 39] L.289 which protocol?

[Response] PLFAs were extracted from the soils following a previously described

method (Bossio & Scow, 1998). We have clearly stated this point in the revised MS (Page 26, line 552-553).

[Comment 40] L 295 unfinished sentence

[Response] We have rephrased the sentences as follows: “The abundance of individual fatty acids was determined as nmol per g of dry soil. PLFAs specific to bacteria (i14:0, i15:0, a15:0, i16:0, a17:0, i17:0, br-17:0, 16:1w7c, cy17:0, 18:1w7, 18:1w5, cy18:0 and cy19:0) and fungi (18:2 w6c, 18:1w9c) were quantified” (Page 27, line 557-559).

[Comment 41] L. 299 Why not also DOC?

[Response] Following the reviewer’s suggestion, we have added more analyses on the other soil parameters and thus have rephrased the sentence as follows: “First, one-way ANOVAs with LSD multiple comparisons were used to explore the differences in abiotic (soil moisture, SOC, TN, soil C:N ratio ($R_{C:N}$), **DOC**, TDN, clay and silt content) and biotic ($B_{C:N}$, C:N imbalance ($R_{C:N/B_{C:N}}$ and $R_{DOC:TDN/B_{C:N}}$)) variables as well as the priming effect along the thaw sequence” (Page 27, line 563-566).

[Comment 42] L. 300 Unclear how you 'measured' microbial N-limitation, did you determine a Threshold elemental ratio?

[Response] No, we didn’t. Instead, we used the C:N imbalance between resources and microorganism (both the total form ($R_{C:N/B_{C:N}}$) and labile form ($R_{DOC:TDN/B_{C:N}}$)) as a proxy of microbial N limitation. This index has been widely used in published literatures (Manzoni *et al.*, 2012, Mooshammer *et al.*, 2014b, Wild *et al.*, 2015). Higher C:N imbalances correspond to lower N availability relative to C availability and could thereafter be used as a proxy of microbial N limitation (Manzoni *et al.*, 2012, Mooshammer *et al.*, 2014b, Wild *et al.*, 2015). We have clearly mentioned these points in the revised MS (Page 23-24, line 489-493).

[Comment 43] L. 315 Were your data normally distributed?

[Response] Most of the data we used are normally distributed. For the data that are not normally distributed, we used log transformation to normalize the data before analysis of variance analyses. We have clearly stated this point in the revised MS (Page 27, line 562-563).

[Comment 44] L. 350 Fig. 3c I interpret the not significant interaction between glucose and nitrogen as indicative of NO effect of nitrogen availability on the PE, please discuss this

[Response] Thanks for the reviewer's insightful comments! The comment enabled us to have a deeper thinking on this issue. We first noticed that the concentration of added N in the N addition experiment (3000 mg N kg⁻¹ soil) was far beyond the natural range of the difference in soil TDN concentration (20.1~50.8 mg N kg⁻¹ soil) between the thaw stages (Yang *et al.*, unpublished data). This high N addition would completely relieve the microbial N limitation, which in turn results in the microbial C limitation. This could be the reason why the priming intensity was similar before and after N addition, as mentioned by the reviewer (Fig. 3c in original MS). Given that this high N addition is not realistic in natural ecosystems, we have removed the results of the original N addition experiment in the revised MS.

To resolve this issue, we then took two months to conduct a new N addition experiment including ¹³C-labelled glucose treatment with two levels of N addition in the form of NH₄NO₃. This approach resulted in the following treatments: water (control), glucose (G), low N, G + low N, high N and G + high N. The N addition treatments were designed to simulate the high N availability at the early stage of collapse. Given that the natural difference in TDN concentration between early- and late-stage sites ranged from 20.1 to 50.8 mg N kg⁻¹ soil (Yang *et al.*, unpublished data), the rate of NH₄NO₃ addition were set to 20 and 50 mg N kg⁻¹ soil in our low- and high-N treatment, respectively. **The results derived from our new N addition experiment showed that there was a significant interaction between glucose and N addition on SOM-derived C release** ($P < 0.05$, Fig. R6). **The enhancement in**

SOM-derived C release induced by glucose addition decreased with added N concentrations. More directly, the result of one-way mixed model confirmed that N addition led to a significant decrease in the priming effect ($P < 0.01$; Fig. R7), supporting our hypothesis. We have added all the results derived from the new N addition experiments in the revised MS (Page 8, line 163-169). Thanks again for the reviewer’s insightful comments, which makes us have a deeper thinking on this issue!

Figure R6 The SOM-derived CO₂-C release from soils amended with different levels of glucose and N addition. Data represent means + SE (standard error).

Figure R7 The priming effect from late-stage soils amended with different levels of N. Data represent means + SE (standard error). Significant differences are denoted by different letters ($P < 0.05$).

[Comment 45] L. 366 decrease relative to what?

[Response] It is relative to the control treatment. Combining the suggestions from this reviewer and the other two reviewers, we have rephrased this sentence as follows:

“Labile C supply also significantly decreased microbial C use efficiency (CUE) ($P < 0.01$; Fig. 4b) and enhanced metabolic quotient ($q\text{CO}_2$) ($P < 0.01$; Fig. 4c)” (Page 9, line 180-181).

[Comment 46] L. 367 Fig. 5a, b Please discuss your interaction effects (Fig. 5) here and in the discussion. L. 367 Fig 5a If your 'time' is the N-gradient, than what does it mean that there is no interaction effect in Fig. 5a (no effect of time/N-gradient on the increase in $q\text{CO}_2$ due to the PE?) whereas there IS an interaction effect in Fig. 5b? Please discuss this (apparent?) contradiction in the text as well.

[Response] Very good comment! Following the reviewer’s comments, we have discussed the interaction effects as well as the contradiction between the two experiments in the Supplementary Information of the revised MS (Page 1, line 9-22). With regard to Fig. 5a in the original MS, **no interaction between thaw time and glucose addition** suggested that the significant enhancement on SOM- $q\text{CO}_2$ induced by glucose addition did not vary with thaw sequence. In contrast, in Fig. 5b, **the significant interaction between glucose and nitrogen addition** ($G \times N$) on SOM- $q\text{CO}_2$ ($P = 0.02$), suggesting that the effect of glucose addition on SOM- $q\text{CO}_2$ was dependent on N addition. Specifically, the increment in $q\text{CO}_2$ induced by glucose addition decreased under high N condition (Fig. 5b). This contradiction observed here could be attributed to the following two aspects. **First, the “time” gradient can only represent different collapse stages but cannot be fully equivalent to the N gradient given that many other factors also varied with thaw time** (Table R1). These confounding factors together with N availability co-affected the pattern in the $q\text{CO}_2$ across the thaw sequence. It is thus normal to find different effects of “time” and “nitrogen” treatment on the $q\text{CO}_2$. **Second, the different range of TDN concentration between the two experiments may also result in the apparent contradiction.** As mentioned above, the concentration of added N in the N addition experiment ($3000 \text{ mg N kg}^{-1} \text{ soil}$) was far beyond the natural range of the difference in soil TDN concentration ($20.1\sim 50.8 \text{ mg N kg}^{-1} \text{ soil}$) among the thaw stages.

As mentioned above, given that this high N addition is not realistic in natural ecosystems, **we have conducted a new N addition experiment** including ^{13}C -labelled glucose treatment with two levels of N addition in the form of NH_4NO_3 . This approach resulted in the following treatments: water (control), glucose (G), low N, G + low N, high N and G + high N. The rate of NH_4NO_3 addition were set to 20 and 50 mg N kg^{-1} soil in our low- and high-N treatment, respectively. **The results of the new experiment also showed that there was a significant interaction between glucose and N addition on $q\text{CO}_2$ ($P = 0.04$, Fig. R8). The magnitude of $q\text{CO}_2$ promoted by glucose addition decreased with increasing N concentration.**

It should be noted that the N addition experiment was mainly conducted to examine the effect of increased N availability rather than the interaction between glucose and N treatment. Due to this point, we preferred to explore the effects of N addition directly through one-way mixed-effect model. This data analysis is also the most frequently used method in previous N addition experiments (Nottingham *et al.*, 2012, Koranda *et al.*, 2013, Chen *et al.*, 2014, Wang *et al.*, 2014b). Thus, we presented all the results of the one-way mixed-effect model in the main text. Nevertheless, **we also provided the results of two-way mixed-effect models in the Supplementary Information, and clearly mentioned the interaction effect in the *Result* section. Additionally, we also added additional discussions on the reasons for the contradictions between the two experiments in Supplementary Information of the revised MS (Page 1, line 9-22).**

Figure R8 Microbial metabolism quotient ($q\text{CO}_2$) under different levels of glucose and N addition. Data represent means + SE (standard error).

[Comment 47] L. 368 Fig. 5c: I see a saturating function, which linear increase are you referring to?

[Response] According to the previous experimental results, it should be a saturating function. However, as mentioned above, we have conducted a new N addition experiment in the revised MS. The latest results showed that there were no saturating functions, but linear increase in the new correlations between priming effect and CUE (Fig. 4e, Fig. 5c) and $q\text{CO}_2$ (Fig. 4f, Fig. 5d). We have clearly mentioned these points in the revised MS (Page 9-10, line 188-189, line 200-201).

[Comment 48] L. 368 Fig 5c: please explain the reader in your methods or results section which information this graph offers, since it is not an easy to interpret graph. If I understand it correctly, what you plot is in fact the increase in $q\text{CO}_2$ DUE TO the PE against the PE itself. Would this mean that we would expect a linear 1:1 relationship if all increase in $q\text{CO}_2$ was due to the PE, and all deviation would be changes in $q\text{CO}_2$ caused by other factors than the PE? I might be wrong, please explain and discuss it in the text.

[Response] We would like to mention that, the regression analysis was conducted to examine the relationship between the priming effect and the change in SOM- $q\text{CO}_2$. When the regression was statistically significant ($P < 0.05$), it indicated that the

change in SOM- $q\text{CO}_2$ was responsible for the varied priming pattern along the thaw sequence. We would also like to mention that if all increase in $q\text{CO}_2$ was due to the changes in respiration, we would expect a linear relationship with $r^2=1$, rather than a linear 1:1 relationship. In this study, the r^2 is not equal to 1, which means that other factors could also influence the relationship between these two parameters.

Anyhow, we agree with the reviewer that, there are potential data non-independency between the priming effect and $\Delta\text{SOM-}q\text{CO}_2$. **To resolve this issue, we directly calculated the microbial CUE based on C:N stoichiometry. The negative correlation between the priming effect and CUE observed in both gradient experiment (Fig. R2c) and N addition experiment (Fig. R2d) jointly supported the hypothesis that decreases in the priming effect could be due to the increased microbial metabolism efficiency.** In addition, as done in previous studies (Carrillo *et al.*, 2014, Wang *et al.*, 2014a), we have also used $q\text{CO}_2$ as a proxy of carbon use efficiency (CUE) but modified all the $\Delta\text{SOM-}q\text{CO}_2$ to $q\text{CO}_2$ in the revised MS to reduce the potential data non-independence. **Nevertheless, we could delete the $q\text{CO}_2$ result if the reviewer still has this concern.**

[Comment 49] L. 379-380 Based on the findings presented in Chen et al. 2016, which show that the potential C-losses due to permafrost thawing of the Tibetan Plateau permafrost are relatively low (C-loss by 2100 under RCP8.5): with roughly 0.28-0.37 Pg C from Tibetan Plateau permafrost (Chen 2016), vs. 100 Pg C (\pm 3% s.e.) from northern circumpolar permafrost soils (Schuur 2015), I find this statement ('important implications ...feedback to climate warming') not applicable to this study.

[Response] Following the reviewer's suggestion, we have deleted this argument in the revised MS.

[Comment 50] L. 383-384 Please rephrase this sentence into a more modest statement, reflecting your actual results. The priming effect in this study was not actually measured in the field, but in an incubation experiment. You can not conclude

from this that 'the supply of labile C induced a widespread positive priming across the thaw gradient'.

[Response] Following the reviewer's comment, we have rephrased the sentence to avoid overstating the laboratory incubation results as follows: "In our laboratory incubation, the supply of glucose induced soil C loss from all collapse stages along the thermo-erosion gully, with the maximum loss of 26.6% from the late-stage topsoil" (Page 10, line 207-209).

[Comment 51] L. 386 Please compare your priming effect results with the referred studies 16 and 29, using actual numbers instead of using the phrasing 'barely affected'.

[Response] Following the reviewer's suggestion, we have added more detailed information about other studies as follows: "In our laboratory incubation, the supply of glucose induced soil C loss from all collapse stages along the thermo-erosion gully, with the maximum loss of 26.6% from the late-stage topsoil. The response in our organic soil was comparable to that of mineral soils (14~31%) (Wild *et al.*, 2014, Wild *et al.*, 2016), but much larger than that of organic soils in arctic ecosystems (-3~7%) (Hartley *et al.*, 2010, De Baets *et al.*, 2016, Wild *et al.*, 2016)" (Page 10, line 207-211).

[Comment 52] L. 391 Please express your results per volume, not per area, using bulk density data;

[Response] Following the reviewer's suggestion, we have expressed soil organic C density and soil total N density per volume in the Table R1 in the Supplementary Information of the revised MS (Page 4, line 50-56).

[Comment 53] L.393 What do you mean with 'stronger response to organic C input'? Which experiment do you refer to?

[Response] The "stronger response to organic C" refers to the higher magnitude of the priming effect observed in this study compared to that in arctic tundra, which is

based on the gradient experiment. We have clearly stated this point in the revised MS (Page 10-11, line 219-221).

[Comment 54] L. 401 Fig. 1e does not back up this statement;

[Response] Sorry for the typo. Combining the suggestions from this reviewer and the other two reviewers, the paragraph has been reorganized and this sentence has been deleted in the revised MS.

[Comment 55] L. 403 Fig. 1f does not show this

[Response] Combining the suggestions from this reviewer and the other two reviewers, the paragraph has been reorganized and this sentence has been deleted in the revised MS.

[Comment 56] L. 485 -486 'the efficiency of C incorporated into microbial biomass versus released as CO₂' this sounds as carbon use efficiency (CUE), you claim this is the 'reciprocal of qCO₂'. However, "while both the CUE and the qCO₂ are related to C use, they cannot directly be converted into each other" (Spohn 2014). Please take this difference between the CUE and the qCO₂ into account in your Discussion.

[Response] Very good comment! We acknowledge that $q\text{CO}_2$ does not equal to carbon use efficiency (CUE) and have clearly mentioned the difference in the revised MS (Page 26, line 546-547). In this study, we used $q\text{CO}_2$ as a proxy of carbon use efficiency (CUE), as done in previous studies (Carrillo *et al.*, 2014, Wang *et al.*, 2014a). To further explore whether $q\text{CO}_2$ could indirectly reflect CUE, we calculated the microbial CUE based on C:N stoichiometry (Sinsabaugh *et al.*, 2016), and examined the relationship between the two parameters. Our additional analyses revealed **the negative correlation between microbial CUE_{C:N} and $q\text{CO}_2$ (Fig. R9), demonstrating that higher $q\text{CO}_2$ corresponded to lower CUE.**

Figure R9. Relationship between qCO_2 and CUE in the gradient experiment (a) and N addition experiment (b).

[Comment 57] Fig. 1: The + G bars are in my opinion superfluous (we can understand this from adding the cumulative priming effect and the control CO_2 release). I am not sure which additional information the analysis of the total CO_2 release offers, if presented here at least it should be discussed in the text. Further, it would be interesting to know if there are interactive effects between time and G-treatment/PE.

[Response] We agree with the reviewer that there are some information overlap in panel (e–g). Given that SOM-derived CO_2 under the glucose treatment (panel g) is more informative than that under control treatment (panel f), we have deleted the panel (f) and kept the panel (g) in the revised figure. We also added a graph

representing the changes in the relative priming effect along the thaw sequence in panel (f) and discussed that content in the main text.

Regarding the interaction effect, we would like to mention that two-way mixed-effects models cannot be used to analyze the data of priming effect (As depicted in Table R2). By contrast, only one-way ANOVA can be used to explore how this priming effect varied along the thaw time, as we have done in Fig. 1e.

Table R2. The datasheet for SOM-derived CO₂ release and the priming effect (PE) in this study.

	A	B	C	D	E
1	Time	Treatment	SOM-CO2	PE	
2	Grassland	Glucose	6.00	1.14	
3	Grassland	Glucose	6.82	1.22	
4	Grassland	Glucose	8.20	1.83	
5	Grassland	Glucose	6.53	1.46	
6	Grassland	Glucose	6.59	1.10	
7	early-stage	Glucose	6.69	0.37	
8	early-stage	Glucose	9.21	0.95	
9	early-stage	Glucose	7.49	0.53	
10	early-stage	Glucose	7.64	0.54	
11	early-stage	Glucose	6.69	1.02	
12	mid-stage	Glucose	8.63	0.97	
13	mid-stage	Glucose	8.08	0.95	
14	mid-stage	Glucose	8.22	1.23	
15	mid-stage	Glucose	7.13	0.78	
16	mid-stage	Glucose	8.20	0.58	
17	late-stage	Glucose	8.55	2.06	
18	late-stage	Glucose	7.70	1.77	
19	late-stage	Glucose	9.28	1.94	
20	late-stage	Glucose	7.20	1.52	
21	late-stage	Glucose	7.67	1.37	
22	Grassland	water	4.86	-	
23	Grassland	water	5.60	-	
24	Grassland	water	6.37	-	
25	Grassland	water	5.07	-	
26	Grassland	water	5.49	-	
27	early-stage	water	6.32	-	
28	early-stage	water	8.67	-	
29	early-stage	water	6.95	-	
30	early-stage	water	6.37	-	

Although two-way mixed-effect model cannot be used to analyze the data of priming effect, we examined the interactive effects between thaw time and glucose addition on SOM-derived CO₂-C release. **The analyses indicated that there was a significant interaction between thaw time and glucose addition on SOM-derived CO₂-C release (Fig. R10), suggesting that the magnitude of the positive effect of glucose addition on SOM-derived CO₂-C release depended on the thaw stage. Specifically, the glucose effect was larger in the late and grassland stage than in**

early and middle stage. This result is consistent with our one-way ANOVA result shown in Fig. 1e. We have added this supplementary figure in the revised MS.

Figure R10. Cumulative SOM-derived CO₂-C release in control soils and glucose amended soils across different collapse stages. Data represent means + SE (standard error).

[Comment 58] Fig. 2: To better understand what the result presented here indicates, I would like to have to extra panels: Microbial C:N and DOC. Could the imbalance also mean a surplus of available C?

[Response] Following the reviewer’s suggestion, we have added more panels including DOC and microbial C:N (B_{C:N}) in the revised Figure 2. The changes in soil and microbial properties along the thaw sequence (including DOC and B_{C:N}) were presented in the Table R1. The results showed that despite the concentration of soil dissolved organic C (DOC) not exhibiting significant changes along the thaw sequence ($P = 0.06$), the concentration of soil total dissolved nitrogen (TDN) increased significantly during the early stage and subsequently dropped with collapse time. Meanwhile, a significant decrease in microbial C:N ratio was found at the late-stage site ($P < 0.05$). Correspondingly, the C:N imbalance between resources and microorganism increased significantly with collapse time (Table R1). It has been suggested that biomass stoichiometry reflects the stoichiometric requirements of

microorganisms (Mooshammer *et al.*, 2014a) and is not suitable as an indicator for nutrient limitation (Xu *et al.*, 2013). Therefore, we used the C:N imbalance to represent the microbial N limitation: higher C:N imbalances correspond to higher microbial N limitation (Mooshammer *et al.*, 2014a). Higher imbalance means lower availability of N relative to C (Mooshammer *et al.*, 2014a), but does not reflect that there is a surplus of available C. We have clearly mentioned these points in the *Results* session of the revised MS (Page 23, line 486-491).

[Comment 59] Fig. 3 The colour scheme used here to indicate 'treatment' is the same as used in the previous graph to indicate 'years since collapse'. Please make this less confusing.

[Response] Following the reviewer's comment, we have changed the color scheme throughout the revised MS.

[Comment 60] Fig. 5 I think your unit for the qCO_2 is not correct: why do you include soil here? Please explain in your methods the difference between your approach and, for example, Spohn 2014.

[Response] Sorry for the carelessness. We have corrected the unit in the revised MS. The difference between our SOM- qCO_2 and traditional qCO_2 was that we used isotope labelling technique to quantify the fraction of the total respiration and microbial biomass pool that is derived from SOM. This technique makes it possible to calculate the metabolic quotient corresponding only to the SOM-C (Carrillo *et al.*, 2014). Nevertheless, given the potential data non-dependency, we have deleted the results of SOM-derived qCO_2 , and instead used the traditional qCO_2 in the revised MS (Page 26, line 542-549).

[Comment 61] Fig. 6 I much appreciate this schematic representation, which I believe improves the understanding of the context of the results presented in this manuscript. However, I also have many questions and comments related to this Figure:

6a. Please mention in the legend what F/B (fungal bacteria ratio) stands for, it is mentioned in the main text but not in the legend;

[Response] Done as suggested.

[Comment 62] 6b. Why 'bacterial' biomass and not 'microbial' biomass, this is not discussed in the text?

[Response] Sorry for the confusion. What we want to highlight here is that the changes in microbial metabolic efficiency could be partly attributed to the **changes in microbial community structure**. To avoid the confusion, we have deleted the bacterial biomass and used the F/B ratio to represent microbial community structure in the revised figure (Fig. R11). We have clearly mentioned these points in the revised MS (Page 37, line 810-824).

Figure R11. Conceptual scheme showing variations in the priming effects along the thaw sequence and associated mechanisms.

[Comment 63] 6c. This Figure suggest that there is a rather thick organic layer in the grassland stage, which contradicts the reported very thin organic layer in Tibetan permafrost areas due to a dry climate (Gao et al., 1985; Wu & Zhang, 2010)? Please adapt or include explicitly in your Discussion what this means for the general relevance of your data on the priming effect in organic soil of Tibetan alpine permafrost;

[Response] In our swamp meadow site, there is a thick (0.7 m) organic layer (>20% SOM content) (Fig. R12). Similar thick organic layer was also reported in Tibetan alpine wetland (Ma *et al.*, 2016). Nevertheless, as the reviewer mentioned, this thick organic layer observed in swamp meadow or wetland is different from the thin organic layer in two other grassland types (alpine steppe and alpine meadow) on the Tibetan Plateau. **Considering these differences and the suggestions from other reviewers, we have reorganized the whole manuscript to completely focus on the swamp meadow where thermokarst mainly develops across the Tibetan permafrost region.** In addition, to avoid the confusion, we have deleted the layer of organic soil and mineral soil in the revised figure (Fig. R11), and clearly mentioned these points in the revised MS (Page 16, line 331-333; Page 17, line 352-354).

It should be highlighted that **the swamp meadow is of critical importance for the Tibetan C cycle for the following two reasons.** On one hand, **it has the highest soil organic C density (SOCD) among the three major grassland types on the plateau.** This high C density in swamp meadow is comparable with those across the northern circumpolar permafrost regions (65.0 vs. 58.2 kg C m⁻²) (Ding *et al.*, 2016). Based on the C stock estimated in the swamp meadow (3.3 Pg C) and the vertical distributions of relative proportions (Ding *et al.*, 2016), **about 50% of the soil C is stored in the thick organic layer, which equals to 1.65 Pg C.** This accounts for about 10% of the total soil C stock across the Tibetan alpine grassland. On the other hand, **swamp meadow is also the typical ecosystem where the thermokarst mainly develops across the Tibetan permafrost region** due to its high ground ice content in permafrost layer (Mu *et al.*, 2016, 2017).

Figure R12. Vertical distribution of soil organic matter (SOM) (mean + SE) in the study site. The SOM content was estimated from the soil organic C concentration using the conversion factor of 1.724. Mean values with different letters indicate significant differences among soil depths ($P < 0.05$).

[Comment 64] 6d. this Figure also suggest that the permafrost is contains ice-wedges in the intact 'grassland'-stage, the melting of which can indeed lead to severe karst. However, as I understood from Chen et al. 2016 (Supplementary Table 2), the Tibetan plateau permafrost is particularly dry and does not have a high ice-content. Please address this in the text and/or make consistent with this Figure. Also, to prevent confusion when studying this Figure 6, please make the site description more informative and consistent with previous studies on Tibetan Plateau permafrost (or indicate why and how this specific site is different from other sites on the Tibetan plateau and how this might affect the interpretation of the results).

[Response] As mentioned above, the permafrost layer in our study site do contain much ice (Fig. R13). This is a main characteristic of swamp meadow soil, and that is also why the thermokarst mainly occurs in this grassland type. Nevertheless, we could not determine whether these ices belong to ice-wedge. Thus, **to avoid the confusion, we have removed 'ice-wedge' in the revised figure (Fig. R11).**

We agree with the reviewer that this high ground ice content in swamp meadow is different from other grassland types (alpine steppe and alpine meadow) across Tibetan

permafrost region. To avoid the confusion, we have clearly mentioned this point as follows: “In contrast to the ice-poor permafrost deposits in two other grassland types (alpine steppe and meadow) (Ding *et al.*, 2016), the permafrost under swamp meadow is characterized as ice-rich due to its poor drainage (Mu *et al.*, 2017, Yang *et al.*, 2017)” (Page 16-17, line 329-331). Considering these differences and the suggestions from other reviewers, **we have reorganized the whole manuscript to focus on the swamp meadow where thermokarst mainly develops across the Tibetan permafrost region.**

Additionally, we also added one paragraph with more detailed information of the study site (*e.g.*, vegetation type, plant composition, root distribution, soil type, and permafrost condition) to demonstrate its representative for the upland thermokarst region on the Tibetan Plateau in the *Methods* session of the revised MS. Thanks for your understanding!

Figure R13. Picture showing the headwall of the thermo-erosion gully on the Tibetan Plateau. The grey dotted line indicates the position of the permafrost table. Ice in the permafrost layer (white particles) can be clearly seen.

[Comment 65] 6e. THE PLANTS HERE ARE ROOT-LESS. For this particular article that is inappropriate, since the priming effect is driven by labile C-inputs from plant-roots. It is thus relevant to see which layers of the soil these roots have access to, and whether this changes after permafrost thawing. Please adapt the figure

accordingly.

[Response] Very good comment! The rooting depth of the study site is approximately 70 cm, and the root distribution did not reveal significant changes along the thaw sequence (Fig. R1). Based on these additional measurements, we have adapted the figure accordingly.

[Comment 66] 6f. Do plants not grow on the mineral soil? If so, the priming effect would only occur in the thin organic layer, which is not frozen even in the grassland stage.

[Response] Yes, plants can grow on the mineral soil, and thus the priming effect could occur in the both organic and mineral soil.

[Comment 67] 6g. The Figure is slightly misleading as it suggests that the results explain processes in the permafrost itself, or at least in the thawed mineral soil layer. In fact, only the organic soil was sampled. This Figure needs to be adapted to reflect that.

[Response] Very good suggestion! We have adapted the figure accordingly.

[Comment 68] 6h. Are there no carbon dynamics in the mineral soil? Only the organic layer was sampled, yet from this Figure it becomes clear that in the later stages the organic layer is not the main soil pool. How much of the total Tibetan Plateau permafrost soil is stored in these organic patches? Perhaps change the title to soil carbon dynamics in the organic layer, in order to not overstate findings?

[Response] Regarding the mineral soil C dynamics, we think that it will exist due to the following two reasons: 1) plants could reach the mineral soil layer (Fig. R1) and produce root exudates; 2) the downward leaching of the DOC may also induce the fresh C input to the mineral soil.

Regarding the C pool size in organic soils, we would like to mention that **the organic soil layer is still the main soil C pool of the active layer even at the later stages.**

Specifically, the proportion of SOCD stored in the organic layer in the 1-m soil profile was estimated at ~100 %, ~ 76%, ~53%, and ~64% in the undisturbed grassland, early-, mid- and late-stage sites, respectively (Fig. R14). More broadly, organic soils across the whole Tibetan alpine swamp meadow is also an important C pool. Based on the C stock estimated in the swamp meadow (3.3 Pg C) and the vertical distributions of relative proportions (Ding *et al.*, 2016), about 50% of the soil C is stored in organic layer, which equals to 1.65 Pg C. This accounts for about 10% of the total soil C stock across the Tibetan alpine grassland. Nevertheless, **following the reviewer’s comment, we have revised the title as follows**: “Nitrogen availability regulates topsoil carbon dynamics after permafrost thaw by altering microbial metabolic efficiency”.

Figure R14. Vertical distributions of soil organic matter (SOM) (mean + SE) (a-d) and relative SOCD proportions (e-h) at 10-cm intervals (mean + SE) in undisturbed grassland (a, e), 1-year (b, f), 10-year (c, g) and 16-year (d, h) of permafrost collapse. The SOM content was estimated from the soil organic C content using the conversion factor of 1.724. The brown shaded regions in panel (e-h) indicate the position of the organic layer (SOM content > 20%). The SOCD for a give depth was calculated using the following equation: $SOCD = T \times BD \times OC$, where T, BD and OC are soil thickness (cm), bulk density ($g\ cm^{-3}$) and organic C content ($g\ kg^{-1}$), respectively.

[Comment 69] One last general comment: Without stating here that all research

should strictly follow the principles of the critical rationalism (I know this is unrealistic), I would like to ask the authors to acquaint themselves with the idea behind 'falsification' and the importance of formulating testable hypotheses. While the work presented here is truly interesting, the authors seem to want to 'proof that' nitrogen availability regulates soil carbon dynamics. In my view, wanting to 'proof' can reduce scientific open-mindedness (and the validity of drawn conclusions). For example, in the literature, multiple mechanistic explanations have been suggested for the priming effect. These include, but are not limited to, 'microbial N mining'. Other explanations are 'energy limitation', 'co-metabolism' and 'microbial community shifts' (Kuzyakov 2000, Fontaine 2007, Bengtson 2000). In field-situations, the rhizosphere priming effect (induced by living plant roots, as opposed to the PE which is induced by a substrate added under laboratory conditions, Huo 2017) can be influenced by even more factors, such as water availability, pH values, redox potentials and soil aggregation (Bengtson 2000, Huo, 2017). None of these alternative underlying mechanisms are addressed by the authors. Neither is the difference between what can be measured in an incubation study versus what actually happens in the field (priming effect versus rhizosphere priming effect). Moreover, the way the data are currently presented does not allow the reader to judge whether another explanation than the one presented by the authors (N availability) could also be considered. I would like to see this changed, throughout the manuscript (Introduction, Results, Discussion).

[Response] Very good comment! We do agree with the reviewer that there are multiple other mechanistic explanations and influencing factors for the priming effect. Following the reviewer's comments, **we analyzed the relationship between the priming effect and other factors (i.e., soil C:N ratio, DOC concentration, microbial C:N, pH, clay, silt and sand content)** and found that there were **no significant correlations between these factors and the priming effect** (Fig. R15).

Figure R15. Relationship between the priming effect and various soil properties including (a) soil C:N ratio, (b) dissolved organic C concentration (DOC), (c) microbial biomass C:N (MBC:MBN), (d) pH, (e) clay and (f) silt content in the top 15 cm.

Following the reviewer’s comments, we also **reorganized the Introduction session to provide more possible reasons for the changing priming effect along the thaw gradient** as follows: “Thermokarst is an abrupt permafrost thaw process (ground surface collapse caused by thawing of ice-rich permafrost) that can dramatically impact soil and hydrologic properties (Pizano *et al.*, 2014, Abbott & Jones, 2015, Olefeldt *et al.*, 2016). **Post-thaw changes in soil nutrient, moisture, texture and pH** (Abbott *et al.*, 2015, Finger *et al.*, 2016) **may influence SOM turnover by altering the priming intensity** (Cheng *et al.*, 2014, Nottingham *et al.*, 2015, Huo *et al.*, 2017)” (Page 3, line 69-72). Nevertheless, based on the comments of reviewer 2# **“the relationship between priming and N availability (and therefore N mining) is interesting and could be made a larger part of the manuscript. As of how it is really only mentioned in the discussion, but instead could be used in the introduction to really set up the discrepancy studies have observed and be presented as a hypothesis.”**, we did not reduce the part regarding the role of N availability in regulating the priming effect throughout the *Introduction* session. Thanks for your understanding!

Following the reviewer's comments, we further added two paragraphs in the Result session to depict the changes in other factors after permafrost collapse and their linkage to the priming effect as follows: “Permafrost collapse had significant influences on abiotic and biotic parameters (Table S1). The soil moisture significantly decreased in the vegetated patches after permafrost collapse ($P < 0.05$). Similarly, both SOC and total N (TN) content decreased at the late-stage site ($P < 0.05$; Table S1). Despite the concentration of soil dissolved organic C (DOC) not exhibiting significant changes along the thaw sequence ($P = 0.06$), the concentration of soil total dissolved nitrogen (TDN) increased significantly during the early stage and subsequently dropped with collapse time, leading to a 20.5% reduction at the late-stage site ($P < 0.01$, Table S1). Permafrost collapse also affected the microbial stoichiometry in addition to soil properties, with a significant decrease in microbial C:N ratio ($B_{C:N}$) at the late-stage site ($P < 0.05$). The microbial N limitation, indicated by a stoichiometric imbalance between resources and microorganisms (both the total form ($R_{C:N}/B_{C:N}$) (Fig. S4a) and the labile form ($R_{DOC:TDN}/B_{C:N}$)(Table S1)), correspondingly increased with collapse time, resulting in maximal microbial N limitation at the late-stage site. Although permafrost collapse altered many biotic and abiotic variables, variations in the priming effect along the thaw sequence were only negatively related to TDN concentration ($P < 0.05$; Fig. 2b,f) and was positively associated with stoichiometric imbalance ($P < 0.05$; Fig. 2d,h). No significant correlation was observed between the priming effect and other factors, such as the soil DOC concentration, C:N ratio, $B_{C:N}$, pH, clay and silt content (Fig. 2, Fig. S5). These results indicated that soil N availability and C:N imbalance might induce the variations in primed CO_2 -C release along the thaw sequence” (Page 7-8, line 138-162).

Additionally, following the reviewer's comment, we added one paragraph in the Discussion session to discuss the other mechanisms and soil factors that may affect the priming effect as follows: “While our study provides the experimental

evidence for the role of N availability in regulating the priming effect after permafrost thaw, some uncertainties still exist. First, given that the priming effect is determined through laboratory incubation without plants, biotic attributes regulating the rhizosphere priming in situ have not been considered. For example, the quantity (Cheng *et al.*, 2014, Huo *et al.*, 2017), quality (Wild *et al.*, 2014, Qiao *et al.*, 2016) and timing of the C inputs (Qiao *et al.*, 2014) could affect the priming intensity, but it is challenging to simulate these biotic impacts with substrate addition under laboratory conditions. In situ rhizosphere priming experiments are thus needed to better elucidate this important plant-soil interaction in permafrost ecosystems. Second, this study concentrates on soil N availability and microbial metabolic efficiency, but other edaphic properties, such as water availability (Zhu & Cheng, 2013), SOM quality (Qiao *et al.*, 2016) and soil aggregation (Huo *et al.*, 2017), may also contribute to the varied priming intensity. Extending further studies to incorporate these alternative factors should further improve our understanding of potential determinants of the priming effect” (Page 13-14, line 274-286). Thanks for your understanding!

[Comment 70] Lastly, although I am aware that I did not review this version of the manuscript very favourably, I do wish to thank you for all the hard work that is obviously behind these data. I find the data conceptually very interesting, and I look forward to reading this manuscript in its revised form.

[Response] We are very grateful to the reviewer for the insightful comments on our manuscript! These comments guided us to have deeper thinking on related issues, and greatly improved the manuscript. Thank you!

Reviewer #2 (Remarks to the Author):

[Comment 1] Here the authors present an interesting study linking carbon dynamics and N availability in permafrost, and explore how these interactions may change over time since thaw. Further, they address a very important microbial component, demonstrating that microbial metabolism can be important in predictive models. The study is relevant for a number of reasons including (1) the obvious effects of climate change on carbon feedback cycles, (2) the increased awareness of including microbes into global models, (3) significant remaining questions regarding N availability (especially N mining) with soil respiration and finally (4) acknowledging the large differences in permafrost systems (alpine vs arctic). However, while there are many compelling points within this study there are a number of significant points that need to be addressed.

[Response] Thanks for the reviewer's positive comments.

[Comment 2]1. The sampling design is problematic: (L158 Field design) It can be argued that this is pseudo replication within single replicate plots of each time stage, instead of 5 replicates at each stage. Ideally, each time stage should have 3-4 separate thermos-erosion gullies. There needs to a strong defense of this experimental design –it is a critical point – especially if the results are to be used for broad/global statements about carbon dynamics, microbial response and the use of enzyme analyses.

[Response] Very good comment! We are sorry for the poor description about the sampling design in previous MS. Actually, four sites representing undisturbed grassland, early (1-year since collapse), middle- (10-year since collapse) and late-stage (16-year since collapse) of permafrost collapse were established in 2014 (Fig. R16). The three collapse sites were dispersed at intervals of about 100 m along the gully. In 2015, **five independent plots (~3 m × 5 m) were set up within a 10 m × 20 m area at each collapse stage as replicates to do soil sampling.** Within each plot, top 15 cm soil were collected from all the vegetated patches using a 3 cm diameter hand corer to reduced spatial heterogeneity (Fig. R16). On average, **about**

10~12 soil cores were collected inside each plot and were mixed in the field as one replicate for subsequent analyses. In other words, these 10~12 soil cores within each plot were the pseudo replications, but the five independent plots were the five real replications. We have clearly mentioned these points in the revised MS (Page 18, line 375-376; Page 19, line 395-404).

Figure R16. Landscapes of different collapse stages within a typical thermo-erosion gully on the Tibetan Plateau and the sampling schematic diagram in undisturbed grassland and inside the thermos-erosion gully. The years since permafrost collapse for 1, 10 and 16 years were named as early-, middle- and late-stage of permafrost collapse, respectively.

Despite of this, we do agree with the reviewer that more separate thermos-erosion gullies would be ideally. However, it is challenging to conduct soil sampling in several gullies due to the following practical constraints. **First, other thermo-erosion gullies in this area are too short to repeat the late-stage of permafrost collapse.**

To better understand the changes in the priming effect along the thaw sequence, the soil sampling was conducted along the longest gully (~ 250 m, first developed in the 1990s, and is approximately two decades old) in this area. Other gullies in this study area have developed less than 10 years and are still at the early or middle-stage of permafrost collapse. **Second, both the difficulty in field sampling and high cost of laboratory analysis ($\delta^{13}\text{C}$ analysis for both CO_2 and soil samples) prevented us from sampling multiple gullies.** As mentioned by Reviewer 3# “*There are many limitations when performing complex experiments at remote field sites, and this study is a valuable contribution*”, and by Reviewer 1# “*I am aware of only two studies addressing this issue thus far, in arctic permafrost soils (Wild 2014, 2016). Hence, the data presented in the manuscript provided by Chen et al. could be a welcome addition.*”. Although gullies with similar collapse stages could be found in faraway areas (>220 km), the inconvenience of traffic and the lack of manpower largely increase the difficulty of soil sampling far away. Moreover, the pretty high cost (~\$28/sample) of $\delta^{13}\text{C}$ analysis for both CO_2 and soil samples, also provide additional constraint for larger sampling size.

Actually, these difficulties and budget limits have also been the great challenges met by international researchers who focused on the thaw sequences in permafrost regions. Therefore, **the sampling method used in current study is also commonly used in previous studies. According to the statistics, more than 70% of the studies regarding the thaw gradient set their replicates within the sequence rather than used multiple sequences as replicates** (Table R3).

Table R3. Replicate setting in all the studies focus on thaw gradient or thaw sequence.

Replicate type	Permafrost collapse gradients	Reference
Within one sequence	Minimal- Moderate- Extensive thaw	(Schuur et al. , 2009)
Within one sequence	Minimal- Moderate- Extensive thaw	(Vogel et al. , 2009)
Within one sequence	Undisturbed- Recent- Historic active layer detachment	(Pautler et al. , 2010)

Within one sequence	Minimal- Moderate- Extensive thaw	(Lee et al. , 2010)
Within one sequence	Minimal- Moderate- Extensive thaw	(Lee et al. , 2011)
Within one sequence	Minimal- Moderate- Extensive thaw	(Hicks Pries et al. , 2011)
Within one sequence	Minimal- Moderate- Extensive thaw	(Trucco et al. , 2012)
Within one sequence	Minimal- Moderate- Extensive thaw	(Hicks Pries et al. , 2013)
Within one sequence	Tundra- Active slump- Stabilized slump	(Jensen et al. , 2014)
Within one sequence	Minimal- Moderate- Extensive thaw	(Deng et al. , 2015a)
Within one sequence	Minimal- Moderate- Extensive thaw	(Yuan et al. , 2018)
Different sequences	Recent- Intermediate- Old thaw slump	(Pizano et al. , 2014)
Different sequences	Bog- Fen	(Hodgkins et al. , 2014)
Different sequences	Forest- Drunken forest- Most- Bog	(Finger et al. , 2016)
Different sequences	Young- Intermediate- Old bog	(Jones et al. , 2017)

Among them, the gradient field site in the Eight Mile Lake study area (EML) is the most famous one in which a series studies focusing on the thaw gradient have been conducted (Schuur *et al.*, 2009, Vogel *et al.*, 2009, Hicks Pries *et al.*, 2011, Trucco *et al.*, 2012, Deng *et al.*, 2015a). The thaw gradient contains three sites named minimal, moderate, and extensive thaw representing different durations of permafrost thaw (Fig. R17). At each site, C flux was usually estimated at 6 locations spaced 8 m apart along a 40 m transect. In each location, two chamber bases were treated as one replicate, for $n = 6$ replicates per site (Schuur *et al.*, 2009, Vogel *et al.*, 2009, Trucco *et al.*, 2012); with regards to soil sampling, 6 soil cores each at extensive, moderate and minimal thaw sites were usually used as replicates (Hicks Pries *et al.*, 2011, Deng *et al.*, 2015a).

Figure R17. The gradient field site in the Eight Mile Lake study area (EML). Symbols represent the location of the extensive thaw site (black), the moderate thaw site adjacent to the borehole (grey), and the minimal thaw site (white). The schematic diagram represent how soil C flux was estimated in each thaw stage (Adapted from Schuur *et al.* (2009)).

Taken together, we think that the sampling design of five independent replicates at each collapse stage in the thermo-gully is acceptable for this study. Definitely, future studies with more replicated gullies should be conducted to better explore the C-N interaction after permafrost collapse. We have clearly stated this point in the revised MS (Page 14, line 286-294). Thanks for your understanding!

[Comment 3]2. The relationship between priming and N availability (and therefore N mining) is interesting and could be made a larger part of the manuscript. As of how it is really only mentioned in the discussion, but instead could be used in the introduction to really set up the discrepancy studies have observed and be presented as a hypothesis. On that point, figure 6 is interesting and a case could be made to having it moved up in the text (even the introduction) as the hypothesis for how this system works.

[Response] Very good suggestion! Following the reviewer's suggestion, **we have reorganized the Introduction session on the basis of the current discrepancy studies.** Specifically, we used the contradictory studies to illustrate the challenges in

understanding C-N dynamics, thereby highlighting to the need to reveal the underlying mechanism as follows: “Among these changes, the widespread increase in nitrogen (N) availability after permafrost collapse (Harms *et al.*, 2013, Finger *et al.*, 2016), driven by enhanced N mineralization (Finger *et al.*, 2016) and the additional N released from thawing permafrost (Keuper *et al.*, 2012), may play an important role in regulating the priming intensity, since both vegetation growth and topsoil microbial activity in arctic (Sistla *et al.*, 2012, Wild *et al.*, 2015) and alpine ecosystems (Song *et al.*, 2007, Gao *et al.*, 2016, Kou *et al.*, 2017) are N limited. Despite this recognition, the link between N availability and the priming effect in permafrost ecosystems remains ambiguous because previous studies have revealed a positive (Wild *et al.*, 2014), a negative (Hartley *et al.*, 2010, De Baets *et al.*, 2016), and no effect (Wild *et al.*, 2014, De Baets *et al.*, 2016) of N addition on priming intensity. The diverging views probably reflect the ecosystem differences in microbial C and N demands (Wild *et al.*, 2016), which are closely related to the microbial physiological responses. Thus, to better understand this discrepancy, it is urgently needed to reveal the underlying mechanisms governing this C-N interaction, especially the role of microorganisms” (Page 3-4, line 72-83).

In the next paragraph, we proposed an important but poorly studied microbial physiological property-microbial metabolism efficiency that might contribute to the negative linkage between N availability and priming effect. **Following the reviewer’s suggestion, we presented a hypothesis as follows: “Here, we hypothesize that elevated N availability after permafrost collapse inhibits the priming effect by increasing the microbial metabolic efficiency”** (Page 5, line 102-103). It should be noted that although we could hypothesize the linkage between the priming effect and CUE, but it is hard to know the detailed pathway underlying that linkage before the experiment. Due to this point, we prefer to keep the initial Figure 6 (conceptual scheme including the detailed pathway how N availability regulate the priming effect) in the original position Thanks for your understanding!

[**Comment 4**]3. *Too much weight is being put on the enzyme results, because they are not in line with other results. This would be more convincing if the dataset was broader. Further, these data are correlative, and a better explanation/discussion needs to be made on why these results are different from all other results. It cannot merely be that this site functions different from all others, and if it does why?*

[**Response**] Very good comment! Following the reviewer's comment, **we have shortened the enzyme part in both Results and Discussion session**, and only kept the changes in LAP in the main text. The changes in BG, POX and NAG were moved to the Supplementary Information because of the insignificant change after the glucose addition. **We also have added more discussion about the potential reason for the different enzyme results** as follows: "In previous studies, the significant correlation between SOM-derived CO₂ flux and enzyme activity was considered as the evidence of the role of enzyme activity in priming effect (Zhu & Cheng, 2011, Chen *et al.*, 2014, Zhu *et al.*, 2014). Similarly, a significant linkage between soil C release and enzyme activity was also observed in our study (Fig. S7d). However, unlike previous results obtained in low fertility soils (Zhu & Cheng, 2011, Nottingham *et al.*, 2012, Chen *et al.*, 2014), we found that N-acquiring enzyme (*i.e.*, LAP) rather than C-acquiring enzyme (*e.g.*, BG and POX) contributed to the enhanced SOM-derived CO₂ release after glucose addition (Fig. 4). **This discrepancy could be attributed to the relatively weaker microbial energy limitation in our swamp meadow soil than that in other soils** (indicative of higher soil organic C concentration: 18.6% vs. 0.94~2.8%) (Nottingham *et al.*, 2012, Zhu & Cheng, 2011, Chen *et al.*, 2014). **This relatively weak energy limitation in our study site could be relieved after glucose addition, which in turn aggravates the N limitation** (Fig. 2d, h; indicative of higher C:N imbalance after glucose addition) **and thus promotes the N-acquiring enzyme activity**. This assumption was also supported by the non-significant response of C-acquiring enzyme after labile C addition in a tundra soil characterized by high organic C content (48%) and low energy limitation (Sistla *et al.*, 2012). Additionally, given that the enzyme activity usually varies with time, different

sampling time during incubation may also partly contribute to the discrepancy among studies.” All these discussions regarding the role of enzyme activity in regulating the priming effect have been moved to the Supplementary Information (Page 2, line 24-43).

[Comment 5] Abstract: L27: ‘aggravate’ is a weird word choice to describe soil C.

[Response] We have rephrased it to “induce soil C loss” in the revised MS (Page 2, line 31).

[Comment 6] L33: as presented, this study does little to shed light on microbial mechanisms, rather creates more questions in the end. Perhaps highlight a more specific gap this study addresses.

[Response] The “microbial mechanism” in the original MS refers to the finding that **“the microbial metabolic efficiency regulates the magnitude of the priming effect in permafrost region”**. Despite that theoretical and empirical evidence have revealed the response of microbial metabolic efficiency to nutrient availability (Manzoni *et al.*, 2012, Mooshammer *et al.*, 2014a), **the role of microbial metabolic efficiency in regulating the priming effect under changing N condition has rarely been assessed**. This is the specific gap in the mechanisms underlying the C-N interaction. Following the reviewer’s suggestion, we have clearly stated this point by rephrasing the sentence as follows: “Particularly, the role of microbial metabolic efficiency in this C-N interaction is poorly understood” (Page 2, line 36-37).

[Comment 7] Introduction: Much of this section could be rewritten to clarify better the challenges in understanding C-N dynamics and how a better understanding of microbial activity is the answer. Many studies have already examined this, what new research does this study specifically add?

[Response] The innovation of our study is that we provide first experimental evidence that microbial metabolic efficiency regulates the magnitude of the priming effect in permafrost region. To better highlight this point, we have rewritten the whole

Introduction session to clarify the challenge in understanding C-N dynamics and associated microbial mechanisms as follows: “Despite this recognition, the link between N availability and the priming effect in permafrost ecosystems remains ambiguous because previous studies have revealed a positive (Wild *et al.*, 2014), a negative (Hartley *et al.*, 2010, De Baets *et al.*, 2016), and no effect (Wild *et al.*, 2014, De Baets *et al.*, 2016) of N addition on priming intensity. **The diverging views probably reflect the ecosystem differences in microbial C and N demands (Wild *et al.*, 2016), which are closely related to the microbial physiological responses. Thus, to better understand this discrepancy, it is urgently needed to reveal the underlying mechanisms governing this C-N interaction, especially the role of microorganisms.**

Microbial metabolic efficiency, the partitioning of C substrate between microbial biomass and carbon dioxide (CO₂) production, is a key microbial physiological property that determines the fate of soil organic C (SOC) (Manzoni *et al.*, 2012, Frey *et al.*, 2013). By definition, microorganisms with higher microbial metabolic efficiency allocate more C to microbial growth than to respiration (Mooshammer *et al.*, 2014a, Sinsabaugh *et al.*, 2016), resulting in a lower soil CO₂ flux (Allison *et al.*, 2010, Manzoni *et al.*, 2012, Riggs & Hobbie, 2016). Consequently, higher microbial metabolic efficiency promotes C stabilization in soils, while lower microbial metabolic efficiency favors C loss via microbial respiration (Manzoni *et al.*, 2012). Besides affecting C cycling process, microbial metabolic efficiency may also play an important role in governing C-N interaction. Adjustment in microbial metabolic efficiency is a common microbial physiological response to natural or experimental variations in N availability (Riggs & Hobbie, 2016). Microbial metabolic efficiency generally increases with N availability (Manzoni *et al.*, 2012, Mooshammer *et al.*, 2014a). This physiological adjustment can facilitate the microorganisms to cope with their N demand and maintain a balanced biomass C:N ratio (Mooshammer *et al.*, 2014b). **Therefore, shifts in microbial metabolic efficiency may also contribute to the varied priming effects along the N gradient after thermokarst formation.**

However, to date, it remains obscure that whether and how microbial metabolic efficiency regulates the priming effect under different N conditions” (Page 4-5, line 77-100).

[Comment 8] Background references should show breadth of study, and how there is quite a bit of variation and difficulty in predicting permafrost/C response to warming (Potential carbon emissions dominated by carbon dioxide from thawed permafrost soils, Nature Climate Change 6, 950–953, (2016), doi:10.1038/nclimate3054. AND “Predicted responses of arctic and alpine ecosystems to altered seasonality under climate change.”

[Response] Very good comment! Following the reviewer’s comment, we have reorganized the paragraph as follows: “A warming climate is predicted to induce massive carbon (C) release from thawing permafrost, triggering a positive C-climate feedback (Koven *et al.*, 2011, Schuur *et al.*, 2015). However, the magnitude of this feedback remains highly uncertain, with potential C release from the permafrost zone ranging from 37 to 174 Pg by 2100 (Schuur *et al.*, 2015). The uncertainty in permafrost C feedback is partly attributed to the fact that ecosystem C balance is regulated by intricate plant-microbial-soil interactions after permafrost thaw (Ernakovich *et al.*, 2014, Koven *et al.*, 2015, Schädel *et al.*, 2016)” (Page 3, line 53-58).

[Comment 9] L54: ‘may’ predicted?

[Response] We have deleted the word “may” in this sentence.

[Comment 10] L56: clarify direct and indirect

[Response] Combining the suggestions from this reviewer and the other two reviewers, this sentence has been deleted in the revised MS. We have reorganized the paragraph as follows: “The uncertainty in permafrost C feedback is partly attributed to the fact that ecosystem C balance is regulated by intricate plant-microbial-soil interactions after permafrost thaw (Ernakovich *et al.*, 2014, Koven *et al.*, 2015,

Schädel *et al.*, 2016). In these interactions, enhanced plant C fixation may mitigate part of soil C loss (Koven *et al.*, 2011), but it may also induce soil C loss by accelerating the turnover of native soil organic matter (SOM), the so-called ‘priming effect’ (Kuzuyakov *et al.*, 2000, Fontaine *et al.*, 2004, Hartley *et al.*, 2012, Cheng *et al.*, 2014)” (Page 3, line 56-61).

[Comment 11] L74: *relieve*

[Response] Done as suggested.

[Comment 12] L78-90: *The point needs to be made stronger how this study will address the gap. Why can't N addition experiments or N gradients be informative on their own? Clarify the challenge/propose the mechanism for why combining them in one study will be the solution?*

[Response] Very good suggestion! Following the reviewer’s suggestion, we have reorganized the paragraph to clarify the methodological advantages as follows: “To test this hypothesis, we used a thermokarst-induced natural N gradient on the Tibetan Plateau (Fig. S1) and a subsequent N addition experiment to explore changes in the priming effect and microbial metabolic efficiency under different N conditions. While gradient studies are valuable for exploring variations in the priming effect along a natural N gradient, they cannot explicitly rule out other confounding factors that co-vary with N along the thaw gradient (Pizano *et al.*, 2014, Abbott & Jones, 2015). In contrast, although N manipulation experiments can explicitly reveal the N effect (Hartley *et al.*, 2010, Wild *et al.*, 2014, De Baets *et al.*, 2016), the single high N concentration involved in previous experiments (Koranda *et al.*, 2013, De Baets *et al.*, 2016) prevents us from understanding the trajectory of priming intensity along the natural N gradient. The combination of these two approaches is thus expected to generate a comprehensive understanding on the role of microbial metabolic efficiency in regulating the priming effect under different N conditions” (Page 5, line 103-114).

[Comment 13] L113: But does it hold a lot of the world's carbon? Why is it an important ecosystem

[Response] The Tibetan alpine permafrost plays an important role in terrestrial C cycle for the following reasons: **First**, it is the largest alpine permafrost region in the world (Zhang *et al.*, 2008), storing about 15.3 Pg C in the top 3 m soil (Ding *et al.*, 2016). Particularly, the soil organic C density of swamp meadow is comparable to that across the northern circumpolar permafrost regions (65.0 vs. 58.2 kg C m⁻²) (Ding *et al.*, 2016). **Second**, the Tibetan Plateau has risen 0.5 °C per decade over the last 30 years, twice as fast as the global average (Ding *et al.*, 2017), making it as one of the most vulnerable ecosystems to climate change. **Third**, the rapid warming has induced widespread thermokarst landscapes across swamp meadow on the plateau (Mu *et al.*, 2016, Yang *et al.*, 2018), which may further result in substantial soil C loss in a relatively short time.

Accordingly, we have rephrased the paragraph as follows: “The Tibetan Plateau is the largest alpine permafrost region in the world (Zhang *et al.*, 2008), **storing approximately 15.3 Pg C in its top 3 m soil** (Ding *et al.*, 2016). During the past several decades, climate warming induced **widespread thermokarst across swamp meadow on the plateau** (Mu *et al.*, 2016, Yang *et al.*, 2018). In contrast to the ice-poor permafrost deposits in two other grassland types (alpine steppe and meadow) (Ding *et al.*, 2016), the permafrost under swamp meadow is characterized as ice-rich due to its poor drainage (Mu *et al.*, 2017, Yang *et al.*, 2018). Moreover, soils in swamp meadow exhibit the highest C content among the soils of various grassland types on the plateau (Ding *et al.*, 2016), **with a SOC density comparable to that of the high-latitude permafrost regions** (65.0 vs. 58.2 kg C m⁻²) (Ding *et al.*, 2016)” (Page 16, line 326-333).

[Comment 14] Methods: L140: typical for the area?

[Response] Typical for the Tibetan Plateau. To avoid the confusion, we have reorganized the sentence as: “The study site (~3848 m altitude, 37°28' N, 100°17' E) is located in the eastern tributary of the Altun-Qilian mountains range, which is one of the five typical permafrost zones on the Tibetan Plateau (Jin *et al.*, 2011)” (Page 16-17, line 344-346).

[Comment 15] L147/149: remove ‘typical’

[Response] Done as suggested.

[Comment 16] L147-156: The site description needs to be clarified. From this description it seems it was not ideal for the study, but I don't think that was intended.

[Response] Sorry for the confusion. After re-analyzing the plant survey data, we found that although the dominant species did not change along the thaw sequence, the cover of these plants exhibited substantial change. Accordingly, we have clarified how plant productivity, species composition and root distribution changed along the thaw sequence as follows: “The middle-stage site was set at a location that had experienced 10 years of permafrost collapse, where the vertical distance between the original soil surface and collapsed soil was approximately 80 cm. Both above- and belowground plant biomass increased at this stage (Fig. S14) and the cover of *K. tibetica* increased at the expense of that of *C. atrofusca*. These changes in plant production and community composition could induce shifts in the priming effect (Cheng *et al.*, 2014). Despite a shift in plant community composition with thaw time, root distributions did not reveal significant changes along the thaw sequence” (Page 18, line 382-387, Page 19, line 391-393).

[Comment 17] L161: how were these stages designated? Is 16 year the oldest possible? Is 1 year the youngest necessary to see effects? Explain more directly.

[Response] Regarding the sampling design involved in this study, four sites were established in 2014 based on the age and stages of permafrost collapse: the undisturbed grassland site, early-stage site, middle-stage site, and late-stage site. Of

them, the undisturbed grassland site is the control site, without permafrost collapse. The early-stage site (1 year since collapse) is the most recent collapse site, with the deepest trench to the undisturbed soil surface (~2 m). Permafrost collapse tears apart the grassland into numerous vegetated rafts, resulting in substantial soil exposure. **Substantial water drainage occurs for the vegetated patches at this stage, possibly altering the microbial activities and thereby inducing changes in soil C dynamics.** The middle-stage site was set at a location that had experienced 10 years of permafrost collapse, where the vertical distance between the original soil surface and collapsed soil was approximately 80 cm. Both above- and belowground plant biomass increased at this stage (Fig. S14) and the cover of *K. tibetica* increased at the expense of that of *C. atrofusca*. These changes in plant production and community composition could induce shifts in the priming effect (Cheng *et al.*, 2014). The late-stage site was located at a downstream position of the gully with 16 years of permafrost collapse. The surface inside the gully was almost level with undisturbed area. **Plant recolonizations had occurred in the original exposed area at this stage, indicating the collapse has entered the oldest stage of recovery** (Page 18-19, line 375-393).

Regarding the age range used in this study, we would like mention that, based on our results (Fig. 1e), the priming effect was significantly reduced even only after 1 year of collapse, demonstrating that thermokarst could induce significant change in soil C release in a relatively short time. Additionally, there were clear environmental gradients (especially for N availability) along the thaw gradient (from 1-year to 16-year of collapse), providing an ideal platform to examine the relationship between the priming effect and post-thaw N availability. We have clearly mentioned these points in the revised MS (Page 18-19, line 375-393).

[Comment 18] L165: how many cores per replicate were taken?

[Response] On average, about 10~12 soil cores were collected inside each plot and

were composited in the field as one replicate for subsequent analyses. We have clearly mentioned this point in the revised MS (Page 19, line 398-399).

[Comment 19] L251: add the reference

[Response] Done as suggested.

[Comment 20] Results L323: add 'stage'

[Response] Done as suggested.

[Comment 21] Discussion: L373-380: This does not clarify why THIS study shows a strong relationship between N and priming – clearly state what was observed and why this means there is a connection.

[Response] Following the reviewer's comments, we have clearly stated the main point and the potential ecological mechanisms by reorganizing the *Discussion* section as follows: "Our study also presented a negative linkage between N availability and priming effect. Such a pattern is consistent with the 'microbial N mining' hypothesis, which assumes that microorganisms use labile C to decompose recalcitrant SOM and thus acquire N (Craine *et al.*, 2007). This hypothesis suggests that higher demands of microorganisms for N result in their greater response to the addition of labile C. However, the synthesis of extracellular enzymes in the N mining process (*i.e.*, SOM decomposition) has a high energy cost (Schimel & Weintraub, 2003). Increased N availability induced by mineral N supply might thus switch the microbial preferred substrate from SOM to added labile N (Blagodatskaya *et al.*, 2007, Nottingham *et al.*, 2015), thereby constraining C release from SOM" (Page 11, line 225-232).

[Comment 22] L382: For this section – need to elucidate why differences between the ecosystems are important and what this means for our understanding at a global scale.

[Response] Following the reviewer's comment, we have added more implications of

the differences between the ecosystems as follows: “Notably, the higher priming susceptibility in Tibetan organic soil than in arctic organic soil highlights the variation in priming intensity among permafrost ecosystems. When this regional difference is not considered, the role of priming effect in regulating soil C dynamics at the global scale may be underestimated” (Page 10-11, line 219-223).

[Comment 23] Fig. 3 Colors should be different so it does not appear to be related to time stage

[Response] Very good suggestion. We have changed the color scheme.

Overall, we are very grateful to the reviewer for the insightful comments on our manuscript! These comments guided us to have deeper thinking on related issues, and greatly improved the manuscript. Thank you!

Reviewer #3 (Remarks to the Author):

[Comment 1] The authors describe a study focusing on priming in Tibetan alpine permafrost, and the importance of N availability in this context. They combine two laboratory incubation experiments to reach this aim, (1) inducing priming by glucose addition along a natural, thermokarst-induced N availability gradient, and (2) specifically testing the interaction of glucose and N availability by adding both in a factorial design to one of the soils along the gradient. The data suggest that Tibetan permafrost soils are susceptible to priming, that priming is strongest where N availability is low, and that this effect is likely not driven by changes in enzyme activities as often assumed, but by changes in microbial C use efficiency. I find the approach appropriate and the data conclusive.

[Response] Thanks for the reviewer's positive comments.

[Comment 2] However, I am wondering to what extent the observed patterns can be generalized. First, it is not clear whether the C input simulated in the experiments is realistic, both quantitatively (amount of C input) and qualitatively (chemical composition of C input, timing, etc.). The authors did not test the impact of actual plant C input, but simulate it in the laboratory, and the discussion needs to be more careful to account for this limitation (e.g., lines 391-398).

[Response] We are very grateful to the reviewer for the insightful comments on our manuscript! Following the reviewer's comments, we have clearly stated that the C addition rate in this study is 2.2 mg ¹³C g⁻¹ dry soil, which approximately equaled 1% of SOC, similar to previous priming studies (range: 0.5%~ 2%) (Hartly *et al.*, 2010, Qiao *et al.*, 2014, Wild *et al.*, 2014) (Page 20-21, line 427-430). Despite of this, we agree with the reviewer that the laboratory experiment did not test the impact of actual plant C input, which could induce uncertainties in predicting the magnitude of in-situ priming effect. Therefore, **we have rephrased the Discussion session and interpreted our results more carefully to prevent overstating the findings.**

Following the reviewer’s comments, we have also added a paragraph to discuss the potential uncertainties derived from the incubation experiment as follows:

“While our study provides the experimental evidence for the role of N availability in regulating the priming effect after permafrost thaw, some uncertainties still exist. First, given that the priming effect is determined through laboratory incubation without plants, biotic attributes regulating the rhizosphere priming in situ have not been considered. For example, the quantity (Cheng *et al.*, 2014, Huo *et al.*, 2017), quality (Wild *et al.*, 2014, Qiao *et al.*, 2016) and timing of the C inputs (Qiao *et al.*, 2014) could affect the priming intensity, but it is challenging to simulate these biotic impacts with substrate addition under laboratory conditions. In situ rhizosphere priming experiments are thus needed to better elucidate this important plant-soil interaction in permafrost ecosystems” (Page 13, line 274-281). Thanks for your understanding!

[Comment 3] Second, only one site is studied. There are of course many limitations when performing complex experiments at remote field sites, and this study is a valuable contribution. Still, it is impossible to tell if the same patterns would be observed at other, similar sites.

[Response] Very good comment! It is true that one study site might not represent the total Tibetan permafrost, nevertheless, we think that our results could still provide important clues for other upland thermokarsts across the Tibetan Plateau for the following reasons: **First, the study site is located in Tibetan permafrost zone of Altun-Qilian Mountains, which is one of the five typical permafrost zones on the Tibetan Plateau** (Jin *et al.*, 2011). In this region, typical thermokarst landscapes such as thermo-erosion gully is frequently found in swamp meadow due to its high ice content in permafrost layer (Mu *et al.*, 2016, 2017). Previous studies about the thermokarst in the Tibetan permafrost region were all conducted in this permafrost zone (Mu *et al.*, 2016, 2017). Moreover, due to the typical permafrost characteristics in this region, a research station named “Qilian Cryosphere Research Station” (<http://www.skics.ac.cn/qlsz/1324.html>) has been set up here by geocryologists and permafrost engineers for the long-term permafrost monitoring. We have clearly

mentioned these points in the revised MS (Page 16-17, line 344-346).

Second, soil C dynamics found in our study site is similar to those from another typical thermo-erosion gully on the Tibetan Plateau. To be specific, a significant decrease in soil organic C content due to enhanced mineralization, photodegradation, and lateral displacement has been found both in our study site (Liu *et al.*, 2018) and another thermokarst site across Tibetan alpine permafrost region (Mu *et al.*, 2016).

Third, the magnitude of the priming effect reported in this study fell within the range derived from another incubation experiment, which includes 30 sites along a 2200-km transect across the Tibetan alpine permafrost (Fig. R18) (Chen *et al.* unpublished data). Specifically, the relative priming effect of the study site ranged from 5.8~31.8%, which is comparable to that derived from the 30 sites across the whole Tibetan Plateau (3.8~45.1%).

Figure R18. Map of sampling sites across the Tibetan Plateau.

Last but not the least, **the C density and C:N ratio of the study site is comparable to the average of swamp meadow** (54.1 vs. 46.1 kg C m⁻³; 12.3 vs. 11.9) (Chen *et al.*, 2016b, Ding *et al.*, 2016). Moreover, **vertical distributions of root observed in this study site are also similar to those reported across the whole swamp meadow on the Tibetan Plateau**, with 78% and 81% of roots occurred in the top 30 cm soil, respectively (Figure R1) (Page 17, line 357-363).

Taken together, we think that our result is representative of other thermokarst sites across the Tibetan Plateau. Definitely, future studies with more thermo-erosion gullies are necessary to better explore the pattern of the priming effect after permafrost collapse. Nevertheless, **we have added this uncertainty in the Discussion session** as follows: “Third, this study focuses on one typical upland thermokarst site. Although the observed decrease in soil C content is similar to that found in another thermokarst site on the Tibetan Plateau (Mu *et al.*, 2016) and the observed magnitude of the priming effect (5.8~31.8%) is comparable to those obtained across the whole Tibetan Plateau (3.8~45.1%) (Chen *et al.* unpublished data), site-level observations may induce uncertainties in predicting the response of soil CO₂ release to increased N availability after permafrost thaw. Future studies with a greater number of thermo-erosion gullies and grassland types are necessary to better explore the C-N interactions after permafrost thaw” (Page 14, line 286-294). Thanks for your understanding!

[Comment 4] Third, there are many studies on the connection between C and N cycling in the context of priming. These studies give very different results, as the authors also outline in the introduction (lines 75-78). These findings suggest that the effect of N on priming depends on parameters we currently do not understand. I therefore ask for caution when generalizing the findings of this study in the discussion, and in particular when suggesting to incorporate them into Earth System Models.

[Response] Very good comment! Following the reviewer’s comments, we have deleted the sentences about the “*results of this work should be included in Earth system models*” and also have rephrased the sentences regarding the implication of this study and limited our discussion in the context of Tibetan upland thermokarst as follows: “First, the negative association between post-thaw N availability and the priming effect suggests that the release of nutrient from Tibetan upland thermokarst may decelerate soil C loss by reducing the SOM-derived CO₂ release. Combining with its positive effect on vegetation C sequestration (Koven *et al.*, 2015), increased

soil N availability after permafrost thaw is thus expected to regulate ecosystem C balance across Tibetan upland thermokarst regions” (Page 15, line 311-315).

[Comment 5] The authors further hint at systematic differences in C content, C vs. N limitation of microorganisms, and susceptibility to priming, between arctic and Tibetan alpine permafrost surface soils (e.g., lines 114-117). This is an interesting point, but no data or references are provided that support it. In fact, not even C contents from the soils used in this study are presented. I suggest adding these data and discussing how representative they are for Tibetan permafrost soils. I particularly recommend to pay attention to differences between mineral and peat soils, and between soil horizons.

[Response] Very good comment! **Following the reviewer’s comment, we have added a supplementary table (Table R1) including the key soil and microbial properties** (soil moisture, bulk density, soil organic C concentration, organic C density, total N concentration, N density, soil C:N ratio, DOC concentration, TDN concentration, clay and silt content, microbial biomass C:N ratio, stoichiometric imbalance) in Supplementary Information of the revised MS (Page 4, line 51-57). Based on these datasets, we found that the C density and C:N ratio of the study site is higher than the average of the whole Tibetan Plateau (C density: 54.1 vs. 10.9 kg C m⁻³; C:N ratio: 12.3 vs. 8.9) (Chen *et al.*, 2016b, Ding *et al.*, 2016) but comparable to the average of swamp meadow (54.1 vs. 46.1 kg C m⁻³; 12.3 vs. 11.9) (Chen *et al.*, 2016b, Ding *et al.*, 2016). Thus, our study site could represent the swamp meadow, a typical grassland type where thermokarst mainly develops across the Tibetan permafrost region. We have clearly mentioned these points in the revised MS (Page 17, line 357-360).

Following the reviewer’s comments, we have also compared the differences between soil horizons and found that the response in our organic soil was comparable to that of mineral soils (14~31%) (Wild *et al.*, 2014, Wild *et al.*, 2016), but much larger than that of organic soils in arctic ecosystems (-3~7%) (Hartley *et al.*,

2010, De Baets *et al.*, 2016, Wild *et al.*, 2016). Such a difference could be due to variations in microbial C and N limitation across soil horizons and ecosystems (Wild *et al.*, 2014, Wild *et al.*, 2016). The decrease in soil C content (21.4~48.6% vs. 3.0~20.1%) and C:N ratio (25.6~41.3 vs. 11.9~27.0) from organic soil to mineral soil (Hartley *et al.*, 2010, De Baets *et al.*, 2016, Wild *et al.*, 2016) in arctic tundra ecosystem illustrated a shift from predominant microbial N limitation to C limitation with increasing soil depth (Sistla *et al.*, 2012, Wild *et al.*, 2014, Wild *et al.*, 2016). In contrast, the soil C content and C:N ratio of organic soils in our study site were 18.7% and 12.3, respectively (Table R1). These differences indicate that microbial activity in Tibetan organic soil is still predominantly subject to C limitation (Chen *et al.*, 2016b), thereby contributing to its high susceptibility to labile C input. We have clearly mentioned these points in the revised MS (Page 10, line 206-219).

[Comment 6] I would also like to read more discussion that links the findings of this laboratory study to natural ecosystems. In arctic permafrost soils, plant rooting and consequently plant C input are quite shallow (Iversen et al., 2015, New Phytologist), and the shallow soil shows low susceptibility to priming (Wild et al., 2016, Scientific Reports). Do the data presented here imply a different pattern in Tibetan alpine permafrost? The authors seem to hint at that. I would like to read some discussion on the matter that considers typical plant rooting depths and indicators of C versus N limitation in the shallow soil in Tibetan permafrost, as well as changes in vegetation and rooting with thermokarst formation. I would also like to know to what extent the increase in N availability with thermokarst formation could be linked to decreased plant N uptake due to physical disturbance. I suppose that there is not a lot of literature on these topics, but they should be at least (carefully) addressed.

[Response] Very good comment! Our data do suggest a higher priming susceptibility of Tibetan organic soil (17.9%) than that of arctic organic soil (-3~7%). To address the reviewer's detailed comments, we have provided the following information:

✧ With regards to the comparison of alpine and arctic permafrost ecosystems, we have **added root distributions as well as the changes in vegetation and rooting**

with thermokarst formation in Methods session as follows: “More importantly, in contrast with the relatively shallow root distribution in arctic tundra ecosystems (Iversen *et al.*, 2015, Wild *et al.*, 2016, Wild *et al.*, 2018), approximately 81% of plant roots are distributed in the top 30 cm of the soil in the swamp meadow (Fig. S12a), less than in arctic tundra (96%). The deep root distribution may facilitate the potential regulation of the priming effect on deep soil C dynamics across Tibetan upland thermokarst regions” (Page 16, line 338-342). Our additional data revealed that both the above- and belowground biomass increased at the mid and late-stage. At the same time, there was a shift in plant community composition (increased cover of *K. tibetica* but decreased cover of *C. atrofusca*) with thaw time. Despite of these changes, the root distribution did not exhibit significant change along the thaw sequence (Page 18, line 385-386; Page 19, line 391-393).

✧ With regards to indicators of C versus N limitation in the shallow soil in Tibetan permafrost, we provided this kind of information as follows: “Compared to the pasture or grassland soils in non-permafrost region (Fontaine *et al.*, 2011, Zhu *et al.*, 2014), the microbial activity in Tibetan swamp meadow is less C limited due to the relatively higher C content. However, **compared to the arctic tundra organic soils, the microbial activity in Tibetan thermokarst region is still limited by C**, indicated by **relatively lower C content** (18.7% in our study site vs. 21.4~48.6 % in arctic tundra) **and soil C:N ratio** (12.3 in Tibetan swamp meadow vs. 25.6~41.3 in arctic tundra) (Hartley *et al.*, 2010, De Baets *et al.*, 2016, Wild *et al.*, 2016). Similarly, the negative net N mineralization rate in alpine meadow and wetland (Song *et al.*, 2007, Gao *et al.*, 2016), higher proportion of microbial N immobilization to total gross N mineralization in Tibetan alpine grasslands (1.6 ± 0.2) (Kou *et al.*, 2017) and the increase in soil CO₂ flux after low N supply (Deng *et al.*, 2015b) jointly suggested that **N availability could also constrain the function of microbial communities in this region.** Hence, **topsoil microbial activity in Tibetan alpine permafrost is assumed to be co-limited by C and N availability**” (Page 16, line 334-338).

✧ Based on the above-mentioned information, **we explained the different magnitude of priming effect between alpine and arctic permafrost ecosystems from C versus N limitation** as follows: “Such a difference could be due to variations in microbial C and N limitation across soil horizons and ecosystems (Wild *et al.*, 2014, Wild *et al.*, 2016). The decrease in soil C content (21.4~48.6% vs. 3.0~20.1%) and C:N ratio (25.6~41.3 vs. 11.9~27.0) from organic soil to mineral soil (Hartley *et al.*, 2010, De Baets *et al.*, 2016, Wild *et al.*, 2016) In arctic tundra ecosystem illustrated a shift from predominant microbial N limitation to C limitation with increasing soil depth (Sistla *et al.*, 2012, Wild *et al.*, 2014, Wild *et al.*, 2016). In contrast, the soil C content and C:N ratio of organic soils in our study site were 18.7% and 12.3, respectively (Table S1). These differences indicate that microbial activity in Tibetan organic soil is still predominantly subject to C limitation (Chen *et al.*, 2016b), thereby contributing to its high susceptibility to labile C input” (Page 10, line 211-219).

✧ With regard to the potential mechanisms of increased N availability in permafrost systems, we have explored the linkage between soil N availability and plant N pool. To do so, we used the simultaneous measurements of plant nutrient and soil TDN obtained in 2017 and found that soil N availability was not correlated to plant N uptake (Figure R19a). Moreover, the aboveground N pool increased significantly at the early stage of permafrost collapse and dropped back gradually with the thaw time (Figure R19b). These additional analyses suggest that increased soil N availability was not linked to decreased plant N uptake. Based on this fact, we think that the increase in N availability with thermokarst formation should be mainly due to the increased N mineralization as well as additional frozen N released from thawing soil (Keuper *et al.*, 2012, Finger *et al.*, 2016).

Figure R19. Relationship between aboveground nitrogen (N) pool and soil total dissolved N concentration (a), and the changes in aboveground plant N pool along the thaw sequence (b).

[Comment 7] Finally, although the language is understandable, there are many errors that should be fixed. I did not point them out in detail, but I recommend some serious language polishing.

[Response] Following the reviewer’s suggestion, we have asked Dr. Maggie C.Y. Lau from Princeton University and an English language editing service (*i.e.*, Springer Nature Author Services) for language check. Please see the certification at the end of this response letter. By doing so, we feel that the language has been improved a lot.

[Comment 8] Abstract Line 46: Change to “thawing permafrost”.

[Response] Done as suggested.

[Comment 9] Introduction Line 52-53: Both cited references are estimates of future losses. So “might induce” is more adequate. Also, change to “a strong C-climate feedback”.

[Response] Done as suggested.

[Comment 10] Line 55: Please elaborate in more detail by what mechanisms N availability could increase in permafrost systems with warming.

[Response] Following comments from the other two reviewers, we have reorganized the *Introduction* section and moved the sentence regarding the mechanisms of increased N availability after permafrost collapse to the next paragraph as follows: “Among these changes, the widespread increase in nitrogen (N) availability after permafrost collapse (Harms *et al.*, 2013, Finger *et al.*, 2016), driven by enhanced N mineralization (Finger *et al.*, 2016) and the additional N released from thawing permafrost (Keuper *et al.*, 2012), may play an important role in regulating the priming intensity” (Page 4-5, line 72-75).

[Comment 11] Line 57: “Increased plant-derived C input” is not correct here. I think it should be “increased plant C stocks” or “increase plant C fixation” or similar.

[Response] We have rephrased the sentences as follows: “In these interactions, enhanced plant C fixation may mitigate part of soil C loss (Koven *et al.*, 2011), but it may also induce soil C loss by accelerating the turnover of native soil organic matter (SOM), the so-called ‘priming effect’” (Page 4, line 58-61).

[Comment 12] Line 68: I agree that N availability is likely to be important for priming effects in permafrost ecosystems, but I do not think that this statement can be supported by previous studies. So I would either remove “in permafrost ecosystems” or weaken the statement.

[Response] Combining the suggestions from this reviewer and the other two reviewers, we have rewritten the paragraph as follows: “Thermokarst is an abrupt permafrost thaw process (ground surface collapse caused by thawing of ice-rich permafrost) that can dramatically impact soil and hydrologic properties (Pizano *et al.*, 2014, Abbott & Jones, 2015, Olefeldt *et al.*, 2016). Post-thaw changes in soil nutrient, moisture, texture and pH (Abbott *et al.*, 2015, Finger *et al.*, 2016) may influence SOM turnover by altering the priming intensity (Cheng *et al.*, 2014, Nottingham *et al.*, 2015,

Huo *et al.*, 2017). Among these changes, the widespread increase in nitrogen (N) availability after permafrost collapse (Harms *et al.*, 2013, Finger *et al.*, 2016), driven by enhanced N mineralization (Finger *et al.*, 2016) and the additional N released from thawing permafrost (Keuper *et al.*, 2012), may play an important role in regulating the priming intensity, since both vegetation growth and topsoil microbial activity in arctic (Sistla *et al.*, 2012, Wild *et al.*, 2015) and alpine ecosystems (Song *et al.*, 2007, Gao *et al.*, 2016, Kou *et al.*, 2017) are N limited” (Page 4-5, line 69-77).

[Comment 13] Lines 69-71: A range of studies has shown that plants can take up not only mineral N forms, but also some small organic N forms such as amino acids (e.g., Lærkedal Sorensen et al., 2008, Arctic, Antarctic, and Alpine Research; Nordin et al., 2004, Ecology; Schimel and Chapin, 1996, Ecology). Nitrogen mineralization is thus not a requirement for N availability. However, there is consensus on wide-spread N limitation of plants in permafrost systems. I suggest re-writing this argument. Also, ref. 12 seems wrong here.

[Response] We are sorry about the mix up of references when constructing the literature list using the Endnote in previous MS. We have rephrased the sentences as follows: “Among these changes, the widespread increase in nitrogen (N) availability after permafrost collapse (Harms *et al.*, 2013, Finger *et al.*, 2016), driven by enhanced N mineralization (Finger *et al.*, 2016) and the additional N released from thawing permafrost (Keuper *et al.*, 2012), may play an important role in regulating the priming intensity, since both vegetation growth and topsoil microbial activity in arctic (Sistla *et al.*, 2012, Wild *et al.*, 2015) and alpine ecosystems (Song *et al.*, 2007, Gao *et al.*, 2016, Kou *et al.*, 2017) are N limited” (Page 4-5, line 72-77).

[Comment 14] Line 72: See above – I think somewhere in the introduction the mechanisms behind the expected increase in N availability should be explained. See also Koven et al., 2015, PNAS.

[Response] As mentioned above, we have added one sentence to explain the mechanisms behind the expected increase in N availability as follows: “The

widespread increase in nitrogen (N) availability after permafrost collapse (Harms *et al.*, 2013, Finger *et al.*, 2016), driven by enhanced N mineralization (Finger *et al.*, 2016) and the additional N released from thawing permafrost (Keuper *et al.*, 2012), may play an important role in regulating the priming intensity” (Page 4-5, line 72-75).

[Comment 15] Line 76: Ref. 12 does not fit (I suppose some mix up when constructing the literature list; this extends through the whole manuscript). I also think the sentence would be clearer with some more text (e.g., “... with either positive or negative effects of increased N availability on the magnitude of priming.”).

[Response] Sorry about the mix up of references when constructing the literature list using the Endnote in previous MS. Following the reviewer’s comment, we have rephrased the sentences as follows: “Despite this recognition, the link between N availability and the priming effect in permafrost ecosystems remains ambiguous because previous studies have revealed a positive (Wild *et al.*, 2014), a negative (Hartley *et al.*, 2010, De Baets *et al.*, 2016), and no effect (Wild *et al.*, 2014, De Baets *et al.*, 2016) of N addition on priming intensity” (Page 5, line 77-79).

[Comment 16] Line 80: Ref. 16 does not test the effect of N on priming.

[Response] We are sorry about the mix up of references. Combining the suggestions from this reviewer and the other two reviewers, the paragraph has been reorganized and this sentence has been deleted in the revised MS (Page 5, line 77-79).

[Comment 17] Line 85: Ref. 11 does not fit here.

[Response] Combining the suggestions from this reviewer and the other two reviewers, the paragraph has been reorganized and this sentence has been deleted in the revised MS.

[Comment 18] Line 87: Ref. 17 is not about priming, I do not think it is a very good fit here.

[Response] Combining the suggestions from this reviewer and the other two

reviewers, the paragraph has been reorganized and this sentence has been deleted in the revised MS.

[Comment 19] Line 116: Ref. 12 is wrong (see above). See also Sistla et al., 2012, Soil Biology & Biochemistry, but also Melle et al., 2015, Soil Biology & Biochemistry.

[Response] We are sorry about the mix up of references. Following the reviewer's suggestions, we have added these references in the revised MS (Page 17, line 334-335). Notably, to improve the readability of the manuscript, the description about the Tibetan permafrost has been moved from the *Introduction* session to the *Methods* session in the revised MS.

[Comment 20] Lines 126-127: I suggest changing to “an index of microbial C use efficiency”.

[Response] Based on the comment from Reviewer 1# “while both the CUE and the qCO_2 are related to C use, they cannot directly be converted into each other”, we have changed it to “an index of microbial metabolic efficiency”. Thanks for your understanding!

[Comment 21] Material & Methods. Please provide some more information on the sampling site: Elevation, soil type, active layer depth, dominant vegetation. I also suggest adding a (supplementary) table with basic parameters measured in the sampled soils, in particular also showing how they changed along the thaw gradient. This table should at least include organic C content and C/N ratios.

[Response] Very good suggestion! Following the reviewer's suggestion, we have provided more information about the sampling site as follows: “The elevation of the study site is ~ 3848 m. The soil was classified in the Gelisol (Hugelius *et al.*, 2014), with average active layer depth of 0.86 m. The dominant grassland type is swamp meadow with dominant species of *Kobresia tibetica*, *K. royleana* and *Carex atrofusca*” (Page 17-18, line 344-357).

Following the reviewer's comment, we have also added a supplementary table (Table R1) including the key soil and microbial parameters (soil moisture, bulk density, soil organic C content, organic C density, total N content, N density, soil C:N ratio, DOC concentration, TDN concentration, clay and silt content, microbial biomass C:N ratio, stoichiometric imbalance) as well as their changes along the thaw sequence in the revised MS.

[Comment 22] Line 143: "Perennially wet" as in water logged? Were soils anoxic? How did water content change along the thaw gradient? Please provide data.

[Response] The soil in our study site is periodically (in July and August) subjected to water logged. Although we did not directly measure the redox potential and oxygen concentration of the soil, the continued CH₄ emission implies that the soils in the undisturbed site are anoxic (Yang *et al.*, 2018). A significant decrease in soil moisture was found after permafrost collapse, resulting in a well drainage condition for the vegetated patches (Page 8, line 139-140). We have clearly mentioned these points in *Methods* session of the revised MS and also added the soil water content data in the supplementary table (Table R1, Page 18, line 380-382).

[Comment 23] Lines 149-151: Please elaborate on dominant vegetation and changes along the thaw gradient. The priming effect occurring under natural conditions might differ between vegetation types (e.g., linked to rooting patterns or mycorrhizal association), so details on the vegetation are valuable for setting the findings of this study in context.

[Response] Following the reviewer's suggestion, we have listed related vegetation information in the revised MS as follows: "The grassland type of the study site is swamp meadow, with dominant species of *Kobresia tibetica*, *K. royleana* and *Carex atrofusca*. Moreover, there was a shift in plant composition, *i.e.*, increased cover of *K. tibetica* but decreased cover of *C. atrofusca* with thaw time. The rooting depth of the study site is approximately 70 cm, with 78% of plant roots being distributed in the top 30 cm of soil (Fig. R1). The root distribution did not reveal significant changes along

the thaw sequence” (Page 18, line 347-350, line 385-386; Page 19, line 391-393).

[Comment 24] Line 173: Where the soils homogenized after adding the label?

[Response] The soils were homogeneously mixed with the glucose solution in 400 ml amber bottle. We have clearly stated this point in the revised MS (Page 21, line 433).

[Comment 25] Line 184: What is the soil temperature in the field?

[Response] Soil temperature during the growing seasons ranged from 0.1~18 °C, with an average of 7.8 °C. We have added this point in the revised MS as follows: “The incubation temperature was set to 15 °C, representing the upper end of the soil temperature during the growing seasons at the study site (0.1~18 °C)” (Page 21, line 434-435).

[Comment 26] Line 210: I do not understand. According to line 172, glucose was enriched to 99 atom% ¹³C – this would equal much more than 1752‰ (but should in fact not be expressed in delta values).

[Response] Sorry for the confusion. The glucose solution we added to the soil is 3 atom% ¹³C, which was obtained by mixing ¹³C-lable glucose (Sigma-Aldrich, uniformly labeled, 99 at% ¹³C) with unlabeled glucose before application. We have clearly mentioned this point in the revised MS (Page 20, line 425-427).

[Comment 27] Line 236: Do you mean TOC (total organic carbon) or DOC (dissolved organic carbon)? To match with TDN, I suppose the latter. There are some discrepancies between text and figures throughout the manuscript, please fix that.

[Response] We refer to DOC here. Following the reviewer’s suggestion, we have changed all the TOC to DOC throughout the revised MS.

[Comment 28] Line 243: Delete “thermokarst-induced”.

[Response] Done as suggested.

[Comment 29] Line 245: Change to “an index of microbial C use efficiency”.

[Response] Given that Reviewer 1# had mentioned that “*while both the CUE and the qCO_2 are related to C use, they cannot directly be converted into each other*”, we have changed it to “an index of microbial metabolic efficiency”. Thanks for your understanding!

[Comment 30] Lines 247-251: Please be more specific when describing the enzyme target compounds.

[Response] Done as suggested.

[Comment 31] Lines 251: Please add a brief description of the measurement principle (addition of fluorescence-labelled substrates). What was used as standard, and were standards added to soil slurries to account for quenching? I am not sure what you mean with “the usual single-point correction”; the protocols I am familiar with include a multi-point calibration curve. Please also add substrate concentrations.

[Response] Very good suggestion! Following the reviewer’s suggestion, we have added more descriptions about the measurement principle and substrate concentrations in the revised MS (Page 24-25, line 507-518). The activities of BG, LAP and NAG were assayed following the method described by German *et al.* (2011) using fluorometric techniques. In our experiment, standards were added to account for quenching. Specifically, 4-methylumbelliferone (MUB) was added as standard for BG and NAG; 7-amino-4-methylcoumarin (AMC) was used for LAP.

Regarding the single-point correction method, it is a method that does not include a multi-point calibration curve. For example, the protocol from Allison Lab (<http://allison.bio.uci.edu/protocols/fluorimetricenzymeprotocol.pdf>) belongs to this single-point correction method, and it is one of the most frequently used method in previous studies (Saiya-Cork *et al.*, 2002, Allison & Vitousek, 2004, Allison & Vitousek, 2005).

[Comment 32] Line 260: Change to “L-dihydroxy-phenylalanine”. What was the concentration of DOPA? Why was EDTA added?

[Response] Done as suggested. The concentration of DOPA is 25 Mm. EDTA was added to eliminate the potential effects of reducing metal ions (e.g., Fe²⁺) (Allison *et al.*, 2008). We have clearly mentioned these points in the revised MS.

[Comment 33] Line 275: How did you measure the 13C content of fumigated and unfumigated extracts?

[Response] The fumigated and unfumigated extracts were first freeze-dried into powder. After that, the ¹³C abundance of fumigated and unfumigated samples were measured with isotopic ratio mass spectrometry (IRMS 20-22, SerCon, Crewe, UK). Nevertheless, we would like to mention that we have deleted this variable in the revised MS, based on the comment from Reviewer 1# which stated that there was a potential non-independence between the priming effect and SOM-derived *q*CO₂.

[Comment 34] Results Lines 347-356: I find it difficult to distinguish between the two parts of the study (addition of glucose to thaw sequence and addition of glucose, N or both to the 16-year samples). It is additionally confusing that glucose had no significant effect on BG activity in the thaw sequence part, but in the 16-year part.

[Response] Very good comments! To avoid the confusion, we have reorganized the paragraph and clearly distinguished the results of the two experiments by using the phrase “**In the gradient experiment**” and “**the subsequent N addition experiment**” in the revised MS (Page 9-10, line 175, 193-194).

With regards to the results of BG in the two experiments, we would like to first explain how to understand the interaction effect in the two-way ANOVA. **When the interaction effect is not significant in a two-way ANOVA,** it means that there were no combined effects of the two factors, and thus **we could interpret the result only based on the main effects** (Pagano & Gauvreau, 2000). In contrast, **when the interaction effect is significant, it means that the impact of one factor depends on**

the level of the other factor (Pagano & Gauvreau, 2000). Thus, when the interaction effect is present, **the interpretation of the main effects is incomplete**.

As for the Fig. 4a in the original MS, there were no significant glucose (G) × thaw sequence (time) interaction on BG ($P = 0.27$) in the first gradient experiment, indicating that the effects of glucose on these enzyme activities was independent from thaw time. Thus, we could come to a conclusion just based on the main effect: **glucose addition had no significant effects on BG** ($P = 0.96$). In contrast, there was a significant glucose (G) × nitrogen (N) interaction on BG ($P = 0.03$; Fig. 4d) in N addition experiment, indicating that the effect of glucose on BG was dependent on the availability of N. Specifically, **glucose addition only decreased the activity of BG when N was simultaneously added. This is consistent with the first result in which glucose had no effect on BG without N addition.**

We would like to mention that, when replying the first reviewer's comment, we noticed that the concentration of added N in the previous N addition experiment (3000 mg N kg⁻¹ soil) was far beyond the range of natural soil TDN concentration (47.5~89.1 mg N kg⁻¹ soil) along the thaw gradient. **To resolve this issue, we conducted a new N addition experiment including ¹³C-labeled glucose with two levels of N addition in the form of NH₄NO₃.** This resulted in the following treatments: water (control), glucose (G), low N, G + low N, high N and G + high N. To simulate the thermokarst-induced increase in soil N availability, **the TDN concentrations in both N treatments were controlled within the natural range of the difference in soil TDN concentration (20.1~50.8 mg N kg⁻¹ soil) between thaw stages** (Yang *et al.*, unpublished data). Correspondingly, the rate of NH₄NO₃ addition in low and high N treatment were equivalent to 20 and 50 mg N kg⁻¹ soil, respectively. **In this new N addition experiment, we still found a significant interaction between glucose and nitrogen interaction on BG.** Specifically, **glucose addition would only decrease the BG activity when N was added at high concentration** (Fig. R20a). This is also consistent with the first result in which glucose had no effect

on BG without N addition. **Nevertheless, we would like to mention that it is normal to find different effects of “glucose” treatment between the two experiments.** Because in the gradient experiment, not only N availability but also **many other soil factors varied with thaw time** (Table R1). These confounding factors together with N availability co-affected the glucose effect on the response variables across the thaw sequence. However, in the N addition experiment, only N availability changed under different glucose addition treatment.

We would also like to mention that we moved the results of enzyme activities in the N addition experiment to the Supplementary Information, since our first gradient experiment demonstrated that **microbial metabolic efficiency rather than enzyme activity accounted for variations of priming effect along the thaw sequence.** We also discussed the role of enzyme in regulating the priming effect in Supplementary Information (Page 2, line 24-49).

Figure R20. Activity of β -1,4-glucosidase (BG) (a), phenol oxidase (POX) (b), leucine aminopeptidase (LAP) (c) and β -1,4-N-acetylglucosaminidase (NAG) (d)

under different levels of glucose and N addition. Data represent means + SE (standard error).

[Comment 35] Discussion Lines 377-379: See my general comments. Please present more information on the soils studied here to permit comparisons with organic or mineral horizons of arctic permafrost soils. With the data provided, I cannot tell if this comparison is justified.

[Response] As mentioned above, we have added a supplementary table (Table R1) including the key soil and microbial parameters (soil moisture, bulk density, soil organic C content, organic C density, total N content, N density, soil C:N ratio, DOC concentration, TDN concentration, clay and silt content, microbial biomass C:N ratio, stoichiometric imbalance) in Supplementary Information of the revised MS (Page 4, line 51-57).

[Comment 36] Line 384: I suggest comparing C contents per dry soil, not square meter. Carbon content per square meter also depends on bulk density which is not relevant here.

[Response] Following the reviewer's suggestion, we added data of soil C contents in the revised MS, and have also added the data of soil C:N ratio to better reflect the microbial C and N limitation (Wild *et al.*, 2015). Accordingly, we have revised the comparison between the two permafrost regions as follows: "Such a difference could be due to variations in microbial C and N limitation across soil horizons and ecosystems (Wild *et al.*, 2014, Wild *et al.*, 2016). The decrease in soil C content (21.4~48.6% vs. 3.0~20.1%) and C:N ratio (25.6~41.3 vs. 11.9~27.0) from organic soil to mineral soil (Hartley *et al.*, 2010, De Baets *et al.*, 2016, Wild *et al.*, 2016) In arctic tundra ecosystem illustrated a shift from predominant microbial N limitation to C limitation with increasing soil depth (Sistla *et al.*, 2012, Wild *et al.*, 2014, Wild *et al.*, 2016). In contrast, the soil C content and C:N ratio of organic soils in our study site were 18.7% and 12.3, respectively (Table S1). These differences indicate that microbial activity in Tibetan organic soil is still predominantly subject to C limitation

(Chen *et al.*, 2016b), thereby contributing to its high susceptibility to labile C input” (Page 11, line 211-219).

[Comment 37] Line 422-426: I think this discrepancy might be due to variability within arctic soils, e.g., changes in N limitation with soil depth or season (e.g., Melle et al., 2015, Soil Biology & Biochemistry).

[Response] Very good point! We have added this point in the revised MS (Page 12-13, line 235-243).

[Comment 38] Figure 4: There is a typo (nitorgen) in panel (d).

[Response] Sorry for the carelessness. We have corrected it in the revised MS.

Overall, we are very grateful to the reviewer for the insightful comments on our manuscript! These comments guided us to have deeper thinking on related issues, and greatly improved the manuscript. Thank you!

References

- Abbott BW, Jones JB (2015) Permafrost collapse alters soil carbon stocks, respiration, CH₄, and N₂O in upland tundra. *Global Change Biology*, **21**, 4570-4587.
- Abbott BW, Jones JB, Godsey SE, Larouche JR, Bowden WB (2015) Patterns and persistence of hydrologic carbon and nutrient export from collapsing upland permafrost. *Biogeosciences*, **12**, 3725-3740.
- Allison SD, Czimczik CI, Treseder KK (2008) Microbial activity and soil respiration under nitrogen addition in Alaskan boreal forest. *Global Change Biology*, **14**, 1156-1168.
- Allison SD, Vitousek PM (2004) Extracellular enzyme activities and carbon chemistry as drivers of tropical plant litter decomposition. *Biotropica*, **36**, 285-296.
- Allison SD, Vitousek PM (2005) Responses of extracellular enzymes to simple and complex nutrient inputs. *Soil Biology & Biochemistry*, **37**, 937-944.
- Allison SD, Wallenstein MD, Bradford MA (2010) Soil-carbon response to warming dependent on microbial physiology. *Nature Geoscience*, **3**, 336-340.
- Blagodatskaya EV, Blagodatsky SA, Anderson TH, Kuzyakov Y (2007) Priming effects in Chernozem induced by glucose and N in relation to microbial growth strategies. *Applied Soil Ecology*, **37**, 95-105.
- Bossio DA, Scow KM (1998) Impacts of carbon and flooding on soil microbial communities: phospholipid fatty acid profiles and substrate utilization patterns. *Microbial Ecology*, **35**, 265-278.

- Carrillo Y, Dijkstra FA, Pendall E, Lecain D, Tucker C (2014) Plant rhizosphere influence on microbial C metabolism: the role of elevated CO₂, N availability and root stoichiometry. *Biogeochemistry*, **117**, 229-240.
- Chen LY, Liang JY, Qin SQ *et al.* (2016a) Determinants of carbon release from the active layer and permafrost deposits on the Tibetan Plateau. *Nature Communications*, **7**.
- Chen R, Senbayram M, Blagodatsky S *et al.* (2014) Soil C and N availability determine the priming effect: microbial N mining and stoichiometric decomposition theories. *Global Change Biology*, **20**, 2356-2367.
- Chen Y, Chen L, Peng Y *et al.* (2016b) Linking microbial C:N:P stoichiometry to microbial community and abiotic factors along a 3500-km grassland transect on the Tibetan Plateau. *Global Ecology and Biogeography*, **25**, 1416-1427.
- Cheng W, Parton WJ, Gonzalez-Meler MA *et al.* (2014) Synthesis and modeling perspectives of rhizosphere priming. *New Phytologist*, **201**, 31-44.
- Craine JM, Morrow C, Fierer N (2007) Microbial nitrogen limitation increases decomposition. *Ecology*, **88**, 2105-2113.
- De Baets S, Van De Weg MJ, Lewis R *et al.* (2016) Investigating the controls on soil organic matter decomposition in tussock tundra soil and permafrost after fire. *Soil Biology & Biochemistry*, **99**, 108-116.
- Deng J, Gu Y, Zhang J *et al.* (2015a) Shifts of tundra bacterial and archaeal communities along a permafrost thaw gradient in Alaska. *Molecular Ecology*, **24**, 222-234.
- Deng Z, Gao J, Zhou Y, Gao J (2015b) Effect of N deposition on CO₂ emission during freezing-thawing incubation period of peat soil. *Chinese Journal of Soil Science*, **46**, 962-966.
- Ding J, Chen L, Ji C *et al.* (2017) Decadal soil carbon accumulation across Tibetan permafrost regions. *Nature Geoscience*, **10**, 420-424.
- Ding JZ, Li F, Yang GB *et al.* (2016) The permafrost carbon inventory on the Tibetan Plateau: a new evaluation using deep sediment cores. *Global Change Biology*, **22**, 2688-2701.
- Ernakovich JG, Hopping KA, Berdanier AB, Simpson RT, Kachergis EJ, Steltzer H, Wallenstein MD (2014) Predicted responses of arctic and alpine ecosystems to altered seasonality under climate change. *Global Change Biology*, **20**, 3256-3269.
- Fang HJ, Cheng SL, Yu GR *et al.* (2012) Responses of CO₂ efflux from an alpine meadow soil on the Qinghai Tibetan Plateau to multi-form and low-level N addition. *Plant and Soil*, **351**, 177-190.
- Finger RA, Turetsky MR, Kielland K, Ruess RW, Mack MC, Euskirchen ES, Wurzbürger N (2016) Effects of permafrost thaw on nitrogen availability and plant-soil interactions in a boreal Alaskan lowland. *Journal of Ecology*, **104**, 1542-1554.
- Fontaine S, Bardoux G, Abbadie L, Mariotti A (2004) Carbon input to soil may decrease soil carbon content. *Ecology Letters*, **7**, 314-320.
- Fontaine S, Henault C, Aamor A *et al.* (2011) Fungi mediate long term sequestration of carbon and nitrogen in soil through their priming effect. *Soil Biology & Biochemistry*, **43**, 86-96.
- Frey SD, Lee J, Melillo JM, Six J (2013) The temperature response of soil microbial efficiency and its feedback to climate. *Nature Climate Change*, **3**, 395-398.
- Gale MR, Grigal DF (1987) Vertical root distributions of northern tree species in relation to

- successional status. *Canadian Journal of Forest Research-Revue Canadienne De Recherche Forestiere*, **17**, 829-834.
- Gao J, Feng J, Zhang X, Yu F-H, Xu X, Kuzyakov Y (2016) Drying-rewetting cycles alter carbon and nitrogen mineralization in litter-amended alpine wetland soil. *CATENA*, **145**, 285-290.
- German DP, Weintraub MN, Grandy AS, Lauber CL, Rinkes ZL, Allison SD (2011) Optimization of hydrolytic and oxidative enzyme methods for ecosystem studies. *Soil Biology and Biochemistry*, **43**, 1387-1397.
- Harms TK, Abbott BW, Jones JB (2013) Thermo-erosion gullies increase nitrogen available for hydrologic export. *Biogeochemistry*, **117**, 299-311.
- Hartley IP, Garnett MH, Sommerkorn M *et al.* (2012) A potential loss of carbon associated with greater plant growth in the European Arctic. *Nature Climate Change*, **2**, 875-879.
- Hartley IP, Hopkins DW, Sommerkorn M, Wookey PA (2010) The response of organic matter mineralisation to nutrient and substrate additions in sub-arctic soils. *Soil Biology & Biochemistry*, **42**, 92-100.
- Hicks Pries CE, Schuur EA, Crummer KG (2013) Thawing permafrost increases old soil and autotrophic respiration in tundra: partitioning ecosystem respiration using delta(13) C and (14) C. *Global Change Biology*, **19**, 649-661.
- Hicks Pries CE, Schuur EaG, Crummer KG (2011) Holocene carbon stocks and carbon accumulation rates altered in soils undergoing permafrost thaw. *Ecosystems*, **15**, 162-173.
- Hodgkins SB, Tfaily MM, Mccalley CK *et al.* (2014) Changes in peat chemistry associated with permafrost thaw increase greenhouse gas production. *Proceedings of the National Academy of Sciences*, **111**, 5819-5824.
- Hugelius G, Strauss J, Zubrzycki S *et al.* (2014) Estimated stocks of circumpolar permafrost carbon with quantified uncertainty ranges and identified data gaps. *Biogeosciences*, **11**, 6573-6593.
- Huo C, Luo Y, Cheng W (2017) Rhizosphere priming effect: A meta-analysis. *Soil Biology & Biochemistry*, **111**, 78-84.
- Iversen CM, Sloan VL, Sullivan PF *et al.* (2015) The unseen iceberg: plant roots in arctic tundra. *New Phytologist*, **205**, 34-58.
- Jensen AE, Lohse KA, Crosby BT, Mora CI (2014) Variations in soil carbon dioxide efflux across a thaw slump chronosequence in northwestern Alaska. *Environmental Research Letters*, **9**, 025001.
- Jin H, Luo D, Wang S, Lü L, Wu J (2011) Spatiotemporal variability of permafrost degradation on the Qinghai-Tibet Plateau. *Sciences in Cold and Arid Regions*, **3**, 0281-0305.
- Jones MC, Harden J, O'donnell J, Manies K, Jorgenson T, Treat C, Ewing S (2017) Rapid carbon loss and slow recovery following permafrost thaw in boreal peatlands. *Global Change Biology*, **23**, 1109-1127.
- Keuper F, Van Bodegom PM, Dorrepaal E, Weedon JT, Van Hal J, Van Logtestijn RSP, Aerts R (2012) A frozen feast: thawing permafrost increases plant-available nitrogen in subarctic peatlands. *Global Change Biology*, **18**, 1998-2007.
- Koranda M, Kaiser C, Fuchslueger L, Kitzler B, Sessitsch A, Zechmeister-Boltenstern S,

- Richter A (2013) Seasonal variation in functional properties of microbial communities in beech forest soil. *Soil Biology & Biochemistry*, **60**, 95-104.
- Kou D, Peng Y, Wang G *et al.* (2017) Diverse responses of belowground internal nitrogen cycling to increasing aridity. *Soil Biology & Biochemistry*, **116**, 189-192.
- Koven CD, Lawrence DM, Riley WJ (2015) Permafrost carbon-climate feedback is sensitive to deep soil carbon decomposability but not deep soil nitrogen dynamics. *Proceedings of the National Academy of Sciences of the United States of America*, **112**, 3752-3757.
- Koven CD, Ringeval B, Friedlingstein P *et al.* (2011) Permafrost carbon-climate feedbacks accelerate global warming. *Proceedings of the National Academy of Sciences of the United States of America*, **108**, 14769-14774.
- Kuzyakov Y, Friedel JK, Stahr K (2000) Review of mechanisms and quantification of priming effects. *Soil Biology and Biochemistry*, **32**, 1485-1498.
- Lee H, Schuur EaG, Vogel JG (2010) Soil CO₂ production in upland tundra where permafrost is thawing. *Journal of Geophysical Research-Biogeosciences*, **115**.
- Lee H, Schuur EaG, Vogel JG, Lavoie M, Bhadra D, Staudhammer CL (2011) A spatially explicit analysis to extrapolate carbon fluxes in upland tundra where permafrost is thawing. *Global Change Biology*, **17**, 1379-1393.
- Liu F, Chen L, Zhang B, Wang G, Qin S, Yang Y (2018) Ultraviolet radiation rather than inorganic nitrogen increases dissolved organic carbon biodegradability in a typical thermo-erosion gully on the Tibetan Plateau. *Science of the Total Environment*, **627**, 1276-1284.
- Ma K, Zhang Y, Tang S, Liu J (2016) Spatial distribution of soil organic carbon in the Zoige alpine wetland, northeastern Qinghai-Tibet Plateau. *CATENA*, **144**, 102-108.
- Manzoni S, Taylor P, Richter A, Porporato A, Grenn GI (2012) Environmental and stoichiometric controls on microbial carbon-use efficiency in soils. *New Phytologist*, **196**, 79-91.
- Mooshammer M, Wanek W, Hammerle I *et al.* (2014a) Adjustment of microbial nitrogen use efficiency to carbon:nitrogen imbalances regulates soil nitrogen cycling. *Nature Communications*, **5**, 3694.
- Mooshammer M, Wanek W, Zechmeister-Boltenstern S, Richter A (2014b) Stoichiometric imbalances between terrestrial decomposer communities and their resources: mechanisms and implications of microbial adaptations to their resources. *Frontiers in Microbiology*, **5**, 22.
- Mu CC, Abbott BW, Zhao Q *et al.* (2017) Permafrost collapse shifts alpine tundra to a carbon source but reduces N₂O and CH₄ release on the northern Qinghai-Tibetan Plateau. *Geophysical Research Letters*, **44**, 8945-8952.
- Mu CC, Zhang T, Zhang X *et al.* (2016) Carbon loss and chemical changes from permafrost collapse in the northern Tibetan Plateau. *Journal of Geophysical Research: Biogeosciences*, **121**, 1781-1791.
- Nottingham AT, Turner BL, Chamberlain PM, Stott AW, Tanner EVJ (2012) Priming and microbial nutrient limitation in lowland tropical forest soils of contrasting fertility. *Biogeochemistry*, **111**, 219-237.
- Nottingham AT, Turner BL, Stott AW, Tanner EVJ (2015) Nitrogen and phosphorus constrain

- labile and stable carbon turnover in lowland tropical forest soils. *Soil Biology & Biochemistry*, **80**, 26-33.
- Olefeldt D, Goswami S, Grosse G *et al.* (2016) Circumpolar distribution and carbon storage of thermokarst landscapes. *Nature Communications*, **7**.
- Pagano M. & Gauvreau K. (2000). *Principles of Biostatistics* (2nd ed), Pacific Grove, Duxbury Press.
- Pautler BG, Simpson AJ, McNally DJ, Lamoureux SF, Simpson MJ (2010) Arctic permafrost active layer detachments stimulate microbial activity and degradation of soil organic matter. *Environmental Science & Technology*, **44**, 4076-4082.
- Peng Y, Li F, Zhou G *et al.* (2017) Linkages of plant stoichiometry to ecosystem production and carbon fluxes with increasing nitrogen inputs in an alpine steppe. *Global Change Biology*, **23**, 5249-5259.
- Pizano C, Barón AF, Schuur EaG, Crummer KG, Mack MC (2014) Effects of thermo-erosional disturbance on surface soil carbon and nitrogen dynamics in upland arctic tundra. *Environmental Research Letters*, **9**, 075006.
- Qiao N, Schaefer D, Blagodatskaya E, Zou X, Xu X, Kuzyakov Y (2014) Labile carbon retention compensates for CO₂ released by priming in forest soils. *Global Change Biology*, **20**, 1943-1954.
- Qiao N, Xu X, Hu Y, Blagodatskaya E, Liu Y, Schaefer D, Kuzyakov Y (2016) Carbon and nitrogen additions induce distinct priming effects along an organic-matter decay continuum. *Scientific Reports*, **6**, 19865.
- Riggs CE, Hobbie SE (2016) Mechanisms driving the soil organic matter decomposition response to nitrogen enrichment in grassland soils. *Soil Biology & Biochemistry*, **99**, 54-65.
- Saiya-Cork KR, Sinsabaugh RL, Zak DR (2002) The effects of long term nitrogen deposition on extracellular enzyme activity in an *Acer saccharum* forest soil. *Soil Biology and Biochemistry*, **34**, 1309-1315.
- Schädel C, Bader MKF, Schuur EaG *et al.* (2016) Potential carbon emissions dominated by carbon dioxide from thawed permafrost soils. *Nature Climate Change*, **6**, 950-953.
- Schimel JP, Weintraub MN (2003) The implications of exoenzyme activity on microbial carbon and nitrogen limitation in soil: a theoretical model. *Soil Biology & Biochemistry*, **35**, 549-563.
- Schuur EA, Vogel JG, Crummer KG, Lee H, Sickman JO, Osterkamp TE (2009) The effect of permafrost thaw on old carbon release and net carbon exchange from tundra. *Nature*, **459**, 556-559.
- Schuur EaG, McGuire AD, Schadel C *et al.* (2015) Climate change and the permafrost carbon feedback. *Nature*, **520**, 171-179.
- Sinsabaugh RL, Manzoni S, Moorhead DL, Richter A (2013) Carbon use efficiency of microbial communities: stoichiometry, methodology and modelling. *Ecology Letters*, **16**, 930-939.
- Sinsabaugh RL, Turner BL, Talbot JM *et al.* (2016) Stoichiometry of microbial carbon use efficiency in soils. *Ecological Monographs*, **86**, 172-189.
- Sistla SA, Asao S, Schimel JP (2012) Detecting microbial N-limitation in tussock tundra soil: Implications for Arctic soil organic carbon cycling. *Soil Biology & Biochemistry*, **55**,

78-84.

- Song M, Xu X, Hu Q, Tian Y, Ouyang H, Zhou C (2007) Interactions of plant species mediated plant competition for inorganic nitrogen with soil microorganisms in an alpine meadow. *Plant and Soil*, **297**, 127-137.
- Spohn M, Chodak M (2015) Microbial respiration per unit biomass increases with carbon-to-nutrient ratios in forest soils. *Soil Biology & Biochemistry*, **81**, 128-133.
- Tian L, Zhao L, Wu X *et al.* (2017) Vertical patterns and controls of soil nutrients in alpine grassland: Implications for nutrient uptake. *Science of the Total Environment*, **607-608**, 855-864.
- Treat CC, Wollheim WM, Varner RK, Grandy AS, Talbot J, Frohling S (2014) Temperature and peat type control CO₂ and CH₄ production in Alaskan permafrost peats. *Global Change Biology*, **20**, 2674-2686.
- Trucco C, Schuur EaG, Natali SM, Belshe EF, Bracho R, Vogel J (2012) Seven-year trends of CO₂ exchange in a tundra ecosystem affected by long-term permafrost thaw. *Journal of Geophysical Research: Biogeosciences*, **117**, G02031.
- Vogel J, Schuur EaG, Trucco C, Lee H (2009) Response of CO₂ exchange in a tussock tundra ecosystem to permafrost thaw and thermokarst development. *Journal of Geophysical Research*, **114**.
- Wang C, Wang X, Liu D *et al.* (2014a) Aridity threshold in controlling ecosystem nitrogen cycling in arid and semi-arid grasslands. *Nature Communications*, **5**, 4799.
- Wang Q, Wang S, He T, Liu L, Wu J (2014b) Response of organic carbon mineralization and microbial community to leaf litter and nutrient additions in subtropical forest soils. *Soil Biology and Biochemistry*, **71**, 13-20.
- Wild B, Alves RJE, Bárta J *et al.* (2018) Amino acid production exceeds plant nitrogen demand in Siberian tundra. *Environmental Research Letters*, **13**, 034002.
- Wild B, Gentsch N, Capek P *et al.* (2016) Plant-derived compounds stimulate the decomposition of organic matter in arctic permafrost soils. *Scientific Reports*, **6**, 25607.
- Wild B, Schnecker J, Alves RJE *et al.* (2014) Input of easily available organic C and N stimulates microbial decomposition of soil organic matter in arctic permafrost soil. *Soil Biology & Biochemistry*, **75**, 143-151.
- Wild B, Schnecker J, Knoltsch A *et al.* (2015) Microbial nitrogen dynamics in organic and mineral soil horizons along a latitudinal transect in western Siberia. *Global Biogeochemical Cycles*, **29**, 567-582.
- Xu X, Thornton PE, Post WM (2013) A global analysis of soil microbial biomass carbon, nitrogen and phosphorus in terrestrial ecosystems. *Global Ecology and Biogeography*, **22**, 737-749.
- Yang G, Peng Y, Olefeldt D *et al.* (2018) Changes in Methane Flux along a Permafrost Thaw Sequence on the Tibetan Plateau. *Environmental Science & Technology*, **52**, 1244-1252.
- Yang Y, Fang J, Ji C, Han W (2009) Above- and belowground biomass allocation in Tibetan grasslands. *Journal of Vegetation Science*, **20**, 177-184.
- Yuan MM, Zhang J, Xue K *et al.* (2018) Microbial functional diversity covaries with permafrost thaw-induced environmental heterogeneity in tundra soil. *Global Change*

- Biology, **24**, 297-307.
- Zhang T, Barry RG, Knowles K, Heginbottom JA, Brown J (2008) Statistics and characteristics of permafrost and ground-ice distribution in the Northern Hemisphere. *Polar Geography*, **31**, 47-68.
- Zhu B, Cheng W (2011) Rhizosphere priming effect increases the temperature sensitivity of soil organic matter decomposition. *Global Change Biology*, **17**, 2172-2183.
- Zhu B, Cheng W (2013) Impacts of drying–wetting cycles on rhizosphere respiration and soil organic matter decomposition. *Soil Biology and Biochemistry*, **63**, 89-96.
- Zhu B, Gutknecht JLM, Herman DJ, Keck DC, Firestone MK, Cheng W (2014) Rhizosphere priming effects on soil carbon and nitrogen mineralization. *Soil Biology and Biochemistry*, **76**, 183-192.

Nature Research Editing Service Certification

This is to certify that the manuscript titled Nitrogen availability regulates topsoil carbon dynamics by altering microbial metabolic efficiency after permafrost collapse was edited for English language usage, grammar, spelling and punctuation by one or more native English-speaking editors at Nature Research Editing Service. The editors focused on correcting improper language and rephrasing awkward sentences, using their scientific training to point out passages that were confusing or vague. Every effort has been made to ensure that neither the research content nor the authors' intentions were altered in any way during the editing process.

Documents receiving this certification should be English-ready for publication; however, please note that the author has the ability to accept or reject our suggestions and changes. To verify the final edited version, please visit our verification page. If you have any questions or concerns over this edited document, please contact Nature Research Editing Service at support@as.springernature.com.

Manuscript title: Nitrogen availability regulates topsoil carbon dynamics by altering microbial metabolic efficiency after permafrost collapse

Authors: Leiyi Chen, Li Liu, Chao Mao, Shuqi Qin, Jun Wang, Futing Liu, Sergey Blagodatsky, Guibiao Yang, Qiwen Zhang, Dianye Zhang, Jianchun Yu, Yuanhe Yang*

Key: ABD3-A7EE-2E9C-71AF-816P

This certificate may be verified at secure.authorservices.springernature.com/certificate/verify.

Nature Research Editing Service is a service from Springer Nature, one of the world's leading research, educational and professional publishers. We have been a reliable provider of high-quality editing since 2008.

Nature Research Editing Service comprises a network of more than 900 language editors with a range of academic backgrounds. All our language editors are native English speakers and must meet strict selection criteria. We require that each editor has completed or is completing a Masters, Ph.D. or M.D. qualification, is affiliated with a top US university or research institute, and has undergone substantial editing training. To ensure we can meet the needs of researchers in a broad range of fields, we continually recruit editors to represent growing and new disciplines.

Uploaded manuscripts are reviewed by an editor with a relevant academic background. Our senior editors also quality-assess each edited manuscript before it is returned to the author to ensure that our high standards are maintained.

REVIEWERS' COMMENTS:

Reviewer #1 (Remarks to the Author):

My compliments to the authors for their thorough revision.

I have only a few remaining comments:

1. You replaced Fig3c with a data from a new experiment. Please keep the original data in the manuscript as well. You say that you added 3000 mg N / kg soil, but from the original Fig. 3a it seems that this was only 300 mg N /kg soil. Either way, please add the original Fig 3c data to your current Fig 3c.

2. See [comment 3 of the first review round]: please remove all mention of 'understanding the permafrost C-climate feedback' (e.g. l. 322) from this article. The Tibetan Plateau permafrost has little to do with this feedback (it represents < 1 % of expected losses from arctic permafrost soils, Chen 2016)and the ecosystem discussed in this manuscript is not comparable to most arctic permafrost soils (C-limitation, see next comment).

3. In your rebuttal letter you explicitly state that: "Compared to the arctic tundra organic soils, the microbial activity in Tibetan thermokarst region is still limited by C, indicated by relatively lower C content (18.7% in our study site vs. 21.4~48.6% in arctic tundra)and soil C:N ratio (12.3 in Tibetan swamp meadow vs. 25.6~41.3 in arctic tundra)(Hartley et al., 2010, De Baets et al., 2016, Wild et al., 2016)."

This is a well-references and interesting statement about the differences between the Tibetan Plateau permafrost and arctic tundra soils. I would like to see this mentioned explicitly in the Discussion of the manuscript as well.

4. The new Table (R1) is good and informative. Add also the number of samples the data are based on (n=), and remove the word 'typical' from the Legend (instead specify how the sampling locations relate to the sampling locations from which the samples for the main results were taken).

5. Fig. 6 is much improved conceptually, but is now of poor graphical/esthetical quality (pixelated, blocks visible) - please improve this?

Reviewer #3 (Remarks to the Author):

General

I find the manuscript much improved and I am satisfied with the authors' responses.

I have one remaining question about CUE: CUE is calculated here using information from enzyme data, microbial and substrate C/N ratios. To what extent is this parameter influenced by the enzyme data, and to what extent by the balance between microbial and substrate C/N? The authors argue that not changes in enzyme activities, but in microbial metabolic efficiency drive the observed influence of N availability on priming intensity, so this point should be more clearly addressed.

Some more minor comments are below.

Abstract

Line 36: Change to "ecosystems".

Introduction

Line 59: Change to "the soil C loss".

Line 71: "Nutrients".

Line 97-99: This sentence implies differences in priming intensity along the thermokarst gradient before this has been shown. Please rephrase to make it more neutral.

Line 99: Remove "that".

Results

Lines 147-148: Change to "Microbial N limitation, indicated by the stoichiometric imbalance ...".

Line 157: Change "was" to "were".

Line 171: Change to "between priming and post-thaw N availability".

Line 181: "The metabolic quotient".

Discussion

Lines 209-211: Are you comparing SOC loss or relative priming response?

Material & Methods

Line 328: Change to "meadows".

Line 335: Ref. 26 is not about N limitation of microbial activity in organic soils, and ref. 45 specifically states that "Microbial activity is not always limited by nitrogen in Arctic tundra soils", referring to the organic layer.

Lines 335-338: I don't understand how low C/N ratio, negative net N mineralization and high ratio of N immobilization over N mineralization suggest that C is co-limiting.

Line 349: The 78% value seems to contradict the 81% in line 339. They are very similar and I assume you are referring to different landscape units or studies, but it is still confusing.

Lines 350-351: Change to "The soil is a Gelisol". How can Hugelius 2014 (a paper about circumpolar SOC stocks) be the source of this information?

Line 354: Change to "a mixture" and "fluvial".

Lines 361-363: This is the third description of rooting depths at the site in the Method section. One is enough.

Line 363-364: Change to "Thus, our study site is well representative of swamp meadows, ..."

Line 373: "Photographs"

Line 418: Change to "a ¹³C-labelled glucose treatment".

Line 424: Change "were" to "was".

Line 448: I seem to remember that IRGA measures only ¹²CO₂, not ¹³CO₂. Considering that the enrichment was only 3 at% ¹³C this is probably negligible, but can you comment on the potential impact on your data?

Line 473: Delete the second "the".

Line 498: Change to "enzyme activities".

Line 501: Change to "enzyme activities".

Lines 509-512: Were the standards added to soil slurries to account for quenching or not? Please specify.

Line 510: Change to "standards".

Lines 539-540: Is labile organic matter C/N assumed to be DOC/TDN?

Line 553: Change to "by an Agilent 6890 gas chromatograph".

Line 583: Change to "the underlying role of".

Figures

Figure 1: Where is the difference between cumulative priming effect and CO₂-¹²C release? Does the latter also include ¹²C from the added substrate? Then it is not a very meaningful parameter.

Figure 2: Change to "in the glucose treatment".

Figure 5: Change to "in the N addition experiment".

Figure 6: Change to "At the early stage of permafrost collapse, microbial N limitation is relieved by

high N availability as a consequence of enhanced microbial N mineralization, further resulting in a lower C:N imbalance".

Response to Reviewer #1:

[Comment 1] My compliments to the authors for their thorough revision. I have only a few remaining comments: You replaced Fig3c with a data from a new experiment. Please keep the original data in the manuscript as well. You say that you added 3000 mg N / kg soil, but from the original Fig. 3a it seems that this was only 300 mg N /kg soil. Either way, please add the original Fig 3c data to your current Fig 3c.

[Response] We would like to mention that, **the rate of NH_4NO_3 addition in the original N experiment was set to 3000 mg N kg⁻¹ soil.** The data of ~300 mg N kg⁻¹ soil in original Fig. 3a is the concentration of total dissolved nitrogen (TDN) after 30-day incubation. This N concentration was much lower than the initial concentration because multiple processes (*e.g.*, microbial N immobilization, mineral N adsorption by soil colloid and N gas release) could result in the loss of TDN during the incubation. In support of this assumption, the TDN concentration determined at day 3 in the original N experiment was 2018.8 mg N kg⁻¹ soil (Fig. R1a). This excessive high N addition resulted in a sharp decrease in C:N imbalance ($R_{\text{DOC:TDN}}/B_{\text{C:N}}$) (Fig. R1b) and a consequent negative priming effect (Fig. R1c). These results were consistent with our conclusion that increased N availability after permafrost thaw inhibited the priming effect.

Despite of this, as mentioned in the last response letter, **the concentration of added N in the previous N addition experiment (3000 mg N kg⁻¹ soil) was far beyond the natural range of the difference in soil TDN concentration (20.1~50.8 mg N kg⁻¹ soil).** This high N concentration is not appropriate to test the hypothesis that “elevated N availability **after permafrost collapse** inhibits the priming effect by increasing the microbial metabolic efficiency”. Due to this fact, we prefer to only include the additional experimental data in the revised MS. Nevertheless, we could add those results derived from original experiment in the revised MS if the reviewer insists his/her opinion. Thanks for your understanding!

Figure R1. Soil TDN concentration and C:N imbalance and the priming effect from late-stage soils amended with different levels of N. Data are represented as the means \pm SE (standard error). Significant differences are denoted by different letters ($P < 0.05$). G, glucose; G+LN, glucose with low N addition; G+HN, glucose with high N addition; G+EHN, glucose with excessive high N addition.

[Comment 2] See [comment 3 of the first review round]: please remove all mention of 'understanding the permafrost C-climate feedback' (e.g. l. 322) from this article. The Tibetan Plateau permafrost has little to do with this feedback (it represents < 1 % of expected losses from arctic permafrost soils, Chen 2016) and the ecosystem discussed in this manuscript is not comparable to most arctic permafrost soils (C-limitation, see next comment).

[Response] Following the reviewer's comment, we have removed all the mention of "understanding the permafrost C-climate feedback" from the article.

[Comment 3] In your rebuttal letter you explicitly state that: "Compared to the arctic tundra organic soils, the microbial activity in Tibetan thermokarst region is still limited by C, indicated by relatively lower C content (18.7% in our study site vs. 21.4~48.6% in arctic tundra) and soil C:N ratio (12.3 in Tibetan swamp meadow vs. 25.6~41.3 in arctic tundra)(Hartley et al., 2010, De Baets et al., 2016, Wild et al., 2016)." This is a well-referenced and interesting statement about the differences between the Tibetan Plateau permafrost and arctic tundra soils. I would like to see this mentioned explicitly in the Discussion of the manuscript as well.

[Response] Following the reviewer's suggestion, we have explicitly stated this point in the *Discussion* session of MS as follows: "In contrast, the soil C content and C:N ratio of organic soils in our study site were 18.7% and 12.3, respectively (Table S1). These differences indicate that compared to the arctic tundra organic soils, the microbial activity in Tibetan thermokarst region is still limited by C (Chen *et al.*, 2016)" (Page 10, Line 196-199).

[Comment 4] The new Table (R1) is good and informative. Add also the number of samples the data are based on (n=), and remove the word 'typical' from the Legend (instead specify how the sampling locations relate to the sampling locations from which the samples for the main results were taken).

Fig. 6 is much improved conceptually, but is now of poor graphical/aesthetic quality (pixelated, blocks visible) - please improve this?

[Response] Done as suggested.

Response to Reviewer #3:

[Comment 1] I find the manuscript much improved and I am satisfied with the authors' responses. I have one remaining question about CUE: CUE is calculated here using information from enzyme data, microbial and substrate C/N ratios. To what extent is this parameter influenced by the enzyme data, and to what extent by the balance between microbial and substrate C/N? The authors argue that not changes in enzyme activities, but in microbial metabolic efficiency drive the observed influence of N availability on priming intensity, so this point should be more clearly addressed.

[Response] Following the reviewer's comments, we analyzed the relationship among CUE, enzyme variables and C:N imbalance to explore the extents of CUE influenced by enzyme data (*i.e.*, $EEA_{C:N}$, calculated as $BG / (NAG + LAP)$) and the C:N imbalance between microbe and substrate (*i.e.*, $L_{C:N} / B_{C:N}$). The regression analysis showed that **CUE was negatively correlated to the C:N imbalance (Fig. R2a-b), but was not associated with $EEA_{C:N}$ (Fig. R2c-d)**. These results further highlight the importance of C:N imbalance in regulating the microbial metabolic efficiency and priming effect. We have added these results in *Discussion* session of the revised MS (Page 12, Line 238-241).

Figure R2. Relationships between $CUE_{C:N}$ and C:N imbalance ($R_{DOC:TDN}/B_{C:N}$) (a-b) and enzyme stoichiometric ratio ($EEA_{C:N}$) (c-d) in gradient experiment and N addition experiment.

[Comment 2] Line 36: Change to “ecosystems”.

Line 59: Change to “the soil C loss”.

Line 71: “Nutrients”.

[Response] Done as suggested.

[Comment 3] Line 97-99: This sentence implies differences in priming intensity along the thermokarst gradient before this has been shown. Please rephrase to make it more neutral.

[Response] Following the reviewer’s comment, we have modified this sentence as follows: “shift in microbial metabolic efficiency is one of the potential pathways that post-thaw N availability could regulate soil C release after thermokarst formation.”

[Comment 4] Line 99: Remove “that”.

Lines 147-148: Change to “Microbial N limitation, indicated by the stoichiometric imbalance ...”.

Line 157: Change “was” to “were”.

Line 171: Change to “between priming and post-thaw N availability”.

Line 181: “The metabolic quotient”.

[Response] Done as suggested.

[Comment 5] Lines 209-211: Are you comparing SOC loss or relative priming response?

[Response] We are comparing relative priming response. To avoid this confusion, we have revised the sentence as follows: “*In our laboratory incubation, the supply of glucose induced positive priming effect on all collapse stages along the thermo-erosion gully, with the maximum relative priming intensity of 26.6% from the late-stage topsoil. The response in our organic soil was comparable to that of mineral soils (14~31%)(Wild et al., 2014, Wild et al., 2016)*” (Page 10, Line 186-189).

[Comment 6] Line 328: Change to “meadows”.

[Response] Done as suggested.

[Comment 7] Line 335: Ref. 26 is not about N limitation of microbial activity in organic soils, and ref. 45 specifically states that “Microbial activity is not always limited by nitrogen in Arctic tundra soils”, referring to the organic layer.

[Response] Following the reviewer's comment, we have removed these two references in the revised MS.

[Comment 8] Lines 335-338: *I don't understand how low C/N ratio, negative net N mineralization and high ratio of N immobilization over N mineralization suggest that C is co-limiting.*

[Response] Sorry for confusion. The microbial C limitation is mainly supported by the low C/N ratio. To be specific, **compared to the arctic tundra organic soils, the microbial activity in Tibetan thermokarst region is limited by C**, indicated by **relatively lower C content** (18.7% in our study site vs. 21.4~48.6 % in arctic tundra) **and soil C:N ratio** (12.3 in Tibetan swamp meadow vs. 25.6~41.3 in arctic tundra) (Hartley *et al.*, 2010, De Baets *et al.*, 2016, Wild *et al.*, 2016). **Meanwhile**, the negative net N mineralization rate in alpine meadow and wetland (Song *et al.*, 2007, Gao *et al.*, 2016), higher proportion of microbial N immobilization to total gross N mineralization in Tibetan alpine grasslands (1.6±0.2) (Kou *et al.*, 2017) and the increase in soil CO₂ flux after low N supply (Deng *et al.*, 2015b) jointly suggested that **N availability could also constrain the function of microbial communities in this region. Hence, microbial activity in Tibetan alpine permafrost is assumed to be co-limited by C and N availability.**

To avoid the confusion, we have revised the sentences as follows: *“However, unlike arctic and subarctic permafrost regions, where the microbial activity in organic soil is solely N limited (Sistla *et al.*, 2012), soil microbial activity in swamp meadow is co-limited by C and N availability, as indicated by the low soil C concentration and C:N ratio(Chen *et al.*, 2016, Tian *et al.*, 2017) (C limitation) combined with negative net N mineralization rate(Song *et al.*, 2007, Gao *et al.*, 2016) and the high ratio of N immobilization to total gross N mineralization(Kou *et al.*, 2017) (N limitation).”* (Page 16, Line 317-321).

[Comment 9] Line 349: *The 78% value seems to contradict the 81% in line 339. They are very similar and I assume you are referring to different landscape units or studies, but it is still confusing.*

[Response] To avoid the confusion, we unified this value to 81% across the whole revised MS.

[Comment 10] Lines 350-351: Change to “The soil is a Gelisol”. How can Hugelius 2014 (a paper about circumarctic SOC stocks) be the source of this information?

[Response] We have removed the reference in the revised MS.

[Comment 11] Line 354: Change to “a mixture” and “fluvial”.

[Response] Done as suggested.

[Comment 12] Lines 361-363: This is the third description of rooting depths at the site in the Method section. One is enough.

[Response] Following reviewer’s suggestion, we have deleted the duplicate descriptions of rooting depths.

[Comment 13] Line 363-364: Change to “Thus, our study site is well representative of swamp meadows, ...”

Line 373: “Photographs”

Line 418: Change to “a ¹³C-labelled glucose treatment”.

Line 424: Change “were” to “was”.

[Response] Done as suggested.

[Comment 14] Line 448: I seem to remember that IRGA measures only ¹²CO₂, not ¹³CO₂. Considering that the enrichment was only 3 at% ¹³C this is probably negligible, but can you comment on the potential impact on your data?

[Response] To eliminate this concern, we have consulted the manufacture of the infrared gas analyser (EGM-5; PP Systems, Haverhill, MA, USA). They confirmed that IRGA method could not distinguish between ¹²CO₂ and ¹³CO₂. In other words, the amount of CO₂ measured by IRGA is the sum of ¹²CO₂ and ¹³CO₂ and thus have no impact on our data.

[Comment 15] Line 473: Delete the second “the”.

Line 498: Change to “enzyme activities”.

Line 501: Change to “enzyme activities”.

[Response] Done as suggested.

[Comment 16] Lines 509-512: Were the standards added to soil slurries to account or quenching or not? Please specify.

[Response] Yes, the standards were added. We have revised the sentence as follows: “The standards (4-methylumbelliferone (MUB) for BG and NAG and 7-amino-4-methylcoumarin (AMC) for LAP) at concentrations of 0, 2.5, 5, 10, 25, 50, and 100 μ M were added to soil slurries to account for quenching.” (Page 24, Line 486-488).

[Comment 17] Line 510: Change to “standards”.

[Response] Done as suggested.

[Comment 18] Lines 539-540: Is labile organic matter C/N assumed to be DOC/TDN?

[Response] Yes, the labile organic matter C/N assumed to be DOC/TDN.

[Comment 19] Line 553: Change to “by an Agilent 6890 gas chromatograph”.

Line 583: Change to “the underlying role of”.

[Response] Done as suggested.

[Comment 20] Figure 1: Where is the difference between cumulative priming effect and CO₂-¹²C release? Does the latter also include ¹²C from the added substrate? Then it is not a very meaningful parameter.

[Response] The CO₂-¹²C release in panel (g) was the C release from soil organic matter in glucose addition treatment, which did not include the C release from added glucose. To avoid the confusion, we have deleted this panel in the revised MS.

[Comment 21] Figure 2: Change to “in the glucose treatment”.

Figure 5: Change to “in the N addition experiment”.

Figure 6: Change to “At the early stage of permafrost collapse, microbial N limitation is relieved by high N availability as a consequence of enhanced microbial N mineralization, further resulting in a lower C:N imbalance”.

[Response] Done as suggested.

References

- Chen YL, Chen LY, Peng YF *et al.* (2016) Linking microbial C:N:P stoichiometry to microbial community and abiotic factors along a 3500-km grassland transect on the Tibetan Plateau. *Global Ecology and Biogeography*, **25**, 1416-1427.
- Gao J, Feng J, Zhang X, Yu F-H, Xu X, Kuzyakov Y (2016) Drying-rewetting cycles

- alter carbon and nitrogen mineralization in litter-amended alpine wetland soil. *CATENA*, **145**, 285-290.
- Kou D, Peng Y, Wang G *et al.* (2017) Diverse responses of belowground internal nitrogen cycling to increasing aridity. *Soil Biology & Biochemistry*, **116**, 189-192.
- Sistla SA, Asao S, Schimel JP (2012) Detecting microbial N-limitation in tussock tundra soil: Implications for Arctic soil organic carbon cycling. *Soil Biology & Biochemistry*, **55**, 78-84.
- Song M, Xu X, Hu Q, Tian Y, Ouyang H, Zhou C (2007) Interactions of plant species mediated plant competition for inorganic nitrogen with soil microorganisms in an alpine meadow. *Plant and Soil*, **297**, 127-137.
- Tian L, Zhao L, Wu X *et al.* (2017) Vertical patterns and controls of soil nutrients in alpine grassland: Implications for nutrient uptake. *Science of the Total Environment*, **607-608**, 855-864.
- Wild B, Gentsch N, Capek P *et al.* (2016) Plant-derived compounds stimulate the decomposition of organic matter in arctic permafrost soils. *Scientific Reports*, **6**, 25607.
- Wild B, Schneckler J, Alves RJE *et al.* (2014) Input of easily available organic C and N stimulates microbial decomposition of soil organic matter in arctic permafrost soil. *Soil Biology & Biochemistry*, **75**, 143-151.